# *Crym*-positive striatal astrocytes gate perseverative behaviour

Matthias Ollivier[1], Joselyn S. Soto[1], Kay E. Linker[1], Stefanie L. Moye[1], Yasaman Jami-Alahmadi[2], Anthony E. Jones[3], Ajit S. Divakaruni[3], Riki Kawaguchi[4], James A. Wohlschlegel[2] & Baljit S. Khakh[1,5 ✉]

Astrocytes are heterogeneous glial cells of the central nervous system[1–3]. However, the physiological relevance of astrocyte diversity for neural circuits and behaviour remains unclear. Here we show that a specific population of astrocytes in the central striatum expresses μ-crystallin (encoded by *Crym* in mice and *CRYM* in humans) that is associated with several human diseases, including neuropsychiatric disorders[4–7]. In adult mice, reducing the levels of μ-crystallin in striatal astrocytes through CRISPR–Cas9-mediated knockout of *Crym* resulted in perseverative behaviours, increased fast synaptic excitation in medium spiny neurons and dysfunctional excitatory–inhibitory synaptic balance. Increased perseveration stemmed from the loss of astrocyte-gated control of neurotransmitter release from presynaptic terminals of orbitofrontal cortex–striatum projections. We found that perseveration could be remedied using presynaptic inhibitory chemogenetics[8], and that this treatment also corrected the synaptic deficits. Together, our findings reveal converging molecular, synaptic, circuit and behavioural mechanisms by which a molecularly defined and allocated population of striatal astrocytes gates perseveration phenotypes that accompany neuropsychiatric disorders[9–12]. Our data show that *Crym*-positive striatal astrocytes have key biological functions within the central nervous system, and uncover astrocyte–neuron interaction mechanisms that could be targeted in treatments for perseveration.

Astrocytes are vital components of neural circuits, and have essential roles in physiology and disease[3,13]. They interact with neurons in multiple species and are predominant glial cells that tile the central nervous system (CNS). Understanding how astrocytes contribute to physiological and pathological processes in the CNS is an emerging topic, with many fundamental open questions.

An important advance has been the discovery that astrocytes are heterogeneous, comprising multiple populations that exhibit diverse properties across different brain regions[1–3,14–18]. Collectively, these data show that specialized astrocytes exist in specific brain regions, and identify their associated neural circuits. However, these studies do not reveal the functions of defined astrocytes in any brain region. Furthermore, how specific populations of astrocytes contribute to the neural circuits in which they reside, and the consequences for physiology, behaviour and disease, remain unclear.

Here, through several lines of evidence, we report the discovery of precisely anatomically allocated *Crym*-positive striatal astrocytes. We show that this population of astrocytes regulates perseveration phenotypes that exist in psychiatric and neurological disorders[9–12], including obsessive-compulsive disorder (OCD) and Huntington's disease (HD), in which the expression of *CRYM* is reduced in post-mortem human tissue[5–7]. Our study reveals the synaptic mechanism by which *Crym*-positive astrocytes gate perseveration, and sheds light on the physiology of μ-crystallin—a hitherto incompletely investigated protein that is associated with multiple human disorders—in the CNS[4].

## *Crym*⁺ striatal astrocytes

Evidence for astrocyte diversity is derived from studies of gene expression in various regions of the CNS in adult mice[1–3,14–18]. Within the striatum, the largest nucleus of the basal ganglia, astrocytes are separable, expressing distinct sets of genes relative to other areas of the CNS[14,15]. In particular, striatal astrocytes highly express a gene called *Crym* (Fig. 1a), which encodes μ-crystallin (ref. 4). μ-crystallin is a poorly understood cytosolic protein that has been suggested to function as a ketimine reductase[19] or to bind NADPH to buffer the thyroid hormone T3, thus controlling the effects of T3 on gene expression[4,20]. *CRYM* is associated with muscle dysfunction, malignancy and non-syndromic deafness[4,20]. In the mouse CNS, *Crym* is found in populations of cortical, hippocampal and amygdala neurons[21,22]. In humans, *CRYM* is highly expressed in the striatum, and has been associated with disorders such as HD, amyotrophic lateral sclerosis and schizophrenia[4]. Little is known about brain μ-crystallin, and nothing is known about the physiology of *Crym*⁺ striatal astrocytes.

[1]Department of Physiology, David Geffen School of Medicine, University of California, Los Angeles, Los Angeles, CA, USA. [2]Department of Biological Chemistry, David Geffen School of Medicine, University of California, Los Angeles, Los Angeles, CA, USA. [3]Department of Molecular and Medical Pharmacology, David Geffen School of Medicine, University of California, Los Angeles, Los Angeles, CA, USA. [4]Center for Neurobehavioral Genetics, Semel Institute for Neuroscience and Human Behavior, David Geffen School of Medicine, University of California, Los Angeles, Los Angeles, CA, USA. [5]Department of Neurobiology, David Geffen School of Medicine, University of California, Los Angeles, Los Angeles, CA, USA. ✉e-mail: bkhakh@mednet.ucla.edu

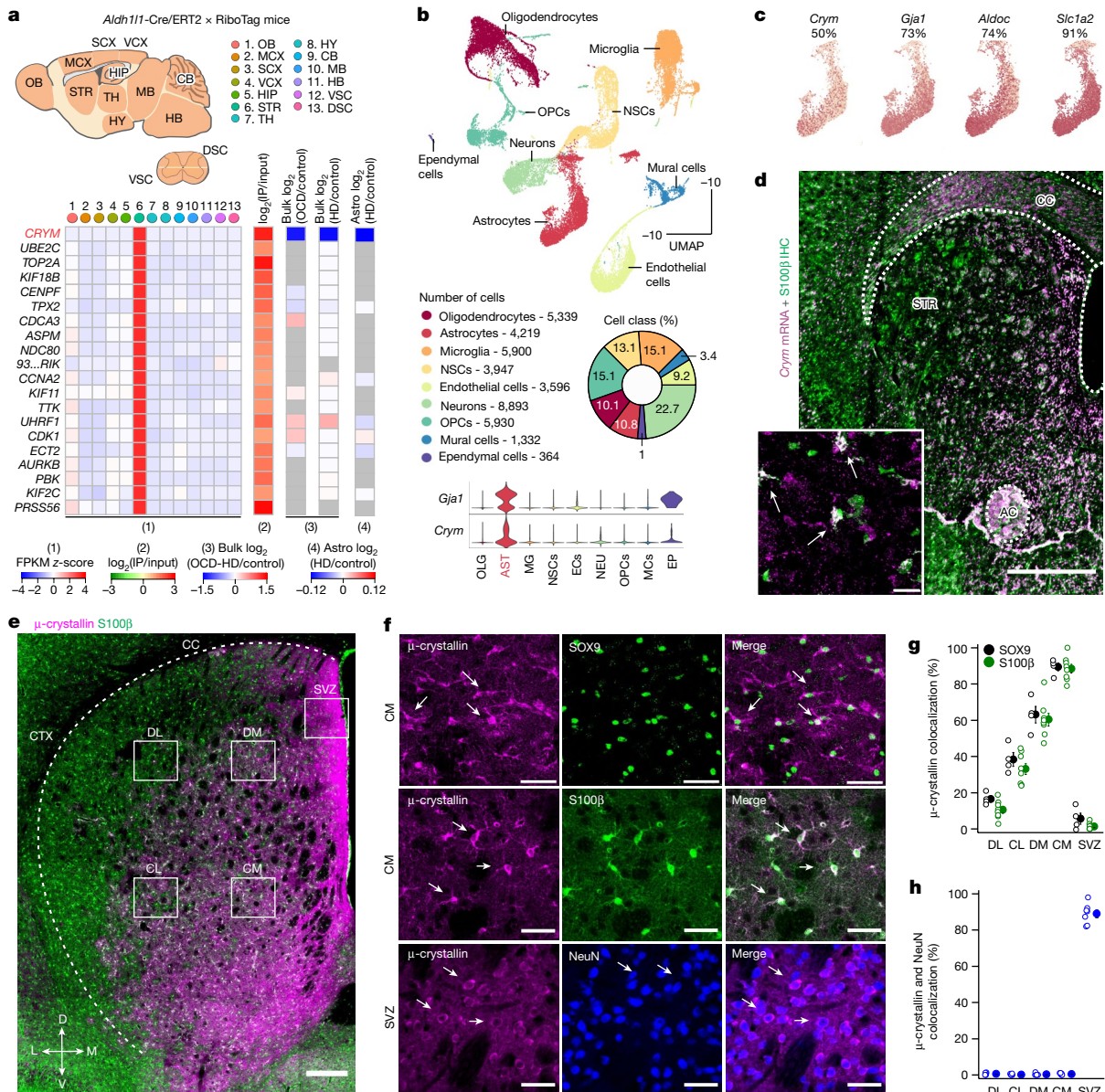

**Fig. 1 | A molecularly defined and allocated *Crym*⁺ population of striatal astrocytes. a**, Top 20 striatal-astrocyte-enriched genes from RNA-seq[15] of 13 brain areas: OB, olfactory bulb; MCX, motor cortex; SCX, somatosensory cortex; VCX, visual cortex; HIP, hippocampus; STR, striatum; TH, thalamus; HY, hypothalamus; CB, cerebellum; MB, midbrain; HB, hindbrain; DSC, dorsal spinal cord; VSC, ventral spinal cord. The fragments per kilobase per million mapped fragments (FPKM) *z*-score shows the 20 striatal-astrocyte-enriched genes as compared with other areas and their enrichment (log₂ immunoprecipitated (IP)/input). The orphan gene 9330182L06RiK has been abbreviated as 93...RiK in **a**. The other heat maps show the genes plotted from bulk RNA-seq data from human OCD[5] and HD[6] and in astrocytes from individuals with HD[7] relative to control individuals. **b**, scRNA-seq of two-month-old striatum optimized for cellular diversity[23], as seen in cell-class percentages. Uniform manifold approximation and projection (UMAP) of 39,156 cells from the striatum shows cell classes, including astrocytes. Violin plots show astrocyte enrichment of *Gja1* and *Crym* (*n* = 4 mice). OLG, oligodendrocytes; AST, astrocytes; MG, microglia; NSCs, neural stem cells; ECs, endothelial cells; NEU, neurons; OPCs,

oligodendrocyte precursor cells; MCs, mural cells; EP, ependymal cells. **c**, *Crym* was expressed in around 50% of astrocytes; astrocytic markers were found in more. **d**, *Crym* mRNA expression along with S100β IHC; inset higher magnification (representative of *n* = 16 sections from 4 mice). Scale bars, 500 μm (main); 20 μm (inset). CC, corpus callosum. AC, anterior commissure. **e**, Images of μ-crystallin (magenta) and astrocytic S100β (green) IHC. μ-crystallin shows a dorsoventral and lateromedial spatial gradient (CTX, cortex; SVZ, subventricular zone; DL, dorsolateral striatum; DM, dorsomedial striatum; CL, centrolateral striatum; CM, centromedial striatum). Scale bar, 200 μm. **f**, Magnified images of μ-crystallin (magenta) and SOX9 (green), and μ-crystallin and S100β (green) IHC in the centromedial striatum (CM). μ-crystallin and NeuN IHC in the SVZ are also shown. Scale bars, 40 μm. **g**,**h**, Percentage of astrocytes (**g**; S100β⁺ and SOX9⁺) and neurons (**h**; NeuN⁺) expressing μ-crystallin (*n* = 4, 8 and 7 mice for SOX9, S100β and NeuN, respectively). The white arrows in panels d and f point to instances of co-localization in the images. Average data shown as mean ± s.e.m. and all statistics reported in Supplementary Table 5.

## Regionally allocated *Crym*⁺ striatal astrocytes

Because striatal astrocytes express disease-related genes[15,23], we analysed post-mortem striatal data for individuals with OCD or HD and evaluated the top 20 genes enriched in striatal astrocytes[5–7]. Of these

genes, *CRYM* was downregulated to similar levels (about 40% of control) in the caudate nucleus in individuals with OCD and those with HD[5–7], and was within the top 4% of downregulated genes—including in mouse models of HD[7,24,25]—implying that *CRYM*⁺ astrocytes have key functions[4] (Fig. 1a). Using single-cell RNA sequencing (scRNA-seq) in

mice to preferentially sample non-neuronal cells[23], we found that *Crym* was highly expressed in *Gja1*[+] astrocytes, but largely absent in other cells (Fig. 1b). *Crym* was expressed in around 50% of astrocytes[15,23]—a lower percentage as compared with several markers of astrocytes (Fig. 1c). This shows that *Crym* expression demarcates a specific population of cells—a finding that was confirmed by RNAscope, which showed that *Crym* was expressed in around 46% of S100β[+] astrocytes, with a specific allocation within the striatum (Fig. 1d). Using immunohistochemistry (IHC), we found that μ-crystallin-expressing astrocytes in the striatum represented around 49% of total astrocytes and were anatomically located, being essentially absent in the dorsolateral regions and enriched ventrally and in the central region (Fig. 1e). In the central striatum, μ-crystallin was expressed in around 90% of S100β[+] and SOX9[+] astrocytes, but was not expressed in any neurons or oligodendrocytes (Fig. 1f–h and Extended Data Fig. 1a,b). Consistent with the scRNA-seq data (Fig. 1b) and previous studies[24,26,27], we also identified a population of μ-crystallin-expressing neurons in the subventricular zone (SVZ) (Fig. 1e–h). The expression of μ-crystallin in striatal astrocytes increased between postnatal day (P)7 and P15 and did not change with age or sex (Extended Data Figs. 1, 2 and 3). GFP[+] astrocytes in *Crym*-GFP reporter mice[21] showed the same striatal location as was seen in the RNAscope and IHC experiments, and exhibited bushy morphologies, S100β immunostaining and the expected electrophysiological properties (Extended Data Fig. 4a–c). Furthermore, although *Crym*[+] astrocytes existed in the striatal striosome and matrix compartments[28], they were more dominant in the matrix compartment (Extended Data Fig. 4d,e), and μ-crystallin-expressing astrocyte territories contained equal numbers of D1 and D2 medium spiny neurons (MSNs)[14,29] (Extended Data Fig. 4f,g). On the basis of RiboTag RNA-seq, scRNA-seq, RNAscope, IHC, evaluations during development and ageing, a GFP reporter mouse and electrophysiology, we thus point to the existence of a molecularly defined and anatomically allocated population of *Crym*[+] striatal astrocytes (Fig. 1 and Extended Data Figs. 1–4).

## Reduction of μ-crystallin in striatal astrocytes

We investigated striatal *Crym*[+] astrocytes using a loss-of-function approach[30,31] to reduce the expression of μ-crystallin in the striatum in mice in vivo (hereafter, *Crym* knockout; *Crym* KO). To reduce the expression of μ-crystallin, we used Cas9-GFP mice[32] together with local adeno-associated virus (AAV) 2/5-mediated[33] delivery of three single-guide RNAs (sgRNAs) that target *Crym* in astrocytes. Three weeks after microinjections of sgRNA AAVs targeting *Crym* in the central striatum, we detected an approximately 80% reduction of μ-crystallin in astrocytes, but not in the SVZ (which was not targeted) (Fig. 2a,b). Subsequently, we compared *Crym* KO mice with control mice that received sgRNA AAVs against GFP (Extended Data Fig. 4h,i).

## Perseveration-related behaviours

*Crym* KO and control mice (Fig. 2a,b) weighed the same, appeared healthy and were indistinguishable in the open-field, footprint and rotarod tests, indicating that motor function and motor learning were normal (Extended Data Fig. 5a,b and Fig. 2c–e). However, in the marble-burying test[34], we detected significantly more buried marbles in *Crym* KO mice (Fig. 2f,g), associated with a shorter latency to start and longer total digging durations (Fig. 2h). We also recorded longer self-grooming durations in *Crym* KO mice (Fig. 2i,j). Spray-evoked grooming was equivalent in control and *Crym* KO mice, implying that differences between groups were not due to differences in ability or drive to groom when evoked (Extended Data Fig. 5c). Consistent with these results, *Crym* KO mice spent more time licking the water bottle spout, with more frequent lick bouts and longer lick durations (Fig. 2k,l and Extended Data Fig. 5e), but there were no differences relative to controls after water deprivation, showing that thirst-evoked licking

was normal (Fig. 2l). *Crym* KO mice were also deficient in the novel object recognition task (Fig. 2m). This result suggests that the mice spent a prolonged time with the familiar object, which could reflect perseveration on that object (Fig. 2f–j).

The lack of anxiety-related phenotypes in *Crym* KO mice with open-field analyses was reproduced with the elevated plus maze (Extended Data Fig. 5d). This is notable with regard to OCD model mice[35], which exhibit repetitive and anxiety phenotypes (Extended Data Fig. 5f). Such distinctions were supported by the finding that fluoxetine did not correct self-grooming in *Crym* KO mice, whereas in OCD model mice it did[35] (Extended Data Fig. 5g). The findings of normal motor function and a lack of anxiety-related phenotypes, along with increased marble burying, digging, self-grooming and licking, and deficits in the novel object recognition task, indicate that astrocyte-specific *Crym* KO resulted in abnormal perseverative behaviours. Perseveration represents the inappropriate continuation or repetition of a response or activity and is associated with psychiatric and neurological disorders such as Tourette's syndrome, autism, OCD, HD and suicide-associated perseveration in HD[9–12]. Our data reveal that the loss of μ-crystallin in striatal astrocytes leads to perseveration, which is of relevance to HD and OCD (Fig. 1a).

## Altered lateral OFC–striatum synapses

We found no evidence of apoptosis or of neuron or astrocyte loss in the striatum of *Crym* KO mice (Extended Data Fig. 6a–h). Furthermore, the morphology and electrophysiology of astrocytes were essentially normal (Extended Data Fig. 7a,b,g). We detected only subtle changes in astrocyte Ca[2+] signalling (Extended Data Fig. 7c–f,g), and there was no change in the expression of astrocytic markers in *Crym* KO mice relative to controls (Extended Data Fig. 7h,i). In the absence of notable astrocyte alterations, we considered whether *Crym*[+] astrocytes exert effects on neuronal function.

The striatum is part of the basal ganglia corticostriatal–thalamocortical loop, receiving cortical input arriving at multiple locations[36]. To determine whether *Crym* KO altered this loop, we used cFOS mapping and found more cFOS[+] neurons in the lateral orbitofrontal cortex (lOFC), central striatum and dorsal thalamus (dTH) in *Crym* KO compared with control mice (Fig. 3a,b). To identify the origin of cortical inputs arriving to striatal areas with *Crym*[+] astrocytes (Fig. 1), we microinjected CTB-647 (transported in a retrograde manner) into the central striatum and identified CTB-647[+] neurons, with dense labelling in the lOFC and lower levels in the M1 motor cortex (Extended Data Fig. 8a,b). We also microinjected AAVs to express ChR2-mCherry (transported in an anterograde manner) into the lOFC and Chronos-GFP into M1, and quantified striatal projections (Extended Data Fig. 8c,d). lOFC projections invaded the central striatum, where the number of *Crym*[+] astrocytes was high (Extended Data Figs. 8d and 12c,d). These findings suggest that *Crym* KO may alter communication between the lOFC and the central striatum. Excessive activation of the OFC-to-ventromedial striatum projection is implicated in OCD[37].

There were no differences in the excitability and membrane properties of MSNs between *Crym* KO and control mice (Extended Data Fig. 9a,b). We next measured AMPA-receptor-mediated excitatory postsynaptic currents (EPSCs) arriving at MSNs during optogenetic stimulation of lOFC–striatal inputs (Fig. 3c). Although there were no differences in the amplitudes of single EPSCs between *Crym* KO and control mice (Fig. 3d), the second EPSC during paired responses was smaller, such that the paired-pulse ratio (PPR) was reduced in *Crym* KO mice (Fig. 3e). Because PPRs reflect the presynaptic release probability[38], these data indicate that the release probability of lOFC–striatal inputs was increased in *Crym* KO mice. By contrast, PPR values of M1–striatal inputs were unaltered (Extended Data Fig. 9c–e). Consistent with the higher release probability, we measured an increased frequency of spontaneous and miniature ESPCs (sEPSCs and mEPSCs, respectively) arriving at MSNs in the central striatum (Fig. 3f,g and

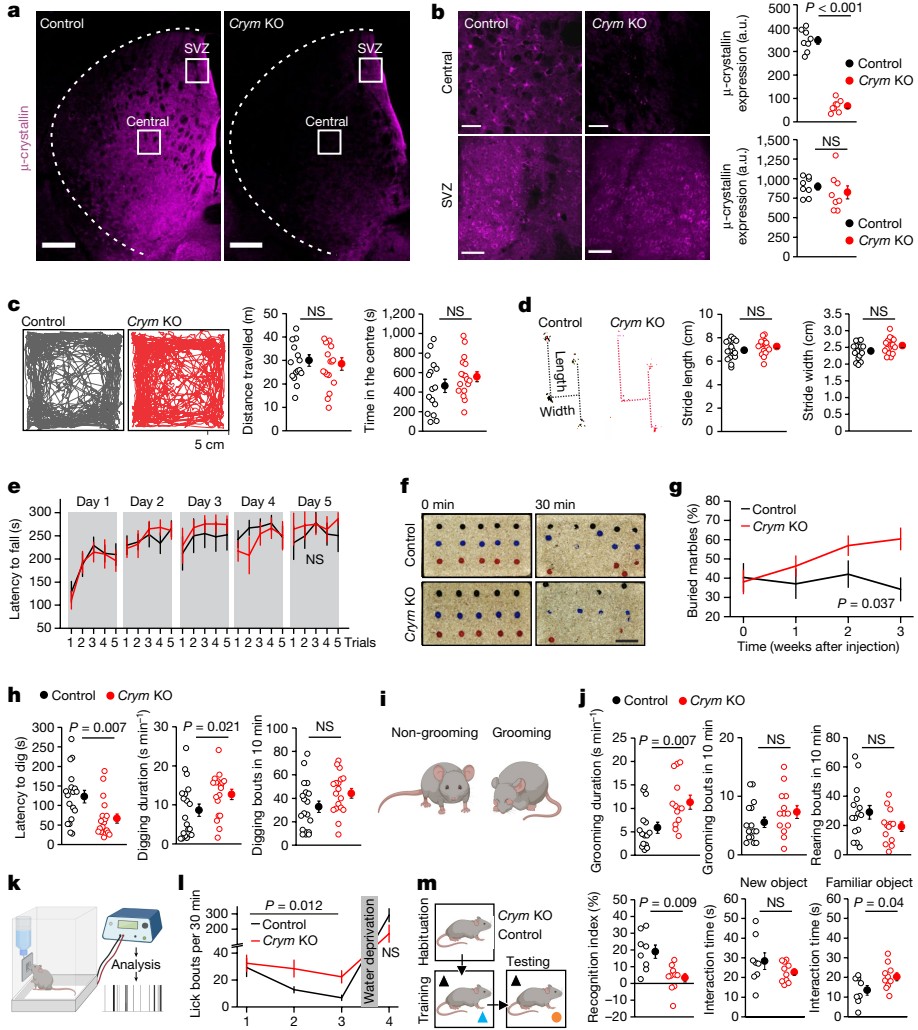

**Fig. 2 | Deletion of astrocytic *Crym* in the central striatum induces perseveration. a**, Striatal expression of μ-crystallin in mice injected with control sgRNA-GFP or *Crym* KO sgRNA-*Crym* AAVs. Scale bars, 200 μm. **b**, μ-Crystallin was reduced in the central striatum but not in the SVZ in *Crym* KO mice (*n* = 8 mice; two-tailed two-sample *t*-test; *P* = 5.3 × 10⁻¹⁰ for μ-crystallin). a.u., arbitrary units. NS, not significant. Scale bars, 50 μm. **c**, Traces of 30-min open-field recordings for control and *Crym* KO mice. Graphs of travel distance and time spent in the centre (*n* = 16 control and *n* = 17 *Crym* KO; two-tailed two-sample *t*-test). **d**, Footprint tests for control and *Crym* KO mice (*n* = 16 control and *n* = 17 *Crym* KO; two-tailed two-sample *t*-test). **e**, Time on the rotarod (*n* = 8 mice; two-way repeated-measures ANOVA followed by Tukey's post-hoc test). **f**, Marble-burying tests in control and *Crym* KO mice. Scale bar, 10 cm. **g**, Buried marbles before and after AAV injection (*n* = 16 mice; two-way repeated-measures ANOVA followed by Tukey's post-hoc test). **h**, Latency to start, total duration and digging bouts over 10 min in control (*n* = 19) and *Crym* KO (*n* = 18) mice (two-tailed Mann–Whitney and two-tailed two-sample *t*-test). **i**, Cartoons of self-grooming behaviour: mice disengaged and engaged in self-grooming. **j**, Self-grooming duration, grooming bouts and rearing bouts in control (*n* = 15) and *Crym* KO (*n* = 13) mice (two-tailed Mann–Whitney and two-tailed two-sample *t*-test). **k**, Schematic of the lickometer. **l**, Lick bouts and the total drinking time over 30 min for each trial (*n* = 8; two-way ANOVA). **m**, Evaluations of novel object recognition. Graphs show recognition index (%) and the interaction time with the new and familiar object (*n* = 8 control and *n* = 10 *Crym* KO; two-tailed Mann–Whitney and two-tailed two-sample *t*-test). Average data shown as mean ± s.e.m. and all statistics reported in Supplementary Table 5.

Extended Data Figs. 9f,g & 13f,g), indicative of increased synaptic release of glutamate from lOFC inputs. In accordance with this, we found no differences in sEPSC and mEPSC amplitudes, MSN dendritic complexity, spine density or spine head size between *Crym* KO and control mice (Fig. 3f,g and Extended Data Figs. 9f–j & 13f–j). We also measured fast GABA_A receptor-mediated spontaneous inhibitory post-synaptic currents (sIPSCs), and noted a decreased frequency onto MSNs in *Crym* KO versus control mice (Extended Data Fig. 9k,l). The dual effect of increased sEPSCs and decreased sIPSCs resulted in increased excitatory/inhibitory (E/I) synaptic ratios after *Crym* KO (Fig. 3h). This could explain the increased expression of cFOS in MSNs after *Crym* KO (Fig. 3a,b), and recalls E/I alterations in psychiatric disorders[39,40]. To investigate bulk striatal changes in glutamate and GABA, we used

gas chromatography–mass spectrometry (GC–MS) analysis of striatal samples from *Crym* KO and control mice. The glutamate/GABA ratio was significantly increased in *Crym* KO mice, (Extended Data Fig. 9m,n), whereas other metabolites were largely unchanged. Thus, *Crym* KO results in an increased E/I ratio and an increased glutamate release probability from lOFC terminals onto MSNs (Fig. 3).

## Presynaptic mechanism

During electrophysiology, we noticed decreased MSN tonic GABA currents in *Crym* KO mice as compared with control mice, indicating lower extracellular levels of GABA in the central striatum (Fig. 3i,j). The tonic GABA currents were blocked in control mice by pre-exposure to

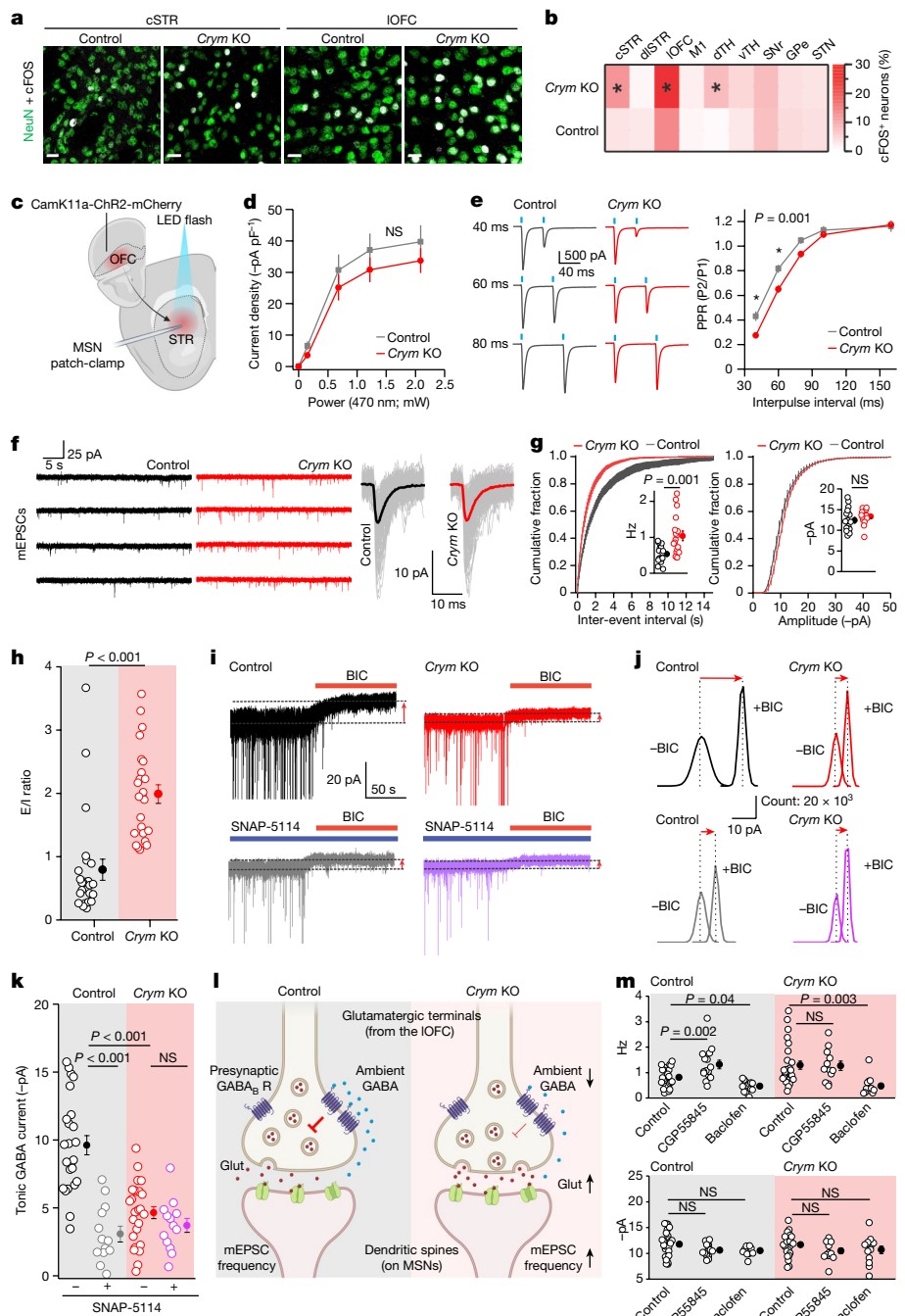

**Fig. 3 | Astrocytic loss of μ-crystallin alters lOFC–striatum synapses.**
**a**, cFOS and NeuN in the central striatum (cSTR) and lateral orbitofrontal cortex (lOFC). Scale bars, 20 μm. **b**, Heat map: percentage of cFOS+ neurons. * indicates $P < 0.05$ (dlSTR, dorsolateral striatum; M1, motor cortex 1; dTH, dorsal thalamus; vTH, ventral thalamus; SNr, substantia nigra reticulate; GPe, globus pallidus external; STN, subthalamic nucleus). SNr and STN, $n = 5$ mice both groups; lateral and dorsal thalamus and GPe, $n = 6$ mice both groups; M1 and striatum $n = 8$ mice both groups, lOFC $n = 7$ control mice and 8 Crym KO mice; two-sample t-tests and Mann–Whitney test). **c**, Injection of AAV2-CamK11a-ChR2-mCherry into the lOFC with recordings from the central striatum. **d**, EPSCs after 2-ms light pulses ($n = 18$ cells (control) and $n = 17$ cells (Crym KO) from 5 mice; two-way repeated-measures ANOVA). **e**, Representative data for evoked EPSCs ($n = 17$ cells (control) and $n = 18$ cells (Crym KO) from 5 mice; two-way ANOVA, *$P < 0.05$). **f**, mEPSC traces and averages from one representative MSN. **g**, Cumulative probability graphs for inter-event interval

and amplitude; inset shows pooled data ($n = 18$ cells from 5 mice for both control and Crym KO; two-tailed Mann–Whitney test). **h**, Excitatory/inhibitory (E/I) ratios ($n = 24$ cells from 5 mice; two-tailed Mann–Whitney test; $P = 1.5 \times 10^{-6}$). **i,j**, Representative data (**i**) and histograms (**j**) used to measure tonic GABA currents before and after GAT3 inhibition by SNAP-5114. BIC, bicuculline. **k**, Tonic GABA currents from experiments such as those in **i** (control and Crym KO: $n = 26$ cells from 8 mice; control and Crym KO treated with SNAP-5114: $n = 13$ cells from 4 mice; two-way ANOVA with Tukey's post-hoc test, overall ANOVA $P = 2 \times 10^{-11}$). **l**, In control mice, ambient GABA inhibits the release of glutamate (Glut) through the activation of presynaptic GABA_B receptors (GABA_B R). Crym KO mice show decreased ambient GABA-induced presynaptic inhibition and increased glutamate release. **m**, mEPSC frequency (top) and amplitude (bottom) before and after treatment with a GABA_B antagonist (CGP55845) or agonist (R-baclofen) ($n = 12$–29 cells from 4 mice; one-way ANOVA with Tukey's post-hoc test). Average data shown as mean ± s.e.m.

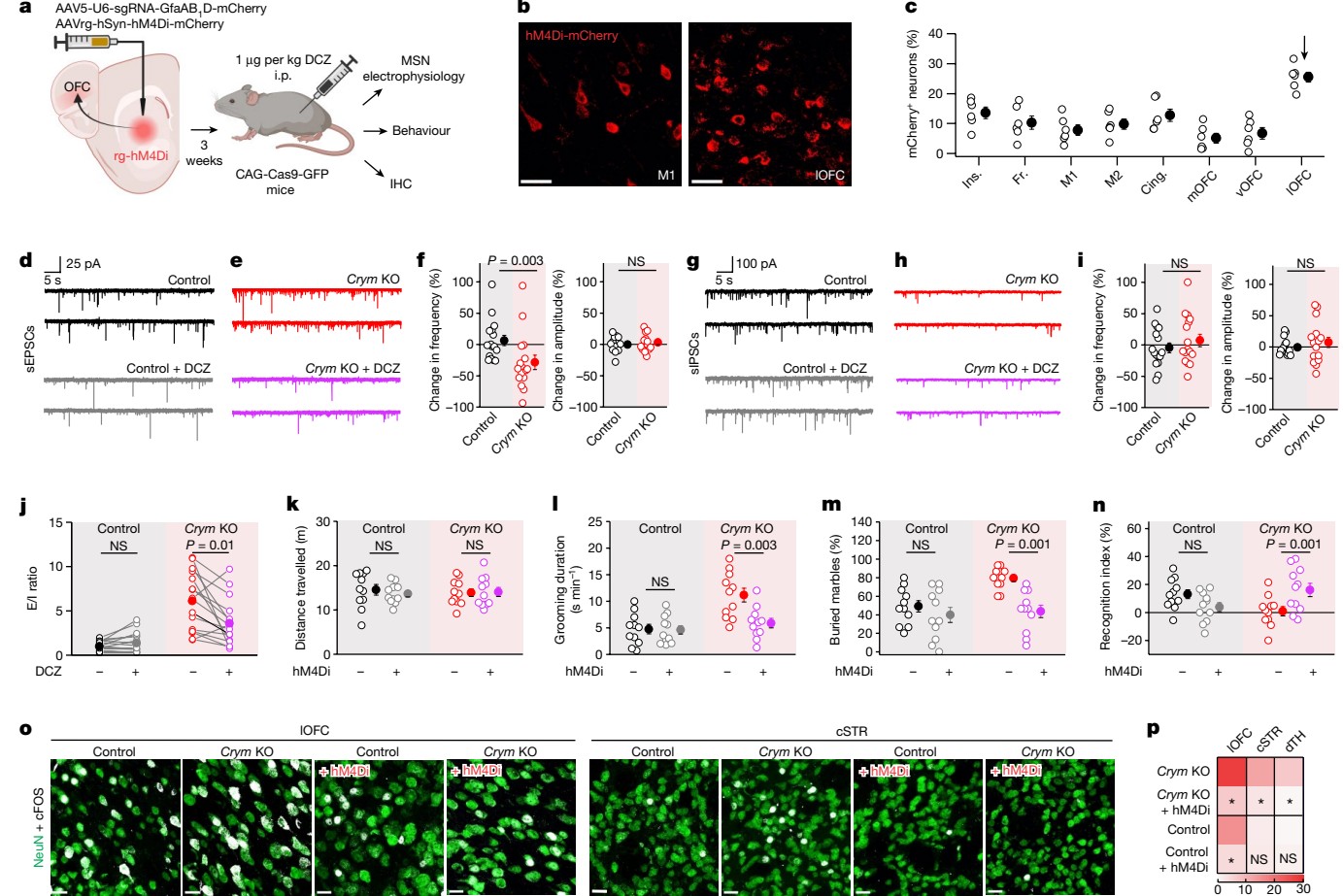

**Fig. 4 | Presynaptic chemogenetics corrects lOFC–striatum synaptic communication in *Crym* KO mice. a**, Approach for the co-expression of retrograde (rg) hM4Di-mCherry (or mCherry as control) and sgRNAs against *Crym* (or against GFP as control). **b**, Retrograde labelling of neurons in M1 and the lOFC using rg-hM4Di-mCherry. Scale bars, 20 μm. **c**, hM4Di-mCherry-positive neurons in various parts of the cortex (*n* = 6 mice). Ins, insular cortex. Fr, frontal cortex. M1, motor cortex 1. M2, motor cortex 2. Cing, cingulate cortex. mOFC, medial orbitofrontal cortex. vOFC, ventral orbitofrontal cortex. lOFC, lateral orbitofrontal cortex. **d**, sEPSC current waveforms from a representative MSN in control mice before (top) and during (bottom) treatment with DCZ (200 nM). **e**, As in **d**, but for *Crym* KO. **f**, Per cent change in sEPSC frequency (left) or amplitude (right) after DCZ applications for control and *Crym* KO (*n* = 16 cells from 4 mice; two-tailed Mann–Whitney or two-tailed 2-sample t-tests). **g–i**, As in **d–f**, but for sIPSCs (*n* = 16 cells from 4 mice; two-tailed

Mann–Whitney or two-tailed two-sample *t*-tests). **j**, Excitatory/inhibitory (E/I) ratio in control and *Crym* KO mice before and after DCZ applications (*n* = 16 paired cells from 4 mice; two-tailed paired-sample *t*-test; *P* = 6.6 × 10⁻³ for *Crym* KO). **k–n**, Distance travelled in open-field test (**k**), grooming (**l**), marble burying (**m**) and novel object recognition (**n**) behaviours of control and *Crym* KO mice without or with rg-hM4Di after treatment with DCZ (*n* = 11 mice per group; two-way ANOVA followed by Tukey's post-hoc test). **o**, cFOS (white) and NeuN (green) expression in the lOFC (left) and the central striatum (right) in control and *Crym* KO mice without or with rg-hM4Di activation in vivo. Scale bars, 20 μm. **p**, Percentage of cFOS⁺ neurons in the four conditions for the lOFC, cSTR and dTH. *\*P* < 0.05 (*n* = 6 mice; two-way ANOVA followed by Tukey's post-hoc test, and one-way ANOVA per brain area). Average data shown as mean ± s.e.m. and all statistics reported in Supplementary Table 5.

the astrocytic[41] GABA transporter type 3 (GAT3) antagonist[42] SNAP-5114 (40 μM) (Fig. 3i–k). However, the reduced tonic GABA currents in *Crym* KO mice were spared (Fig. 3i–k), indicating that GAT3 within the central striatum contributes GABA to the extracellular space[41,42], and that such contributions are reduced in *Crym* KO mice (Fig. 3i–k). Although there were no changes in the expression of GAT3 within astrocytes of *Crym* KO mice (Extended Data Fig. 10a,b), we detected a significant reduction in GABA and monoamine oxidase B (MAOB) (Extended Data Fig. 10c–f). MAOB is an astrocytic enzyme[43–45] that generates GABA, implying that reduced levels of tonic GABA in *Crym* KO mice reflect a reduced GAT3-dependent contribution of GABA to the extracellular space, as well as reduced GABA. Thus, increased synaptic excitation onto MSNs was driven by increased release from lOFC terminals after *Crym* KO (Fig. 3c–g), and was accompanied by lower levels of tonic GABA (Fig. 3i–k). Because GABA acts on presynaptic GABA_B receptors to decrease release probability[46], we hypothesized that lower levels of

tonic GABA result in reduced presynaptic inhibition and an increase in EPSCs emanating from lOFC terminals (Fig. 3l). Consistent with this, the GABA_B receptor antagonist CGP55845, which blocks ongoing activation of the GABA_B receptor, increased the frequency of mEPSCs in controls by 62 ± 18%, not in *Crym* KO mice (0.2 ± 7.6%; Fig. 3m). Furthermore, the GABA_B receptor agonist baclofen decreased the frequency of mEPSCs by 45 ± 7% in controls and 60 ± 4% in *Crym* KO mice (Fig. 3m). There were no changes in mEPSC amplitudes, consistent with a presynaptic mechanism (Fig. 3m). Our data show that the increased EPSCs arriving at MSNs in *Crym* KO mice are from lOFC terminals, reflecting reduced presynaptic inhibition owing to lower striatal levels of GABA (Fig. 3).

## Corrective presynaptic chemogenetics

To test whether reduced presynaptic inhibition was causal for the phenotypes observed in *Crym* KO mice, we designed an experiment to

restore it. We microinjected AAVs for retrograde inhibitory DREADD (rg-hM4Di) into the central striatum (Fig. 4a–c), and found that lOFC neurons were labelled abundantly (Fig. 4b,c). Next, we assessed sEP-SCs arriving at MSNs in control and *Crym* KO mice before and during the activation of rg-hM4Di. The DREADD agonist deschlorocloazapine (DCZ) (200 nM) significantly decreased the frequency of sEPSCs in *Crym* KO mice by around 30%, but had no effect in controls, in which our data show that release was suppressed by GABA (Fig. 4d,f). EPSC amplitudes were not affected (Fig. 4d–f and Extended Data Fig. 10g). As expected with hM4Di expression in lOFC terminals, IPSCs were unaffected (Fig. 4g–i and Extended Data Fig. 10h). Moreover, restoring the presynaptic inhibition of excitatory lOFC inputs onto MSNs in *Crym* KO mice with rg-hM4Di reversed the increase in the, E/I ratio in *Crym* KO, towards the value seen in control mice (Fig. 4j). We next investigated the link between the presynaptic mechanism and perseverative behaviours in *Crym* KO mice (Fig. 2). DCZ (1 μg per kg) administered to *Crym* KO mice that had received rg-hM4Di AAVs in the central striatum reversed the excessive self-grooming and marble-burying behaviours, and the deficit seen in the novel object recognition test, but had no effect on open-field assessments (Fig. 4k–n and Extended Data Fig. 10i–m). Furthermore, DCZ reduced the number of cFOS[+] neurons that were observed in the lOFC, central striatum and dorsal thalamus of *Crym* KO mice towards that seen in control mice (Fig. 4o,p). These data show that lOFC-targeted presynaptic inhibitory chemogenetics restored synaptic excitation, E/I ratios, cFOS levels and key behavioural phenotypes in *Crym* KO mice.

## Properties of *Crym*[+] and *Crym*[–] astrocytes

When we compared *Crym*[+] and *Crym*[–] astrocytes (Fig. 5a), we found that several astrocyte markers were equivalently expressed (Fig. 5b), implying that basic astrocytic functions are likely to be similar. However, in line with physiological studies (Fig. 3), the expression of *Slc6a11* (encoding GAT3) was higher in *Crym*[+] astrocytes (Fig. 5b,c) and was one of the differentially expressed genes that were used for pathway analysis (Fig. 5c). We compared 178 genes that were enriched or depleted in *Crym*[+] astrocytes with the top 200 genes that were shared with other striatal astrocytes, and identified shared and unique pathways for *Crym*[+] astrocytes, including those for neurovascular coupling and insulin growth factor (IGF) signalling (Fig. 5d and Extended Data Fig. 11a). We next performed electrophysiological and imaging studies to document differences between *Crym*[+] astrocytes located in the central striatum and *Crym*[–] astrocytes from the dorsolateral striatum (Extended Data Fig. 12). In accordance with the RNA-seq, which showed that astrocyte markers were expressed equally, we found that the properties of *Crym*[+] and *Crym*[–] astrocytes were similar (Extended Data Fig. 12b–h,l). Consistent with scRNA-seq and functional evaluations in *Crym* KO mice, *Crym*[+] astrocytes in the central striatum exhibited higher levels of GAT3 expression and contributed GABA to the extracellular space, whereas those in the dorsolateral region expressed lower levels of GAT3 and removed GABA (Extended Data Fig. 12i–l). GAT3 is known to remove or to contribute GABA to the extracellular space[41,42,47]. Although there are shared and separable functions, the mechanism that is relevant to *Crym*[+] astrocytes and perseveration is GAT3-mediated GABA homeostasis.

## Astrocyte μ-crystallin interactome

We investigated how astrocytic *Crym* KO affected gene expression in astrocytes[33] and in bulk striatal tissue (Extended Data Fig. 11b–e). *Crym* was the only differentially expressed gene (Extended Data Fig. 11b), consistent with the subtle changes in astrocyte markers and properties in *Crym* KO mice. Genes involved in thyroid hormone signalling[20] were unaltered in astrocyte and bulk RNA-seq (Extended Data Fig. 11c), suggesting that loss of *Crym* alters cellular signalling rather than T3-dependent gene expression. To identify μ-crystallin protein-based

mechanisms, we used proximity-dependent biotinylation[48] (Fig. 5e) to detect μ-crystallin interactors. μ-crystallin–BioID2 and the biotinylated proteins it labelled were expressed in striatal astrocytes (Extended Data Fig. 13a–d), allowing the identification of the 78-protein μ-crystallin interactome, which represents proteins in proximity to μ-crystallin, but not their overall abundance within astrocytes[48] (false discovery rate (FDR) < 0.05; Fig. 5e and Extended Data Fig. 13e–g). By mapping the interactome with that of astrocyte subcompartments[48], we found that 21 proteins were associated with μ-crystallin (Fig. 5f), although the interaction between any protein and μ-crystallin is expected to depend on the totality of their interactions and the binding competition between them. Analysis of the μ-crystallin interactome showed that the pathways associated with μ-crystallin related to nucleic acid binding, intracellular signal transduction and cytoskeletal binding (Fig. 5e and Extended Data Fig. 14). μ-crystallin and its interactors were enriched within the cytosol of astrocytes and near the plasma membrane (Extended Data Fig. 13h). Two μ-crystallin interactors were validated using the proximity ligation assay[48] (PLA) (Fig. 5g–i and Extended Data Fig. 13i,j). Recalling lower expression of *Crym*, around 30% of the μ-crystallin interactors were lower in post-mortem RNA-seq data from individuals with HD and OCD, as compared with control individuals (Fig. 5j), suggesting contributions to these disorders. Included in the μ-crystallin interactome were *SYNGAP1*, *DLGAP2* and *DLGAP3*, *BCR*, *CLINT1*, *GSTM1* and *MT-ND5*, which mapped to subcompartments (Fig. 5j) and are associated with diverse psychiatric disorders[49]. The astrocytic μ-crystallin mechanisms that were identified as causal for perseveration (Figs. 1–5) might thus be relevant to other brain diseases.

## Concluding comments

An exciting finding over the past few years has been that astrocytes exhibit diversity according to the region or subregion of the CNS in which they are located[1–3,14–18]. However, a crucial task has been to understand the functions of astrocyte diversity in specific CNS areas, to determine whether regionally enriched astrocytes regulate the neural circuits in which they reside and to ascertain whether such regulation is consequential for the organism or of relevance to disease phenotypes.

Here we have discovered and studied a molecularly defined population of astrocytes that is precisely anatomically allocated, predominant in the central striatum and identified by its expression of μ-crystallin—a protein whose functions in the brain were previously largely unknown[4,20]. At the molecular level, we provide data on how μ-crystallin works within astrocytes, which will be useful for understanding the wider role of μ-crystallin, including in neurons[24,26,27]. Further methods are needed to fully understand the interactions and signalling cascades that are regulated by μ-crystallin[4,20] (for example, ketimine reductase). At the synaptic level, we found that *Crym*[+] astrocytes regulate the neurotransmitter release probability of lOFC–striatum terminals. The extracellular signalling mechanism by which *Crym*[+] astrocytes gate such phenotypes is through tonic-GABA-mediated presynaptic modulation of neurotransmitter release from lOFC terminals arriving at MSNs in the central striatum. We found that *Crym*[+] astrocytes within the central striatum contribute GABA to the extracellular space, whereas *Crym*[–] astrocytes in the dorsolateral striatum remove it. At the circuit level, after reducing the levels of μ-crystallin locally within striatal astrocytes, we identified the lOFC–striatum projection by which *Crym*[+] astrocytes regulate perseveration.

Perseveration represents the inappropriate continuation or repetition of a response or activity and is associated with psychiatric and neurological disorders such as Tourette's syndrome, autism, OCD and HD[9–12]. In the case of HD, downregulation of *CRYM* increases with the severity of disease in human post-mortem tissue[6] and in striatal tissue in a mouse model of HD[25]. Similar to our findings, *CRYM* was expressed in greater numbers of astrocytes than neurons in human striatal tissue[17], and it decreased in both cell types in human HD samples[7] and

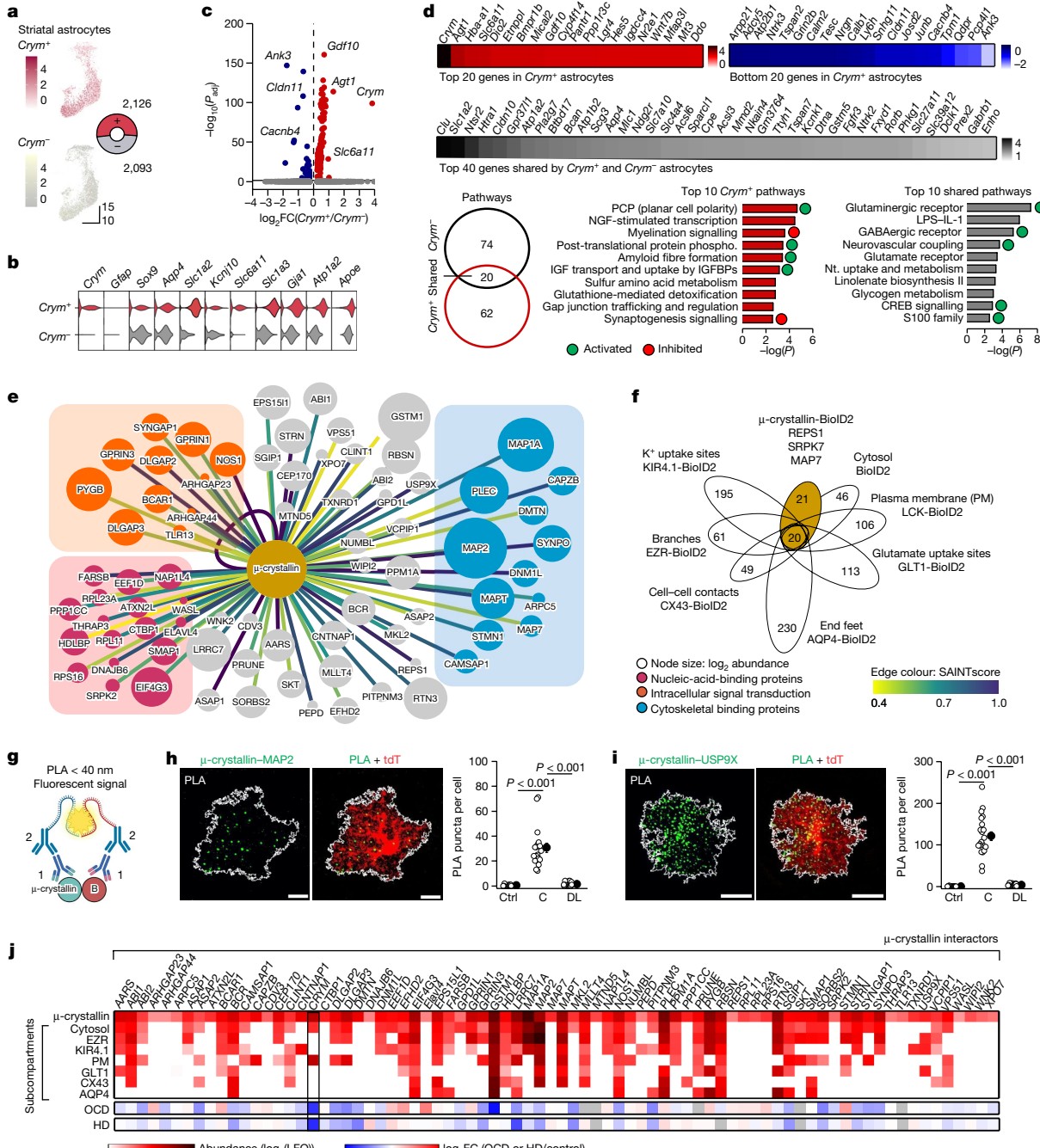

**Fig. 5 | Properties and mechanisms of *Crym*⁺ and *Crym*⁻ astrocytes. a**, UMAP of striatal astrocytes segregated by *Crym* expression. **b**, Violin plot of 10 astrocyte markers in *Crym*⁺ and *Crym*⁻ astrocytes. **c**, Volcano plot of differentially expressed genes between *Crym*⁺ and *Crym*⁻ astrocytes. **d**, The top 20 most (FDR < 0.05) enriched genes and the 20 most depleted 20 genes in *Crym*⁺ astrocytes (scale bar, log₂FC). Top 40 shared genes (of 2,520) in grey (scale bar, log₂FC). Ingenuity pathway analysis (IPA) based top ten shared pathways for *Crym*⁺ and *Crym*⁻ astrocytes and the top ten unique pathways for *Crym*⁺ astrocytes. NGF, nerve growth factor. IGF, insulin like growth factor. IGFBP, insulin like growth factor binding protein. Nt, neurotransmitter. LPS, lipopolysaccharide. **e**, Interaction map of 78 μ-crystallin interacting proteins. Node sizes represent enrichment compared to GFP. Edge colours represent the SAINT protein–protein interaction probability score. All proteins had a SAINT Bonferroni-corrected false discovery

rate (BFDR) less than 0.05. **f**, Clustergram of common and unique proteins (78 proteins) detected in *Crym*-BioID2 relative to astrocyte-specific subcompartments. Proteins represent those that were significant after normalization (log₂FC > 1 and FDR < 0.05 versus GFP controls). **g**, Schematic of the PLA. **h**, Images of PLA puncta for μ-crystallin and MAP2 in tdTomato (tdT)-positive astrocytes. Graph: puncta per tdT⁺ astrocyte in control experiments (ctrl), central (C) and dorsolateral (DL) striatum. Scale bars, 15 μm. **i**, As in **h**, but for and μ-crystallin and USP9X (*n* = 18 and 21 tdTomato⁺ astrocytes from 4 mice per group, one-way ANOVA followed by Tukey's post-hoc test). Scale bars, 15 μm. **j**, Abundance of μ-crystallin interactors in different astrocyte subcompartments from **f**. The bottom heat map shows log₂FC of the μ-crystallin interactors from human OCD[5] and HD[6] post-mortem RNA-seq. Average data shown as mean ± s.e.m. and all statistics reported in Supplementary Table 5.

in mouse models of HD[7,24]. In these regards, our mechanistic findings show that the neurotransmitter release probability of lOFC terminals is regulated by striatal *CRYM*⁺ astrocytes in a manner that is causal for

the perseveration phenotypes that accompany HD. We also show that therapeutic strategies[8] that reduce the release of neurotransmitters from lOFC terminals projecting into the central striatum are likely

to be beneficial for ameliorating perseveration phenotypes that can accompany brain disorders.

Our findings indicate that brain region-enriched astrocytes represent therapeutic targets for diverse disorders that affect specific circuits and brain nuclei. More broadly, for understanding multicellular interactions, our data show that astrocyte diversity has important and behaviourally consequential biological functions in the brain.

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

# Methods

## Mouse models

All animal experiments were conducted in accordance with the National Institutes of Health (NIH) Guide for the Care and Use of Laboratory Animals and were approved by the Chancellor's Animal Research Committee at the University of California, Los Angeles (UCLA). All mice were housed with food and water available ad libitum in a 12-hour light–dark environment at temperatures of 20–22 °C with 40–60% humidity. All mice were healthy with no obvious behavioural phenotype, were not involved in previous studies and were euthanized during the light cycle. Data for most experiments were collected from adult mice aged 9–15 weeks old, but to characterize μ-crystallin expression during development and ageing, mice were used between P0 and 22 months old. For experiments, both male and female mice were used. C57Bl/6NTac mice were maintained as an in-house breeding colony or purchased from Taconic Biosciences. CAG-Cas9 transgenic mice (B6J.129(Cg)-Gt(ROSA)26Sortm1.1(CAG-cas9*,-EGFP)Fezh/J, JAX stock 026179) and SAPAP3$^{-/-}$ mice (B6.129-Dlgap3tm1Gfng/J, JAX stock 008733) were purchased from the Jackson Laboratory and maintained as breeding colonies at UCLA. SAPAP3$^{-/-}$ mice were used at six months of age. Tg(Crym-EGFP) GF82Gsat (strain 012003-UCD) reporter mice were obtained from MMRRC and maintained at UCLA.

## sgRNA design and molecular cloning

To design sgRNAs for CRISPR–Cas9 knockout with minimal off-target effects, six target sequences (crRNAs) for the *Crym* gene were designed using the web tool CHOPCHOP (https://chopchop.cbu.uib.no/). Control target sequences for EGFP were designed using the same method. To select the three most efficient crRNA sequences for *Crym* and the most efficient crRNA sequence for EGFP, in vitro knockout efficiency was assessed using the Guide-it Complete sgRNA Screening System (Takara Bio 632636) by following the manufacturer's instructions. The genomic sequence of *Crym* was obtained from the NCBI GenBank database (gene ID: 12971) and the genomic template was obtained by PCR amplification using the following primers: forward 5′-AGGTGGAACCAGAAAGTCCTCT-3′ and reverse 5′- GCACTTGGTGTATCTGAGCGTG-3′. An original vector containing a U6 promotor followed by two Bbs1 restrictions sites and by the tracrRNA sequence for SpCas9 (pZac2.1-U6-Bbs1-Bbs 1-tracrRNA-GfaABC$_1$D-mCherry-SV40) was created. To insert the three crRNAs for *Crym*, each sequence was separately inserted in chaperone vectors to form three constructs with an sgRNA controlled by the U6 promotor. The three U6-sgRNAs were then combined in the same vector using a BglII restriction site and the In-Fusion cloning kit (Takara Bio 638943). The three crRNA sequences for *Crym* were: 5′-AAGTTAGTCACCTTCTATGA-3′, 5′-GCACCGATGCCTGATGGGAG-3′ and 5′-GGCAGGCGGCGAGATGAAGC-3′. The crRNA sequence for EGFP was 5′-CACCGGGGCGAGGAGCTGTTCACCGGTT-3′. The *U6* sgRNA plasmids for *Crym* and the EGFP control have been deposited at Addgene in the Khakh laboratory repository (Addgene 200067 and 200068). The fully sequenced plasmids were sent to the UPenn Vector Core or Vigene Biosciences to generate AAV2/5 serotype for each construct, yielding a concentration higher than $1.0 \times 10^{13}$ genome copies per ml (gc ml$^{-1}$). The cloning and sequencing strategies were designed with SnapGene software (v.7.0.3, Insightful Science).

## Stereotaxic microinjections of AAVs

All surgical procedures were conducted under general anaesthesia using continuous isoflurane (induction at 5%, maintenance at 1–2% v/v). The depth of anaesthesia was monitored continuously and adjusted when necessary. After the induction of anaesthesia, the mice were fitted into a stereotaxic frame with their heads secured by blunt ear bars and their noses placed into a veterinary grade anaesthesia and ventilation system (VetEquip). Mice were administered with 0.1 mg per kg of buprenorphine (Buprenex, 0.1 mg ml$^{-1}$) subcutaneously before surgery. The surgical incision site was then cleaned three times with 10% povidone iodine and 70% ethanol (v/v). Skin incisions were made, followed by craniotomies of 2–3 mm in diameter above the left frontal or parietal cortex using a small steel burr (Fine Science Tools) powered by a high-speed drill (K.1070, Foredom). Saline (0.9%) was applied onto the skull to reduce heating caused by drilling. Bilateral viral injections were performed by using a stereotaxic apparatus (David Kopf Instruments) to guide the placement of beveled glass pipettes (1B100-4, World Precision Instruments). For the central striatum, the coordinates were 0.8 mm anterior to bregma, 1.85 mm lateral to midline and 2.9 mm from the pial surface. For the lOFC, the coordinates were 2.8 mm anterior to bregma, 1.5 mm lateral to midline and 1.6 mm from the pial surface. For M1, the coordinates were 1.8 mm anterior to bregma, 1.8 mm lateral to midline and 0.6 mm from the pial surface. AAVs were injected by using a syringe pump (Pump11 PicoPlus Elite, Harvard Apparatus). After AAV microinjections, glass pipettes were left in place for at least 10 min before being slowly withdrawn. Surgical wounds were closed with external 5-0 nylon sutures. After surgery, mice were allowed to recover overnight in cages placed partially on a low-voltage heating pad. Buprenorphine was administered twice a day for up to two days after surgery. In addition, trimethoprim sulfamethoxazole was provided in food to the mice for one week to prevent infection.

AAV-injected mice were used for experiments three weeks after surgery. The viruses used were: 0.5 μl or 0.3 μl of AAV2/5 U6-sgRNA-Crym(×3)-GfaABC$_1$D-mCherry virus ($3 \times 10^{13}$ gc ml$^{-1}$) (Addgene 200067), 0.5 μl or 0.3 μl of AAV2/5 U6-sgRNA-GFP-GfaABC$_1$D-mCherry virus ($3.8 \times 10^{13}$ gc ml$^{-1}$) (Addgene 200068), 0.3 μl of AAV9-CaMK11a-hChR2(H134R)-mCherry ($2.1 \times 10^{13}$ gc ml$^{-1}$) (Addgene 26975), 0.3 μl of AAV1-hSyn-Chronos-GFP ($1.4 \times 10^{13}$ gc ml$^{-1}$) (Addgene 59170), 0.3 μl of retrograde AAV-hSyn-hM4D(Gi)-mCherry ($2.4 \times 10^{13}$ gc ml$^{-1}$) (Addgene 50475), 0.3 μl of retrograde AAV-hSyn-mCherry ($2.4 \times 10^{13}$ gc ml$^{-1}$) (Addgene 114472), 0.3 μl AAV2/5-GfaABC$_1$D-Rpl22-HA ($2.1 \times 10^{13}$ gc ml$^{-1}$) (Addgene 111811), 0.5 μl AAV2/5 GfaABC$_1$D-Crym-EGFP ($2.5 \times 10^{13}$ gc ml$^{-1}$) (Addgene 200080), 0.5 μl AAV2/5 GfaABC$_1$D-Crym-BioID2-HA ($3.1 \times 10^{13}$ gc ml$^{-1}$) (Addgene 200070), 0.5 μl AAV2/5 GfaABC$_1$D-jGCaMP8m (Addgene 213010), 0.5 μl AAV2/5 GfaABC$_1$D-tdTomato (Addgene 44332). All AAVs are listed in Supplementary Table 1.

## IHC

**Frozen sections.** For transcardial perfusion, mice were euthanized with isoflurane and perfused with 0.1 M phosphate-buffered saline (PBS) followed by 10% buffered formalin (Fisher SF100-20). Heparin (50 units) was injected into the heart to prevent blood clotting. After gentle removal from the skull, the brain was post-fixed in 10% buffered formalin overnight at 4 °C. For the characterization of striatal μ-crystallin expression during development, P0 and P7 brains were directly post-fixed in 10% buffered formalin without transcardial perfusion. The tissue was cryoprotected in 30% sucrose PBS solution for at least 48 h at 4 °C until use. Forty-micrometre coronal sections were prepared using a cryostat microtome (Leica) at −20 °C and processed for immunohistochemistry. Sections were washed three times in 0.1 M PBS for 10 min each, and then incubated in a blocking solution containing 10% NGS in 0.1 M PBS with 0.2% Triton-X 100 for one hour at room temperature with agitation. Sections were then incubated with agitation in primary antibodies diluted in blocking solution overnight at 4 °C. The following primary antibodies were used: mouse anti-μ-crystallin (1:250, Santa Cruz, sc-376687), chicken anti-GFP (1:1,000; Abcam ab13970), mouse anti-NeuN (1:500; Millipore MAB377), guinea pig anti-NeuN (1:500; Synaptic Systems, 266004), rabbit anti-DARPP-32 (1:1,000, Abcam, ab40801), rabbit anti-S100β (1:1,000; Abcam ab41548), rabbit anti-cFOS (1:1,000; Synaptic Systems 226008), rabbit anti-RFP (1:1,000; Rockland 600–401-379), rabbit anti-mCherry (1:1,000; Abcam, ab167453), rabbit anti-opioid receptor, μ, pain (MOR1) (1:200, Millipore Sigma, AB5511), chicken anti-calbindin D-28K (1:200, Novus Biologicals, NBP2-50028), chicken

anti-GFAP (1:1,000; Abcam, ab4674) and rabbit anti-HA tag (1:1,000; Abcam, ab9110), guinea pig anti-RFP (1:1,000, Synaptic Systems, 390004), rabbit anti-SOX9 (1:500, EMD Millipore, AB5535), rabbit anti-OLIG2 (1:500, EMD Millipore AB9610), rabbit anti-KIR4.1 (1:500, Alomone Labs, APC-035), rabbit anti-ATP1a2 (1:300, Proteintech, 16836-1-AP), rabbit anti-GLT1 (1:500, Synaptic Systems, 250203), mouse anti-GABA (1:500, Abcam, ab86163), rabbit anti-MAOB (1:500, Thermo Fisher Scientific PA5-28338), rabbit anti-GAT3 (1:250, gift from the N. Brecha laboratory at UCLA), rabbit anti-CAPZB (1:250, Thermo Fisher Scientific PA5-83196). The next day, the sections were washed three times in 0.1 M PBS for 10 min each before incubation at room temperature for two hours with secondary antibodies diluted in 0.1 M PBS. Alexa-conjugated (Molecular Probes) secondary antibodies were used at a 1:1,000 dilution (Alexa Fluor 405 goat anti-mouse (A31553), Alexa Fluor 488 goat anti-chicken (A11039), Alexa Fluor 488 goat anti-rabbit (A11008), Alexa Fluor 488 goat anti-mouse (A11010), Alexa Fluor 546 goat anti-mouse (A11030), Alexa Fluor 546 goat anti-rabbit (A11001), Alexa Fluor 647 goat anti-rabbit (A21244), streptavidin and Alexa Fluor 488 conjugate (S11223)). The sections were rinsed three times in 0.1 M PBS for 10 min each before being mounted on microscope slides and sealed in fluoromount-G. Fluorescence images were taken using UPlanSApo 10× 0.4 NA, UPlanFLN 40× 1.30 NA oil immersion or PlanApo N 60× 1.45 NA oil immersion objective lenses on a confocal laser-scanning microscope (Fluoview FV3000; Olympus). Laser settings were kept the same within each experiment. Images represent maximum intensity projections of optical sections with a step size of 1 μm or 5 μm for the entire brain or striatum images. Images were processed with Image J (NIH v.2.1). Cell counting was done on maximum intensity projections using the Cell Counter plug-in; only cells with soma completely within the region of interest (ROI) were counted.

**Acute sections.** Acute sections were used for biocytin staining to assess MSN morphology, spine density and spine head width. Fresh brain slices (300 μm) were placed into 10% buffered formalin overnight at 4 °C and processed as for IHC. Sections were washed three times in 0.1 M PBS with 2% Triton-X 100 for 5 min each, and then incubated in a blocking solution containing 10% NGS in 0.1 M PBS with 1% Triton-X 100 for one hour at room temperature with agitation. Sections were then incubated at room temperature with streptavidin-conjugated Alexa 647 (1:250, S11223) diluted in 0.1 M PBS with 0.4% Triton-X 100 for 3 h. The sections were rinsed three times in 0.1 M PBS for 10 min each before being mounted on microscope slides in fluoromount-G. Images were obtained in the same way as for the IHC for frozen sections, except with a step size of 0.33 μm. For the quantification of spine density, we analysed only spines on dendritic shafts that were parallel to the imaging plane to minimize the possibility of rotational artefacts. Spine density was calculated by dividing the number of spines by the length of the dendritic segment. For quantification of spine head width, a line ROI across the maximum diameter of the spine was made and a profile that has a single peak was obtained. The MSN morphology was determined using a Sholl analysis plug-in with ImageJ (NIH v.2.1).

**TUNEL assay.** To assess apoptosis, a TUNEL assay was performed on frozen sections in control and *Crym* KO mice using the TUNEL Andy Fluor 647 Apoptosis Detection Kit (ABP Biosciences, A052). In brief, DNA strand breaks with 3′-hydroxyl ends are labelled with biotin-11-dUTP in the presence of terminal deoxynucleotidyl transferase (TdT). Once incorporated into the DNA, biotin is detected using HRP- or fluorophore-labelled streptavidin. The experiment was performed following the manufacturer's instructions.

### Retrograde tracing and afferent projections

For retrograde tracing experiments, 300 nl of cholera toxin subunit B, Alexa Fluor 647-conjugated (CTB-647) (Thermo Fisher Scientific C34778) was injected in the central striatum of CAG-Cas9-GFP mice.

Ten days after injection, frozen sections were prepared as described above. The number of CTB-647$^+$ cell bodies per mm$^2$ were manually counted in regions of the prefrontal cortex and motor cortex delineated by the Paxinos Brain Atlas. To visualize the projections from the lOFC and M1, 0.3 μl of AAV9-CaMKIIa-hChR2(H134R)-mCherry and 0.3 μl of AAV1-Syn-Chronos-GFP were injected in the lOFC and M1 respectively. Four weeks after injection, mCherry and GFP fluorescence in the striatum were delineated with a threshold method and the intensity of μ-crystallin was quantified in each ROI.

### Astrocyte morphology

Astrocyte morphology was assessed using Lucifer Yellow iontophoresis. In brief, control or *Crym* KO mice were transcardially perfused with 10 ml of 35 °C Ringer's solution (135 mM NaCl, 14.7 mM NaHCO₃, 5 mM KCl, 1.25 mM Na₂HPO₄, 2 mM CaCl₂, 1 mM MgCl₂ and 11 mM D-glucose, bubbled with 95% O₂ and 5% CO₂) with 0.02% lidocaine followed by 50 ml of 10% buffered formalin (Thermo Fisher Scientific SF100-20). Brains were lightly post-fixed at room temperature for 1.5 h and then washed three times in ice-cold 0.1 M PBS for 10 min. Coronal sections (100 μm) were cut using a Pelco Vibrotome 3000 and then placed in ice-cold PBS for the duration of the experiment. Ten milligrams of Lucifer yellow CH di-Lithium salt (Sigma L0259) was dissolved in 1 ml 5 mM KCl solution and filtered before use. Sharp (around 200 MOhm) glass electrodes were pulled from borosilicate glass capillaries with filament (outer diameter 1.0 mm; inner diameter 0.58 mm). Electrodes were gravity filled with Lucifer yellow solution. Sections were transferred to a PBS solution at room temperature for filling. Astrocytes were identified using mCherry expression and then impaled with the sharp electrode. Lucifer yellow was injected into the cell by applying around 1 V for 10 min. After the astrocyte was filled, the electrode was removed completely. The sections were fixed with 4% paraformaldehyde and the filled astrocyte was imaged with UPlanFLN 40× 1.30 NA oil immersion or PlanApo N 60× 1.45 NA oil immersion objective lenses on a confocal laser-scanning microscope (Fluoview FV3000; Olympus) at a digital zoom of two to three and a 0.3-μm confocal z-step. Territory area and soma area were generated using threshold reconstruction with Image J (NIH v.2.1).

### Dual in situ hybridization with RNAscope and IHC

Fixed-frozen tissue was processed as described above. Serial coronal sections (14 μm) containing striatum were prepared using a cryostat microtome (Leica) at −20 °C and mounted immediately onto glass slides. In situ hybridization was performed using Multiplex RNAscope (ACDBio 320851). Sections were washed for at least 15 min with 0.1 M PBS, and then incubated in 1× Target Retrieval Reagents (ACDBio 322000) for 5 min at 95–100 °C. After washing with ddH₂O twice for 1 min each, they were dehydrated with 100% ethanol for 2 min and dried at room temperature. Then, the sections were incubated with Protease Pretreat solution (ACDBio 322340) for 30 min at 40 °C. The slides were washed with ddH₂O twice for 1 min each and then incubated with probe(s) for 2 h at 40 °C. The following probes were used: Mm-Crym-C3 (ACDBio 466131-C3), Mm-Drd1-C3 (ACDBio 461901-C3) and Mm-Drd2-C3 (ACDBio 406501-C3). The sections were incubated in AMP 1-FL for 30 min, AMP2-FL for 15 min, AMP3-FL for 30 min and AMP4-FL for 15 min at 40 °C with washing in 1× wash buffer (ACDBio 310091) twice for 2 min each before the first incubation and in between incubations. All the incubations at 40 °C were performed in the HybEZ Hybridization System (ACDBio 310010). Slices were washed in 0.1 M PBS three times for 10 min each, followed by IHC that was performed as described above except with antibody dilutions. The following primary antibodies were used: mouse anti-μ-crystallin (1:100, Santa Cruz, sc-376687), rabbit anti-DARPP-32 (1:500, Abcam, ab40801) and rabbit anti-S100β (1:500; Abcam ab41548). Images were obtained in the same way as for the IHC described above, and were processed with Image J (NIH v.2.1). Astrocyte area was demarcated by drawing

an 80-µm-diameter circle around the soma (the average size of striatal astrocytes based on our previous experiments). Then, the percentage of D1 and D2 positive neurons was quantified within this area.

## Pharmacological treatments

To reduce stress, mice were handled and habituated three days before beginning the treatments. For fluoxetine experiments, six-month-old SAPAP3$^{-/-}$ mice were treated with an intraperitoneal (i.p.) injection of fluoxetine (Tocris 0927) at 10 mg per kg (dissolved in 0.9% sodium chloride) for seven consecutive days. Three weeks after AAV microinjection, striatal *Crym* KO mice were treated with an i.p. injection of fluoxetine at 10 mg per kg for 21 consecutive days. For Gi activation, mice were treated with a single i.p. injection of deschloroclozapine (DCZ, Tocris 7193) at 1 µg kg$^{-1}$ for one day (grooming test), two days (open-field test), three days (marble-burying test), seven days (novel object recognition test) or eight days (IHC). For both experiments, untreated mice received the same amount of 0.9% sodium chloride (vehicle). Behavioural tests or euthanasia were performed one to two hours after treatments.

## Behavioural tests

Behavioural tests were performed during the light cycle between 09:00 and 18:00, three to five weeks after AAV injection. Both male and female mice were used in behavioural tests. The mice were randomly allocated to a group as they became available and of age in the breeding colony. All of the experimental mice were transferred to the behaviour testing room at least 30 min before the tests to allow them to acclimatize to the environment and to reduce stress. The temperature and humidity of the experimental rooms were kept at $23 \pm 2\,°C$ and $55 \pm 5\%$, respectively. Background noise ($65 \pm 2$ dB) was generated by a white noise generator (San Diego Instruments). The brightness of the experimental room was kept at less than 20 lux.

**Open-field test.** The open-field chamber consisted of a square arena ($28.5 \times 30$ cm) enclosed by walls made of translucent polyethylene (15 cm tall). The locomotor activity of mice was recorded for 10 or 30 min using a camera located above the open-field chamber. The open-field behaviours were analysed with an automated video tracking software ANY-maze (v.6.3, Stoelting).

**Rotarod test.** Mice were held by the tails and placed on the rod (3 cm diameter) of a single lane rotarod apparatus (ENV-577 M, Med Associates), facing away from the direction of rotation. The rotarod was set with a start speed of 4 rpm. Acceleration started 10 s later and was set to 20 rpm with a maximum speed 40 rpm per min. Each mouse received five trials 30 min apart for five consecutive days and the latency to fall was recorded for each trial. The average latency to fall was used as a measurement for motor coordination.

**Footprint test.** A 1-m-long runway (8 cm wide) was lined with paper. Each mouse had its hind paws painted with non-toxic ink and was placed at an open end of the runway. The mice were allowed to walk to the other end with a darkened box. For the gait analysis, stride length and width were measured and averaged for both left and right hindlimbs over five steps.

**Self-grooming behaviour.** Mice were placed individually into plastic cylinders (15 cm in diameter and 35 cm tall), and allowed to habituate for 20 min. Self-grooming behaviour was recorded for 10 min. A timer was used to assess the cumulative time spent in self-grooming behaviour, which included paw licking; unilateral and bilateral strokes around the nose, mouth and face; paw movement over the head and behind the ears; body fur licking; body scratching with hind paws; tail licking; and genital cleaning. The number of self-grooming bouts and rearing bouts was also counted. Separate grooming bouts were considered when the

pause was more than 5 s or if behaviours other than self-grooming occurred. The self-grooming microstructure was not assessed.

**Spray test.** A standard spray bottle was filled with distilled water and the nozzle was adjusted to the 'misting' mode. Mice were held by the tails on the bench and sprayed four times from 30 cm away to be adequately covered with mist. Mice were placed individually into the plastic cylinders and grooming behaviour was recorded for 10 min after the spray and then analysed as described in the preceding section.

**Marble-burying and digging test.** Fresh, unscented soft wood chip bedding was added to polycarbonate cages ($21 \times 43 \times 20.5$ cm) to a depth of 5 cm. Fifteen sanitized glass marbles were gently placed on the surface of the bedding in five rows of three marbles. Mice were allowed to remain in the cage undisturbed for 30 min. A marble was scored as buried if two-thirds of its surface area was covered by bedding. For the digging test, the same experiment without the marbles was performed. The latency to start digging, the total digging duration and the duration of digging bouts were also counted.

**Novel object recognition.** On day 1 and day 2, mice were placed in an empty open chamber ($28.5 \times 30$ cm) for 10 min for habituation. On day 3 (training day), mice were placed in the same open chamber containing two identical objects evenly spaced apart; the trial was video recorded for 10 min. For the novel object recognition test, at day 4 (testing day), 24 h after training, mice were placed in the same open chamber, with one of the two objects replaced with a novel object. The trial was video recorded for 10 min. Time exploring around the objects was manually measured. The recognition index was calculated as follows: (time exploring the novel object – time exploring the familiar object)/(time exploring both objects) – 50.

**Lickometer test.** Mice were placed in a lickometer system (Stoelting) to measure licking behaviour ($20 \times 20$ cm). In brief, an electrical signal was generated from the tongue touching the sipper tube, with each lick recorded as an event using the ANY-maze software (v.6.3, Stoelting). Mice were placed in the chamber for 30 min during four consecutive days and placed back to their home cage at the end of each session. Before the fourth day, mice were water deprived for 12–16 h before the session. The total number of licks, the total duration of drinking and the latency to start drinking were measured by the software.

**Elevated plus maze.** The elevated plus maze consisted of arms that were $30 \times 7$ cm with closed-arm walls with a height of 20 cm. The maze was elevated 65 cm above floor level and was placed in the centre of the room away from other stimuli. Mice were placed in the centre of the maze facing an open arm. Mice were recorded for 10 min using a camera (Logitech) located above the maze. ANY-maze video analysis software was used to quantify the time spent in open arms and time spent in close arms.

**Food consumption.** Three weeks after AAV injections, 200 g of food was placed in the home cage for 72 h. The remaining food was weighed and divided by the number of days and the number of mice to calculate the consumption in gram per day per mouse.

**Weight.** Mice were weighed the day before the AAVs injection and every week for four weeks after injection.

## Brain-slice preparation and electrophysiological recordings in the striatal slices

Coronal or sagittal striatal slices were prepared from 6–12-week-old *Crym* KO mice, control or *Crym*-GFP mice. In brief, mice were deeply anaesthetized with isoflurane and decapitated with sharp shears. The brains were placed and sliced in ice-cold modified artificial CSF (aCSF)

containing the following: 194 mM sucrose, 30 mM NaCl, 4.5 mM KCl, 1 mM $MgCl_2$, 26 mM $NaHCO_3$, 1.2 mM $NaH_2PO_4$ and 10 mM D-glucose, and saturated with 95% $O_2$ and 5% $CO_2$. A vibratome (DSK Microslicer; Ted Pella) was used to cut 300-μm brain sections. The slices were allowed to equilibrate for 30 min at 33 °C in normal aCSF containing: 124 mM NaCl, 4.5 mM KCl, 2 mM $CaCl_2$, 1 mM $MgCl_2$, 26 mM $NaHCO_3$, 1.2 mM $NaH_2PO_4$ and 10 mM D-glucose, continuously bubbled with 95% $O_2$ and 5% $CO_2$. Slices were then stored at 21–23 °C in the same buffer until use. All slices were used within 6 h of slicing. The brain-slice recordings were performed at room temperature (21–23 °C). Whole-cell patch-clamp recordings were made from astrocytes or MSNs in the central striatum using patch pipettes with a typical resistance of 4–6 MΩ. MSNs were morphologically and electrophysiologically identified. Astrocytes were identified by mCherry or EGFP expression. The intracellular solution for MSN membrane properties, EPSCs, evoked EPSCs and astrocyte recordings comprised the following: 135 mM potassium gluconate, 3 mM KCl, 10 mM HEPES, 1 mM EGTA, 0.3 mM Na-ATP, 4 mM Mg-ATP, 0.1 mM $CaCl_2$ and 8 mM $Na_2$-phosphocreatine, with the pH adjusted to 7.3. The intracellular solution for IPSC recordings comprised the following: 138 mM KCl, 10 mM HEPES, 1 mM EGTA, 0.3 mM Na-ATP, 4 mM Mg-ATP, 0.1 mM $CaCl_2$, 8 mM $Na_2$-phosphocreatine and 3 mM QX314, with the pH adjusted to 7.3. To evoke EPSCs specifically from the M1 or lOFC projections, Channelrhodopsin 2 was injected bilaterally in one of these two areas and afferents in the central striatum were stimulated with an LED light source (Lambda TLED+, Sutter Instrument). To assess input–output function, test stimuli were applied at increasing intensities ranging from 0 to 3 mW. Stimulation intensities were set to evoke responses at 50–60% maximal amplitude to induce paired-pulse responses. Paired pulses were delivered at five different interpulse intervals: 40, 60, 80, 100 and 160 ms apart. To isolate mEPSCs, MSNs were voltage-clamped at −70 mV and pre-incubated with 0.3 μM tetrodotoxin (TTX) before recording. To isolate sIPSCs, MSNs were voltage-clamped at −60 mV and pre-incubated in the presence of 10 μM 6-Cyano-7-nitroquinoxaline-2,3-dione (CNQX) for 10 min before recording. mIPSCs were recorded after incubation with 10 μM CNQX and 0.3 μM TTX for 10 min. To access tonic GABA currents, 25 μM bicuculline was bath-applied in the presence of 10 μM CNQX, 0.3 μM TTX. In some cases, 1 mg ml$^{-1}$ biocytin (Tocris, 3349) was added to the intracellular solution to subsequently visualize patched neuron. All recordings were performed at room temperature, using pCLAMP11.2 (Axon Instruments, Molecular Devices) and a MultiClamp 700B amplifier (Axon Instruments, Molecular Devices). Cells with Ra that exceeded 25 MΩ were excluded from analysis. Analysis was performed using ClampFit 10.7 software.

## Calcium imaging of astrocytes in brain slices

Slice preparation was performed as above. Cells for all of the experiments were imaged using a confocal microscope (Fluoview 1000; Olympus) with a 40× water-immersion objective lens with 0.8 NA and a digital zoom of two to three. We used the 488-nm line of an Argon laser, with the intensity adjusted to 5–10% of the maximum output of 10 mW. The emitted light pathway consisted of an emission high-pass filter (505–525 nm) before the photomultiplier tube. Astrocytes were typically around 25 μm below the slice surface and were scanned at one frame per second for imaging sessions. For pharmacological activation of endogenous G-protein-coupled receptors, agonists were dissolved in water. Stock solutions were diluted in aCSF immediately before use. Analyses of time-lapse image series were performed using ImageJ v.2.1 (NIH). The XY drift was corrected using ImageJ; cells with Z drift were excluded from analyses. Time traces of fluorescence intensity were extracted from the ROIs and converted to $\Delta F/F$ values. For microdomains, GECIquant software (v.1.0) was used. Events were identified on the basis of amplitudes that were at least twofold above the baseline noise of the $\Delta F/F$ trace. Extracted calcium signals were analysed using OriginPro 2017 (OriginLab, v.9.4.2).

## Striatal scRNA-seq

scRNA-seq was performed to profile the whole striatum of four adult mice using a protocol that preferentially captures non-neuronal cells[23]. Male mice at 7–8 weeks old were anaesthetized and decapitated. The brain was immediately dissected out and was sectioned on a vibratome (Microslicer DTK-Zero1; Ted Pella) into 400-μm slices in ice-cold aCSF + trehalose (ACSF-T) (124 mM NaCl, 2.5 mM KCl, 1.2 mM $NaH_2PO_4$, 24 mM $NaHCO_3$, 5 mM HEPES, 13 mM glucose, 2 mM $MgSO_4$, 2 mM $CaCl_2$ and 14.6 mM trehalose, pH adjusted to 7.3–7.4) oxygenated with 95% $O_2$/5% $CO_2$. The slices containing the striatum were immediately transferred to an oxygenated recovery solution (93 mM N-methyl-D-glucamine, 2.5 mM KCl, 1.2 mM $NaH_2PO_4$, 30 mM $NaHCO_3$, 20 mM HEPES, 25 mM glucose, 10 mM $MgSO_4$, 0.5 mM $CaCl_2$, 5 mM sodium ascorbate, 2 mM thiourea, 3 mM sodium pyruvate and 45 μM actinomycin D, with a pH of 7.3–7.4) for 15 min on ice. The striatum was dissected out under a dissecting microscope in ice-cold ACSF-T and cut into small pieces (less than 1 mm in all dimensions). Tissue was then transferred to a Petri dish for digestion with ACSF-T containing 1 mg ml$^{-1}$ pronase (Sigma-Aldrich, P6911) and 45 μM actinomycin D (Sigma-Aldrich, A1410) and incubated at 34 °C for 20 min with aerated carbogen. The digested tissue was transferred to ice-cold oxygenated ACSF-T containing 1% fetal bovine serum and 3 μM actinomycin D. The tissue was dissociated sequentially by gentle trituration through glass Pasteur pipettes with polished tip openings of 500 μm, 300 μm and 150 μm diameter. Actinomycin D was added to the recovery solution at 45 μM, the pronase solution at 45 μM and trituration solution at 3 μM to prevent stress-induced transcriptional alterations. To increase the yield of glial cells[23], filters with various pore sizes (70 μm, 40 μm and 20 μm) were tested, and a 20-μm filter gave the highest yield and therefore was chosen. The dissociated cells were filtered through a 20-μm filter and washed with ice-cold ACSF-T. To remove myelin, the cell pellet was resuspended in PBS and processed with a debris removal kit (Miltenyi Biotec, 130-109-398). The cell density was calculated and isolated cells were diluted to 1,000 cells per microlitre and processed with the 10X Genomics platform within 10 min. Single-cell libraries were generated and sequenced on the Illumina NextSeq500 sequencer.

## scRNA-seq analysis

Sequence reads were processed and aligned to the mouse genome (mm10) using CellRanger 3.0. Striatal cells with fewer than 300 genes and genes expressed in more than 3 cells were used for the subsequent analysis in R. Basic processing and visualization of the scRNA-seq data were performed with the Seurat package (v.4.0.5) in R (v.4.0.3). Scrublet was embedded in the Seurat pipeline to remove doublets. Our initial dataset contained 64,836 cells. Data across batches were integrated with canonical correlation analysis (CCA), and annotated with mouse ID and age with a metadata column. The transcript expression was normalized for each cell by the total expression, and multiplied by a scale factor of 10,000, and log-transformed. Next, principal component analysis (PCA) was performed, and the top 30 principal components (PCs) were stored. Clusters were identified with the FindClusters() function by use of the shared nearest neighbour modularity optimization with a clustering resolution set to 0.08. Cell-class clusters were then annotated on the basis of the expression of cell lineage marker genes. Astrocyte cell class was further analysed. Astrocytes were subset, CCA integrated, scaled and normalized; this was followed by PCA analysis, and shared nearest neighbour modularity optimization with a clustering resolution of 0.08 was performed. Six molecular subclusters were identified. Astrocyte cells that were $Crym^+$ were defined if the Crym transcript expression level was greater than 0.25 (2,126 cells), and $Crym^-$ cells were defined if the Crym transcript expression level was less than 0.25 (2,093 cells). MAST (model-based analysis of single-cell transcripts) comparison identified differentially expressed genes (FDR < 0.05) in $Crym^+$ and $Crym^-$ cells with a threshold criteria of 0.1

(that is, 10% of cells). Genes were considered commonly expressed if they were expressed in all astrocyte cells, but were not differentially expressed in *Crym*+ astrocytes. QIAGEN ingenuity pathway analysis (IPA) was performed to identify canonical pathways associated with these genes (made available by QIAGEN Digital Insights).

### RNA-seq experiments using the RiboTag method
CAG-Cas9-GFP mice (6–8 weeks old) were bilaterally injected in the central striatum with astrocytic RiboTag (AAV2/5-GfaABC1D-Rpl22-HA) in conjunction with AAVs expressing sgRNAs for *Crym* (AAV2/5 U6-sgRNA-Crym(×3)-GfaABC1D-mCherry) or for EGFP (AAV2/5 U6-sgRNA-GFP-GfaABC1D-mCherry). RNA was immunoprecipitated from astrocytes or neurons at 10–12 weeks old as previously described[33]. In brief, freshly dissected striatum tissues were collected and individually homogenized in 1 ml homogenization buffer (50 mM Tris pH 7.4, 100 mM KCl, 12 mM MgCl2, 1% NP-40, 1 mM dithiothreitol (DTT), 1× protease inhibitors, 200 U ml$^{-1}$ RNAsin, 100 µg ml$^{-1}$ cyclohexamide and 1 mg ml$^{-1}$ heparin). RNA was extracted from 20% of cleared lysate as Input (200 µl). The remaining lysate (800 µl) was incubated with 5 µl of mouse anti-HA antibody (Biolegend 901514) with rocking for 4 h at 4 °C, followed by the addition of 200 µl of magnetic beads (Pierce 88803) and overnight incubation with rocking at 4 °C. The beads were washed three times in high-salt solution (50 mM Tris pH 7.4, 300 mM KCl, 12 mM MgCl2, 1% NP-40, 1 mM DTT and 100 µg ml$^{-1}$ cyclohexamide). RNA was purified from the immunoprecipitated and corresponding input samples (RNeasy Plus Micro, QIAGEN 74034). RNA concentration and quality were assessed with Agilent 2100 Bioanalyzer. RNA samples with an RNA integrity number greater than seven were processed with the Ribo-Zero Gold kit (Epicentre) to remove ribosomal RNA. Sequencing libraries were prepared using the Illumina TruSeq RNA sample preparation kit following the manufacturer's protocol. After library preparation, amplified double-stranded cDNA was fragmented into 125-bp (Covaris-S2) DNA fragments, which were (200 ng) end-repaired to generate blunt ends with 5′-phosphates and 3′-hydroxyls and adapters ligated. The purified cDNA library products were evaluated using the Agilent Bioanalyzer and diluted to 10 nM for cluster generation in situ on the HiSeq paired-end flow cell using the CBot automated cluster generation system. Samples in each experiment were multiplexed into a single pool to avoid batch effects and 69-bp paired-end sequencing was performed using an Illumina HiSeq 4000 sequencer (Illumina). A yield of between 51 and 108 million reads was obtained per sample. Quality control was performed on base qualities and the nucleotide composition of sequences. Alignment to the *Mus musculus* (mm10) refSeq (refFlat) reference gene annotation was performed using the STAR spliced read aligner (v.2.7.5c) with default parameters. Further quality control was performed after the alignment to examine: the degree of mismatch rate, mapping rate to the whole genome, repeats, chromosomes, key transcriptomic regions (exons, introns, UTRs and genes), insert sizes, AT/GC dropout, transcript coverage and GC bias. Between 75% and 94% (average 87.6%) of the reads mapped uniquely to the mouse genome. Total counts of read fragments aligned to candidate gene regions were derived using the HTSeq program (https://htseq.readthedocs.io/en/latest/overview.html#overview) with mouse mm10 refSeq (refFlat table) as a reference and used as a basis for the quantification of gene expression. Only uniquely mapped reads were used for subsequent analyses. Differential expression analysis was conducted with R Project and the Bioconductor package limmaVoom. Weighted gene co-expression network analysis (WGCNA) was performed using an R package. Modules of genes that highly correlated with HD samples were selected. RNA-seq data have been deposited in the Gene Expression Omnibus (GEO) repository (https://www.ncbi.nlm.nih.gov/geo/).

### Metabolomics experiments
Four to six weeks after AAV injection, mice were decapitated and striata were dissected and flash-frozen. Five to eight milligrams of each tissue sample was extracted using a Folch-like method (water, methanol and chloroform) and a bead-based mechanical tissue disruptor. The polar phase was dried and derivatized for GC−MS analysis as previously described[50]. The results were scaled against calibrated standards and normalized to the frozen weight of the starting material to obtain nmol per mg values.

### Proteomic experiments
**In vivo BioID2 protein biotinylation.** A detailed protocol is available in a previous report[48]. In brief, three weeks after AAV microinjection with *Crym*-BioID2 AAV, mice were treated with a subcutaneous injection of biotin at 24 mg per kg (Millipore Sigma RES1052B-B7) dissolved in sterile 0.1 M PBS once per day for seven consecutive days. The mice were processed 16 h after the last biotin injection.

**Western blot of in vivo BioID2 samples.** Mice were decapitated and the striata were dissected and homogenized with a dounce and pestle in ice-cold RIPA buffer (150 mM NaCl, 50 mM Tris pH 8.0, 1% Triton-X, 0.5% sodium deoxycholate, 0.1% SDS and Halt protease inhibitor (Thermo Fisher Scientific 78429). The homogenate was incubated at 4 °C while spinning for one hour. The homogenate was sonicated and then centrifuged at 4 °C for 10 min at 15,600$g$. The clarified lysate was collected and the protein concentration was measured using the BCA protein assay (Thermo Fisher Scientific). The samples were then mixed with 2× Laemmli solution (BioRad) containing β-mercaptoethanol. The samples were boiled at 95 °C for 10 min before being electrophoretically separated by 10% SDS−PAGE (30 µg protein per lane) and transferred onto a nitrocellulose membrane (0.45 µm). The membrane was incubated with agitation in a solution containing 5% BSA, 0.1% Tween-20 and 0.1 M PBS for 1 h. The membrane was probed with streptavidin−HRP (Sigma RABHRP3) at 1:250 for two hours. The membrane was then treated with the Pierce chemiluminescence solution for 1 min and imaged. The blot was incubated overnight at 4 °C with rabbit anti-β-actin (1:1,000; Abcam ab8227). IR-dye 800CW anti-rabbit (1:10,000; Li-Cor) secondary was used and images were acquired on a Li-Cor odyssey infrared imager. Signal intensities at the expected molecular weight were quantified using Image J (v.2.1). The streptavidin signal levels were normalized to β-actin by dividing the streptavidin signal intensity by the β-actin signal intensity. Thirty micrograms of protein was loaded into each gel well.

**In vivo pull-down of BioID2 biotinylated proteins.** The purification of biotinylated proteins was performed as previously described[48]. Each AAV *Crym*-BioID2 probe and its counterpart AAV *Crym*-GFP control were injected into the striatum of six-week-old C57/Bl6NTac mice. Three weeks after AAV microinjection, biotin (Millipore Sigma RES1052B-B7) was subcutaneously injected at 24 mg per kg for seven consecutive days. All mice were processed 16 h after the last biotin injection. Eight mice were used for each biotinylated protein purification and each purification was performed independently five times. Mice were decapitated and striata were microdissected. Striata from four mice were dounce homogenized with 600 µl of lysis buffer 1 (1 mM EDTA, 150 mM NaCl and 50 mM HEPES pH 7.5, supplemented with Halt protease inhibitor (Thermo Fisher Scientific 78429). Immediately after homogenization, 600 µl of lysis buffer 2 (2% sodium deoxycholate, 2% Triton-X, 0.5% SDS, 1 mM EDTA, 150 mM NaCl and 50 mM HEPES pH 7.5) was added. The lysed samples were sonicated for 5 min at 60% power and then centrifuged at 15,000$g$ for 15 min at 4 °C. The resulting supernatant was then ultracentrifuged at 100,000$g$ for 30 min at 4 °C. SDS was added to the supernatant for a final concentration of 1%. The sample was then boiled at 95 °C for 5 min. Sample was cooled on ice and incubated with 35 µl of equilibrated anti-pyruvate carboxylase (5 µg; Abcam 110314) conjugated agarose beads (Pierce 20398) for four hours at 4 °C while rotating. Subsequently, the sample was centrifuged at 2,000 rpm for 5 min at 4 °C and the supernatant was incubated with 80 µl NeutrAvidin agarose at 4 °C overnight while rotating. The NeutrAvidin beads were

then washed twice with 0.2% SDS, twice with wash buffer (1% Na deoxycholate, 1% Triton-X and 25 mM LiCl), twice with 1 M NaCl and five times with 50 mM ammonium bicarbonate. Proteins bound to the agarose were then eluted in elution buffer (5 mM biotin, 0.1% RapiGest SF surfactant and 50 mM ammonium bicarbonate) at 60 °C for a minimum of 2 h. The final protein concentration was measured by BCA.

**Analysis of biotinylated proteins and bulk mouse tissue using mass spectrometry.** Protein samples were subjected to reduction using 5 mM Tris (2-carboxyethyl) phosphine for 30 min, alkylated by 10 mM iodoacetamide for another 30 min and then digested sequentially with Lys-C and trypsin at a 1:100 protease-to-peptide ratio for 4 and 12 h, respectively. The digestion reaction was terminated by the addition of formic acid to 5% (v/v) with centrifugation. Each sample was then desalted with C18 tips (Thermo Fisher Scientific 87784) and dried in a SpeedVac vacuum concentrator. The peptide pellet was reconstituted in 5% formic acid before analysis by liquid chromatography–tandem mass spectrometry (LC–MS/MS).

Tryptic peptide mixtures were loaded onto a 25 cm long, 75 μm inner diameter fused-silica capillary, packed in-house with bulk 1.9 μM ReproSil-Pur beads with 120-Å pores. Peptides were analysed using a 140-min water–acetonitrile gradient delivered by a Dionex Ultimate 3000 UHPLC (Thermo Fisher Scientific), operated initially at a flow rate of 400 nl per min with 1% buffer B (acetonitrile solution with 3% dimethyl sulfoxide (DMSO) and 0.1% formic acid) and 99% buffer A (water solution with 3% DMSO and 0.1% formic acid). Buffer B was increased to 6% over 5 min, at which time the flow rate was reduced to 200 nl per min. A linear gradient from 6–28% B was applied to the column over the course of 123 min. The linear gradient of buffer B was then further increased to 28–35% for 8 min followed by a rapid ramp up to 85% for column washing. Eluted peptides were ionized using a Nimbus electrospray ionization source (Phoenix S&T) by application of a distal voltage of 2.2 kV.

The spectra were collected using data-dependent acquisition on an Orbitrap Fusion Lumos Tribrid mass spectrometer (Thermo Fisher Scientific) with an MS1 resolution of 120,000, followed by sequential MS2 scans at a resolution of 15,000. Data generated by LC–MS/MS were searched using the Andromeda search engine integrated into the MaxQuant bioinformatic pipelines against the Uniprot *M. musculus* reference proteome (UP000000589 9606) and then filtered using a 'decoy' database-estimated FDR < 1%. Label-free quantification (LFQ) was performed by integrating the total extracted ion chromatogram of peptide precursor ions from the MS1 scan. These LFQ intensity values were used for protein quantification across samples. Statistical analysis of differentially expressed proteins was done using the Bioconductor package ArtMS. To generate a list of proteins with high confidence, all mitochondrial proteins including carboxylases and dehydrogenases were manually filtered because they are artefacts of endogenously biotinylated proteins. Proteins with a log$_2$-transformed fold change (log$_2$FC) > 1 and FDR < 0.05 versus GFP controls were considered putative hits and used for subsequent comparisons between subcompartments and cell types. To account for variations in pull-down efficiency, all proteins and their LFQ values were normalized to pyruvate carboxylase (Uniprot ID Q05920). Downstream analysis was conducted only on proteins with non-zero LFQ values in three or more experimental replicates. Data analysis for whole bulk tissue analyses was performed identically, except that samples were normalized by median intensity. The gene ontology (GO) enrichment analysis for cellular compartments and biological function was performed using the PANTHER overrepresentation test (GO ontology database released 2020-01-01) with FDR < 0.05 with all *M. musculus* genes used as a reference, and with STRING (https://string-db.org/) with a confidence score of 0.5 and with all *M. musculus* genes used as a reference. The GO pathway analysis for the *Crym*-BioID2 interactome was done with Enrichr (https://maayan-lab.cloud/Enrichr/).

**Protein networks and protein–protein interaction analysis.** Network figures were created using Cytoscape (v.3.8) with nodes corresponding to the gene name for proteins identified in the proteomic analysis. A list of protein–protein interactions from published datasets was assembled using STRING. In all networks, node size is proportional to the fold enrichment over GFP control. To identify interactors of μ-crystallin protein, Significance Analysis of INTeractome (SAINTexpress) was used with an FDR cut-off of 0.05. The Bioconductor artMS package was used to reformat the MaxQuant results to make them compatible with SAINTexpress.

## PLA

The PLA detects native interacting proteins within about 40 nm of each other. Fixed-frozen tissue was processed as described in previous sections. Serial coronal sections (20 μm) containing striatum sparsely labelled with astrocyte-specific AAV2/5 GfaABC$_1$D-tdTomato were prepared using a cryostat microtome (Leica) at −20 °C and mounted immediately onto glass slides. PLAs were performed using the Sigma-Aldrich Duolink PLA fluorescence protocol (Sigma-Aldrich DUO92101 and DUO92013). Sections were baked for 30 min at 60 °C. Sections were washed for at least 15 min in 0.1 M PBS. After washing, sections were incubated in 1× citrate pH 6.0 antigen retrieval buffer (Sigma, C999) for 10 min at 90 °C. After washing three times in 0.2% Triton-X in PBS (PBS-T), the sections were blocked for 45 min at room temperature with 5% donkey serum (Sigma D9663) in PBS-T. Sections were then incubated with the following primary antibodies overnight at 4 °C: rabbit anti-USP9X (1:250, Proteintech 55054-1-AP); rabbit anti-MAPT (1:250, Thermo Fisher Scientific PA-10005); mouse anti-μ-crystallin (1:125, Santa Cruz sc-376687); and guinea pig anti-RFP (1:500; Synaptic Systems 390004). Sections were then incubated with PLA probe cocktail containing the anti-rabbit PLUS primer probe (DUO92002) and the anti-mouse MINUS primer probe (DUO92004) for 1 h at 37 °C. The sections were washed twice in 1× wash buffer A (DUO82049). Sections were then incubated with ligation solution containing ligase for 30 min at 37 °C. Sections were once again washed twice with 1× wash buffer A and then incubated with amplification solution containing DNA polymerase for at least 3 h at 37 °C. Sections were then washed twice in 1× wash buffer B (DUO82049) and then washed in 0.01× wash buffer B. To amplify the tdTomato signal, sections were then incubated with donkey anti-guinea pig Cy3 (1:500; Jackson ImmunoResearch 706-165-148) for 45 min at room temperature. Sections were washed twice with PBS and then coverslips were mounted with DuoLink mounting medium with DAPI (DUO82040). Images were obtained in the same way as for IHC (described above) with a step size of 1 μm. Images were processed using Image J (v.2.1). Astrocyte territories were identified from tdTomato fluorescence, and the number of puncta within each territory was measured. Two negative controls were performed. The first was the same experiment as described above, but for astrocytes located in the dorsolateral striatum that lack μ-crystallin. The second was an independent experiment without the anti-rabbit PLUS primer probe.

## Statistical analysis

Statistical tests were run in OriginPro 2017 (OriginLab, v.9.4.2). Summary data are presented as mean ± s.e.m. Sample sizes were not determined in advance and were based on past studies that are cited at the relevant sections of the manuscript and methods. Statistical tests were chosen as described below. All replicates were biological, not technical. Blinding was not done. For each set of data to be compared, we used OriginPro to determine whether the data were normally distributed or not. If they were normally distributed, we used parametric tests. If the data were not normally distributed, we used non-parametric tests. Paired or unpaired Student's *t*-test, Wilcoxon signed-rank test, or Mann–Whitney tests were used for statistical analyses with two samples (as appropriate). One-way ANOVA, two-way ANOVA or repeated two-way

ANOVA tests followed by Tukey's post-hoc test were used for statistical analyses with more than three samples. Significant differences were defined as $P < 0.05$ and are indicated as such throughout.

### Reporting summary

Further information on research design is available in the Nature Portfolio Reporting Summary linked to this article.

## Data availability

scRNA-seq data have been deposited in the GEO (GSE225741). Raw and normalized RNA-seq data from all experimental groups have been deposited in the GEO (GSE228506). Proteomic data are provided at PRIDE (https://www.ebi.ac.uk/pride/archive/projects/PXD040991). scRNA-seq data for genes enriched in *Crym*⁺ astrocytes as well as those shared with other striatal astrocytes are provided in Supplementary Table 2. RiboTag RNA-seq data for astrocytes from *Crym* KO and control mice are provided in Supplementary Table 3. The μ-crystallin interactome is provided in Supplementary Table 4. The results of statistical comparisons, *n* numbers and *P* values are shown in the figures or figure legends and are reported in Supplementary Table 5. Source data are provided with this paper.

## Code availability

Proteomics: LFQ was performed by the MaxQuant software with the Andromeda search engine integrated (https://www.maxquant.org/). Principal component data visualization was performed with the R package Factoextra fviz 1.0.6 (https://rpkgs.datanovia.com/facto-extra/reference/fviz_pca.html). Differential protein expression and enrichment analysis was performed with the Bioconductor R package limma (https://bioconductor.org/packages/release/bioc/html/limma. html). Protein network visualization, including STRING analysis, was performed with Cytoscape v.3.8 (https://apps.cytoscape.org/apps/stringapp). The artMS package (https://bioconductor.org/packages/release/bioc/html/artMS.html) was used to reformat the MaxQuant results, to make them compatible with the SAINTexpress program. SAINT protein interaction probability scoring was done using http://saint-apms.sourceforge.net/Main.html. RNA-seq: differential gene expression and enrichment analysis used the R package limmaVoom to process RNA counts (https://rdrr.io/bioc/limma/man/voom.html) and batch correction was done with the R package ComBat (https://rdrr.io/bioc/sva/man/ComBat.html). scRNA-seq data were analysed with CellRanger (10X Genomics) (https://support.10xgenomics.com/single-cell-gene-expression/software/pipelines/latest/installation) and Seurat (https://CRAN.R-project.org/package=Seurat).

50. Divakaruni, A. S. et al. Inhibition of the mitochondrial pyruvate carrier protects from excitotoxic neuronal death. *J. Cell Biol.* **216**, 1091–1105 (2017).

**Acknowledgements** This work, B.S.K., M.O., K.E.L. and J.S.S. were supported by the NIH (DA047444, NS111583), the Allen Distinguished Investigator Award, a Paul G. Allen Frontiers Group advised grant of the Paul G. Allen Family Foundation and the Ressler Family Foundation. K.E.L. was supported by F32MH125598. J.S.S. was supported by the National Science Foundation (NSF-GRFP DGE-2034835) and by the UCLA Eugene V. Cota-Robles Fellowship. We thank F. Gao for assistance with RNA-seq; the UCLA Neuroscience Genomics Core for assistance with sequencing; F. Endo and M. Gangwani for guidance with Fig. 1a; A. Theint Theint for help with subcloning GCaMP8m; M. Kaur for help with tissue processing; N. Brecha for the GAT3 antibody; H. Chai and B. Diaz-Castro for discussions during the early stages; and A. Y. Huang and L. Wu for feedback.

**Author contributions** M.O. performed most of the experiments and analysed data. K.E.L. performed and analysed scRNA-seq. J.S.S. performed and analysed proteomic data with help from Y.J.-A. and J.A.W. J.S.S. also analysed scRNA-seq, performed RNAscope, performed behavioural experiments with SAPAP3⁻/⁻ mice and helped with PLA, validations and bioinformatics. S.L.M. performed Lucifer yellow iontophoresis. R.K. helped to guide RNA-seq analyses, and A.E.J. and A.S.D. helped with metabolomics. B.S.K. conceived and directed the experiments, and planned the figures with M.O. B.S.K. also performed some electrophysiology and wrote the paper with help from the other coauthors.

**Competing interests** The authors declare no competing interests.

**Additional information**
**Correspondence and requests for materials** should be addressed to Baljit S. Khakh.

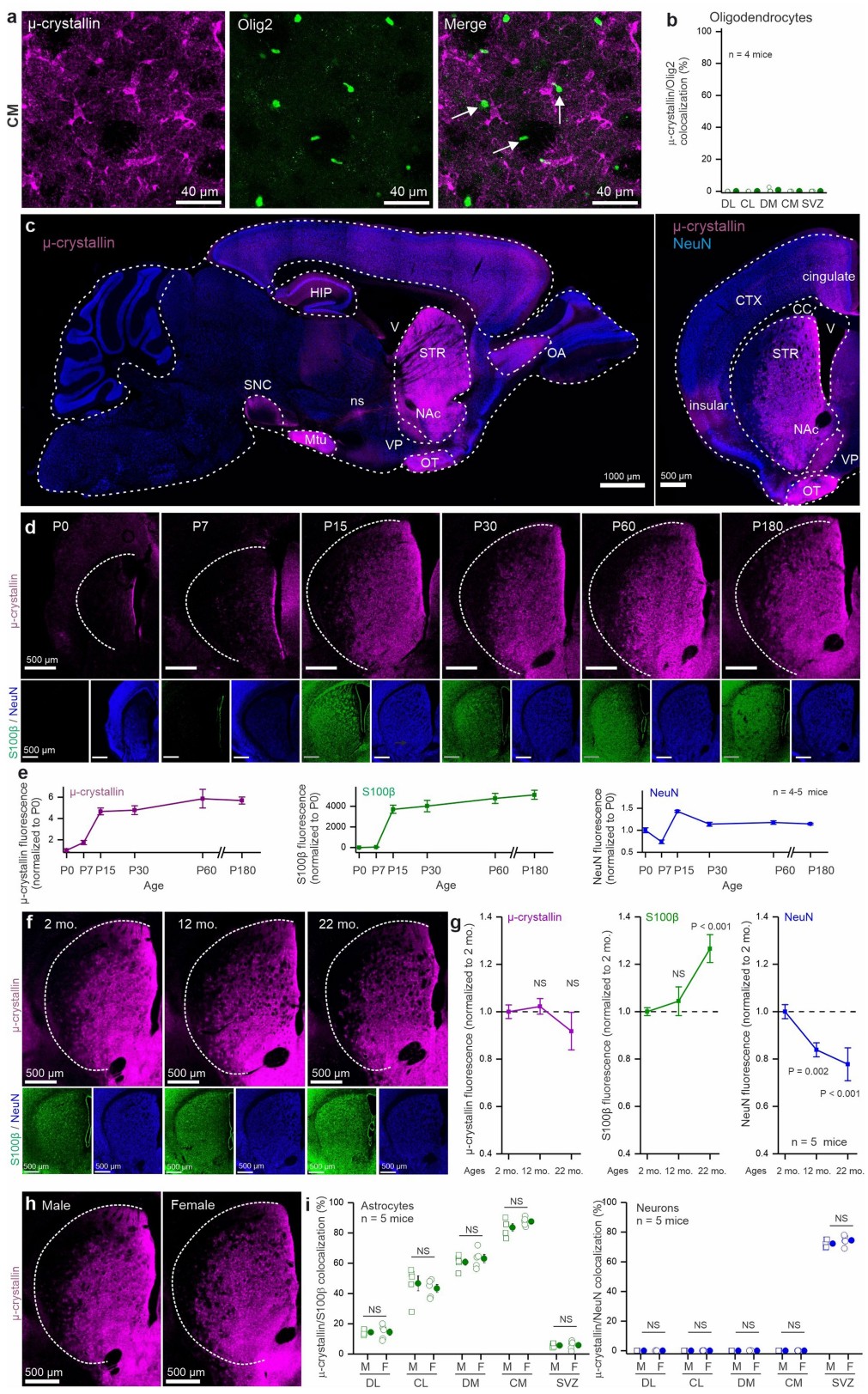

**Extended Data Fig. 1** | See next page for caption.

**Extended Data Fig. 1 | µ-crystallin immunostaining. a**,**b**, µ-crystallin is not expressed in Olig2+ oligodendrocytes on the basis of representative images (**a**) and average data across mice and across regions of the striatum (**b**; n = 4 mice) where µ-crystallin was expressed in astrocytes (see also main text of manuscript and Fig. 1). **c**, Sagittal (left) and coronal (right) whole brain images show µ-crystallin (magenta) and NeuN (blue) expression. (V, ventricle; HIP, hippocampus; CTX, cortex; STR, striatum; OA, olfactory area; VP: ventral pallidum; NAc, nucleus accumbens; SNC, substance nigra compact; ns, nigrostriatal bundle; MTu, medial tuberal nucleus; OT, olfactory tubercule). **d**, µ-crystallin (magenta), S100β (green) and NeuN (blue) striatal expression during postnatal development from P0 to P180 (from left to right). Scale bars = 500 µm. **e**, Quantification of µ-crystallin, S100β, and NeuN expression in the central striatum (n = 4 mice for P0, P15, P30, P60 and n = 5 mice for P7, P180). **f–i**, µ-crystallin expression in different ages (2, 12, and 22 months in **f**,**g**) as well as in male and female mice at 2 months (**h**,**i**). There was no significant change in µ-crystallin expression at 12 and 22 months of age in relation to 2-month-old mice, which are approximately the age used in most of our work (n = 5 mice; One-way ANOVA tests followed by Tukey's post-hoc test, overall ANOVA P = 0.57 for µ-crystallin, 0.0021 for S100ß and 0.0018 for NeuN at 22 months old). There was also no difference in the expression of µ-crystallin between male and female mice (n = 5 mice; two-way ANOVA tests followed by Tukey's post-hoc test). Average data are shown as mean ± s.e.m. and all statistics are reported in Supplementary Table 5.

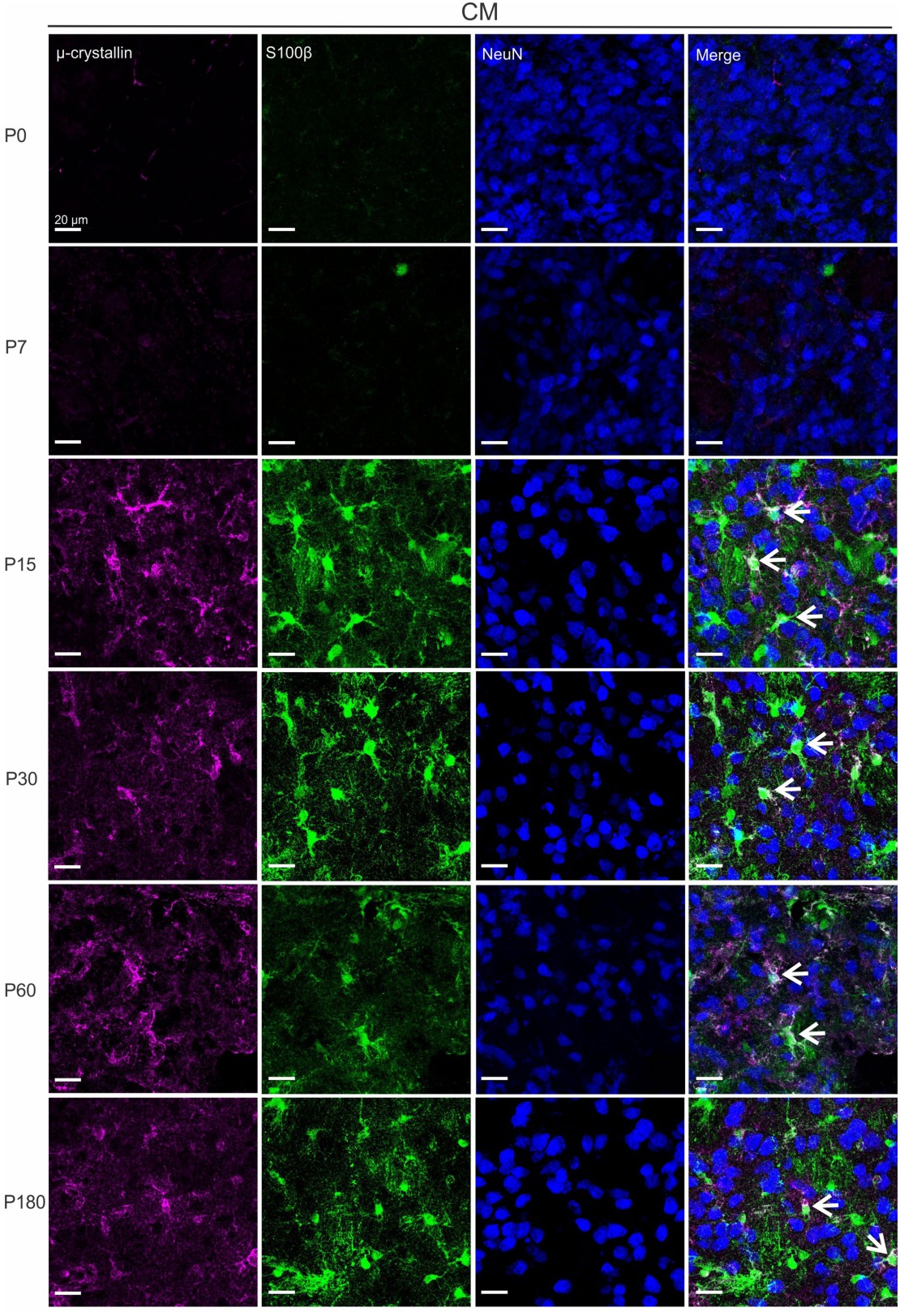

**Extended Data Fig. 2 | μ-crystallin expression in the central striatum during postnatal development.** Images of μ-crystallin (magenta), S100β (green) and NeuN (blue) protein expression in the central striatum (CM) during postnatal development from P0 to P180. White arrowheads show μ-crystallin-positive astrocytes. Scale bars, 20 μm.

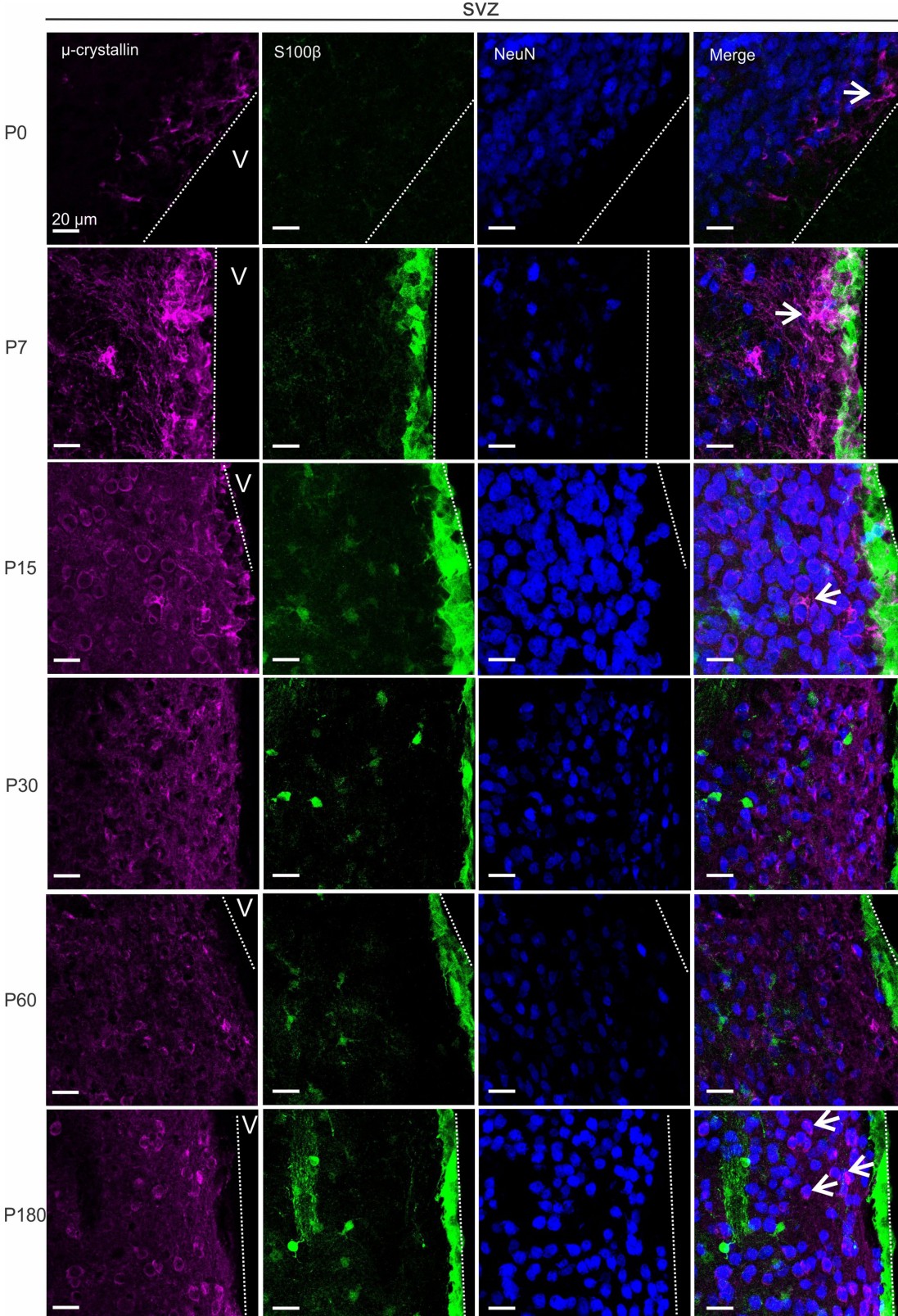

**Extended Data Fig. 3 | μ-crystallin expression in the SVZ during postnatal development.** Images of μ-crystallin (magenta), S100β (green) and NeuN (blue) protein expression in the SVZ during postnatal development from P0 to P180 (V, Ventricle). White arrowheads show μ-crystallin-positive cells. Scale bars, 20 μm.

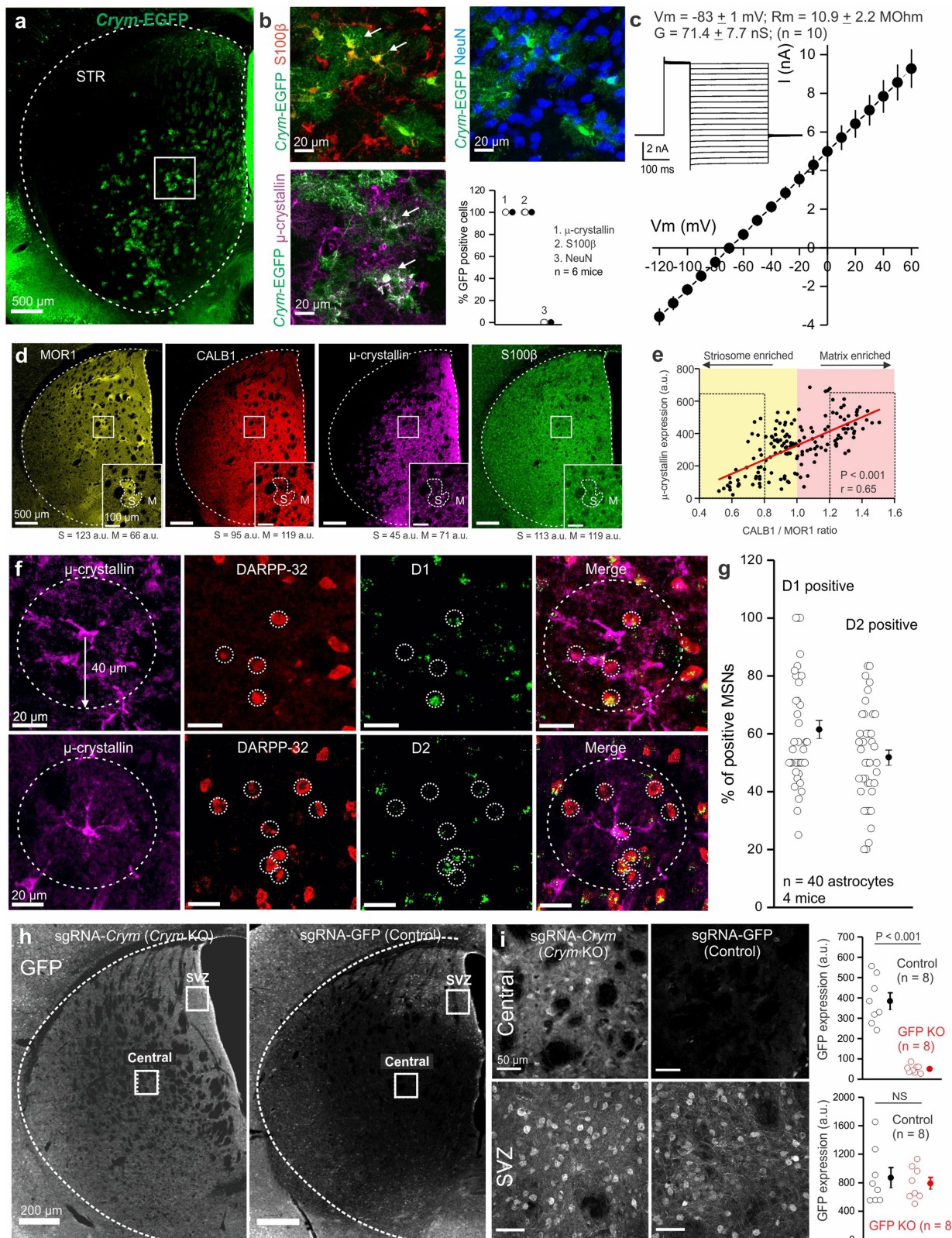

**Extended Data Fig. 4** | See next page for caption.

**Extended Data Fig. 4 | Further characterization of μ-crystallin-positive astrocytes. a**, Whole striatal image of *Crym*-EGFP BAC transgenic mice show bushy GFP positive cells in the central and ventral striatum. **b**, Merged images of GFP positive cells (green) with S100β (red), μ-crystallin (magenta), and NeuN (blue). Scatter graph shows the percent of GFP positive cells who are S100β, μ-crystallin or NeuN positive (n = 6 mice). **c**, Whole-cell voltage clamp performed on GFP positive cells in the central striatum. The waveforms, the current-voltage curve and the membrane properties correspond to astrocyte membrane properties (n = 10 cells from 4 mice). **d**, Striatal expression of the striosome marker MOR1 (yellow), matrix enriched protein CALB1 (red), μ-crystallin (magenta), and S100β (green). Inset panels show zoomed in images and corresponding intensity values (a.u.) for striosome (S) and Matrix (M) compartments. **e**, Correlation of μ-crystallin expression and CALB1/MOR1 ratio. A ratio <1 reflects striosome compartment and a ratio > 1 reflects matrix compartment enrichment, respectively. The Pearson correlation coefficient shows a positive correlation (two-tailed Pearson correlation test, r = 0.65; P = 7.6 ×$10^{-21}$) with matrix enrichment (161 ROIs from n = 4 mice). μ-crystallin expression levels were 173 ± 22 and 458 ± 15 a.u. in striosome and matrix, respectively (n = 4 mice). **f**, Representative images of μ-crystallin-positive astrocytes and DARPP-32, D1 (top) or D2 (bottom) mRNA positive MSNs (green). **g**, D1 and D2 positive MSNs were counted in an area of 80 μm diameter surrounding each μ-crystallin-positive astrocyte. Scatter graph shows the percent of D1 or D2 positive MSNs within a *Crym* positive astrocyte territory (n = 40 astrocytes from 4 mice). **h,i**, CRISPR–Cas9-mediated deletion of GFP in the central striatum. **h**, Whole striatal images show GFP expression in mice injected with control AAV (sgRNA-GFP) or a *Crym* KO AAV (sgRNA-*Crym*). **i**, Images of GFP expression in the central striatum and SVZ. Scatter graphs show that GFP expression was strongly reduced in the central striatum but not changed in the SVZ in GFP KO mice (n = 8 mice, two-tailed Mann–Whitney, P = 9.4 × $10^{-4}$, and two-tailed two-sample *t*-test). Average data are shown as mean ± s.e.m. and all statistics are reported in Supplementary Table 5.

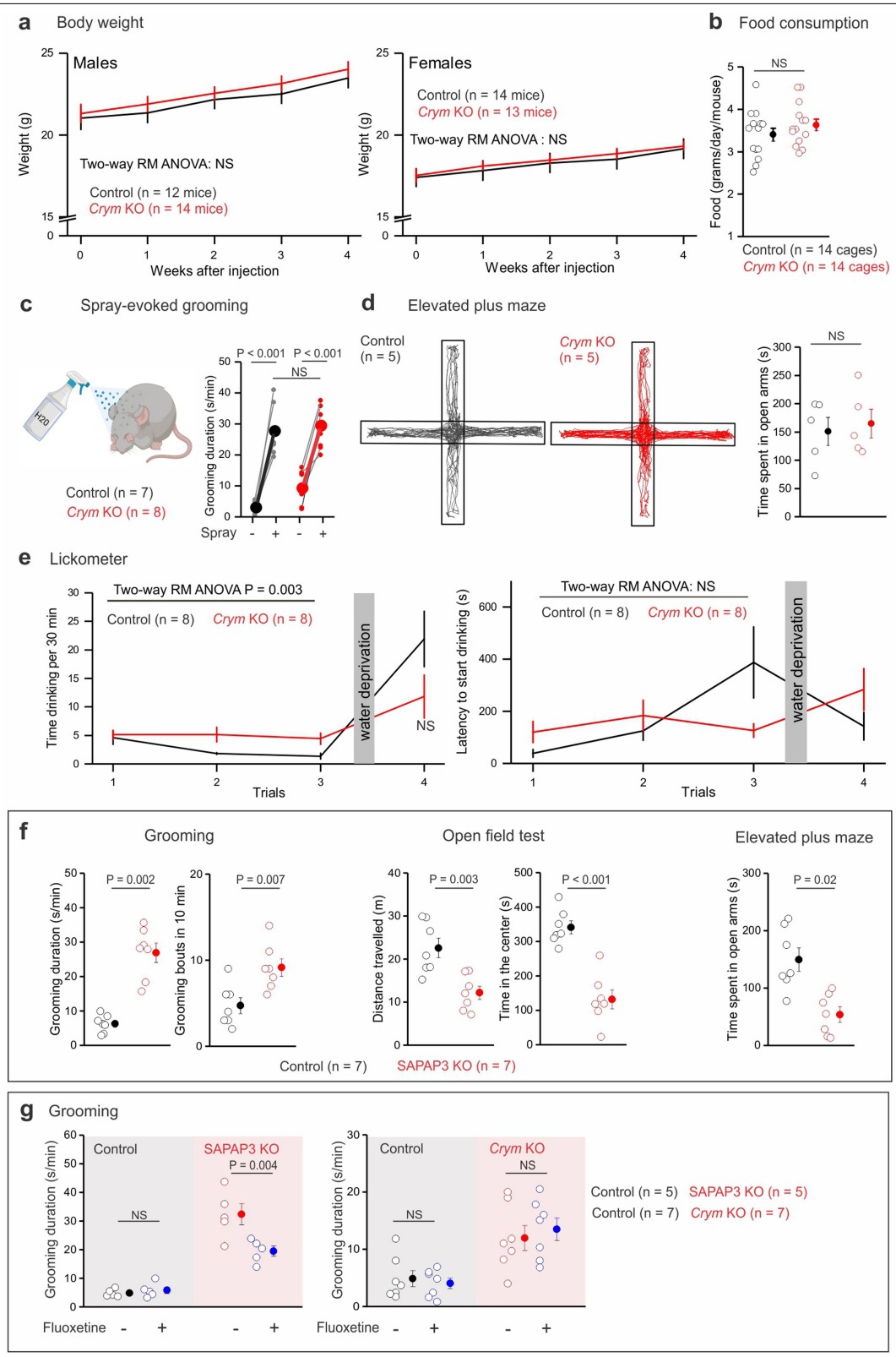

**Extended Data Fig. 5** | See next page for caption.

**Extended Data Fig. 5 | Additional behaviours and comparison with SAPAP3$^{-/-}$ mice. a**, Body weight of males (left) and females (right) before and after AAV injection for 4 weeks (two-way repeated-measures ANOVA followed by Tukey's post-hoc test). **b**, Scatter graph shows the home cage food consumption in control and *Crym* KO mice (n = 14 cages with more than 2 mice per cage; two-sample t-test). **c**, Schematic and graph show grooming duration evoked by spraying the mice with water for control and *Crym* KO mice (n = 7 control mice and n = 8 *Crym* KO mice; two-way repeated-measures ANOVA followed by Tukey's post-hoc test, P = 8.9 x 10$^{-8}$). **d**, Representative 20 min elevated plus maze recording for control and *Crym* KO mice. Scatter graph shows the time spent in open arms for control and *Crym* KO mice (n = 5 mice; two-tailed two-sample t-test). **e**, Scatter graphs show the time drinking and the latency to start drinking over 30 minutes for each trial completed for 4 days (n = 8 mice; two-way repeated-measures ANOVA followed by Tukey's post-hoc test). **f**, Scatter graphs show self-grooming duration, grooming bouts, distance travelled, time spent in the centre, and time spent in the open arms in control and SAPAP3 KO mice (n = 7 mice; two-tailed Mann–Whitney test or two-tailed two-sample t-test, P = 4 x 10$^{-5}$ for the time in the centre). **g**, Scatter graphs show self-grooming duration after fluoxetine treatment (blue) in control and SAPAP3 KO mice (left; n = 5 mice; two-way repeated-measures ANOVA followed by Tukey's post-hoc test) and in control and *Crym* KO mice (right; n = 7 mice; two-way repeated-measures ANOVA followed by Tukey's post-hoc test). Average data are shown as mean ± s.e.m. and all statistics are reported in Supplementary Table 5.

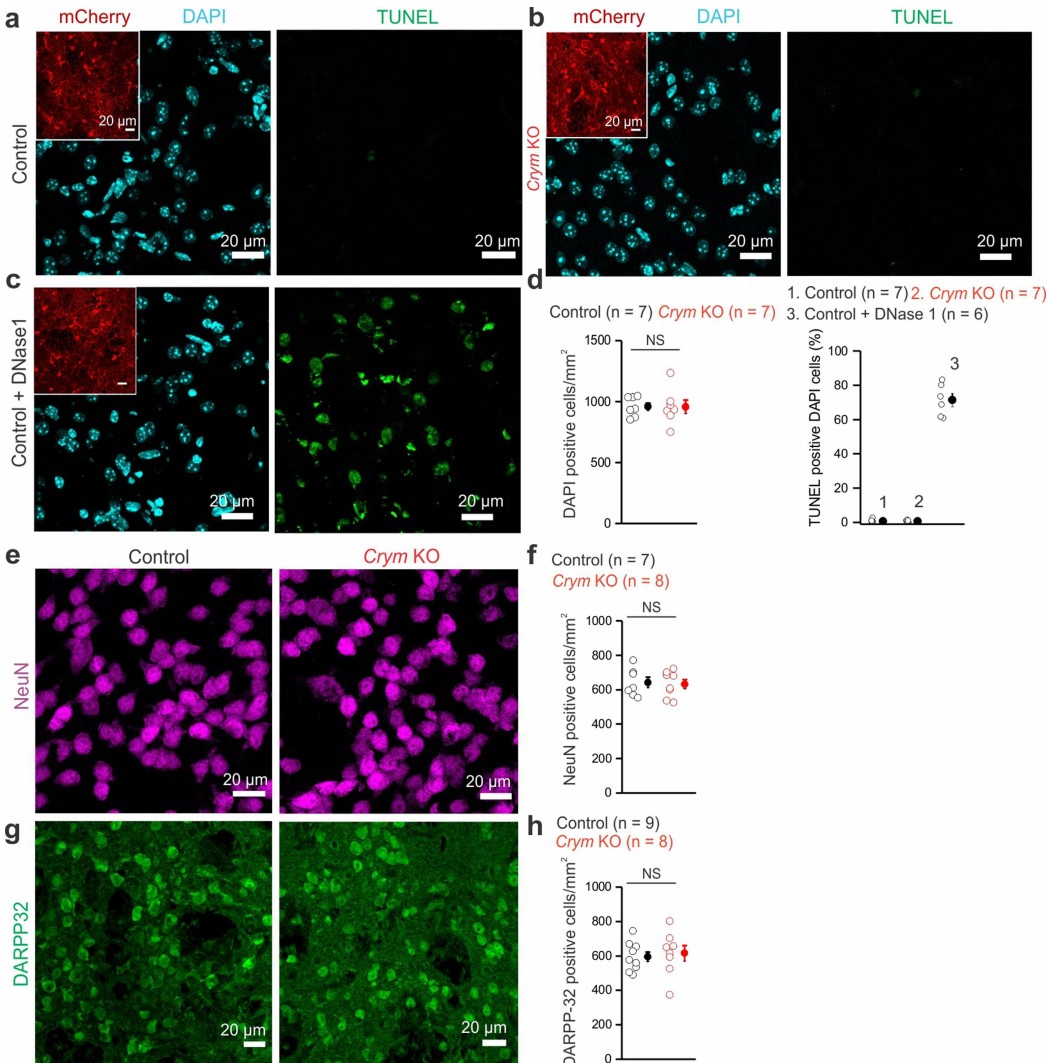

**Extended Data Fig. 6 | No change in apoptosis and neuronal markers in control and *Crym* KO astrocytes. a–c**, Representative images of TUNEL (green), mCherry (red), and DAPI (blue) in control (**a**), *Crym* KO (**b**), and control + DNase1 (**c**) mice. **d**, Scatter graphs show the number of DAPI positive cells (left) and TUNEL positive cells (right) (n = 7 mice for control and Crym KO and n = 6 mice for control + DNase 1, two-tailed two-sample t-test). There were no differences in the number of DAPI+ cells between controls and *Crym* KO mice and no TUNEL staining was observed, except for the positive control. **e,f**, Representative images (**e**) and scatter graphs (**f**) of the number of NeuN positive cells in 7 control mice and 8 *Crym* KO mice (two-tailed two-sample t-test). **g**,**h**, Representative images (**g**) and scatter graphs (**h**) of the number of DARPP-32 positive cells in 9 control mice and 8 *Crym* KO mice (two-tailed two-sample t-test). Average data are shown as mean ± s.e.m. and all statistics are reported in Supplementary Table 5.

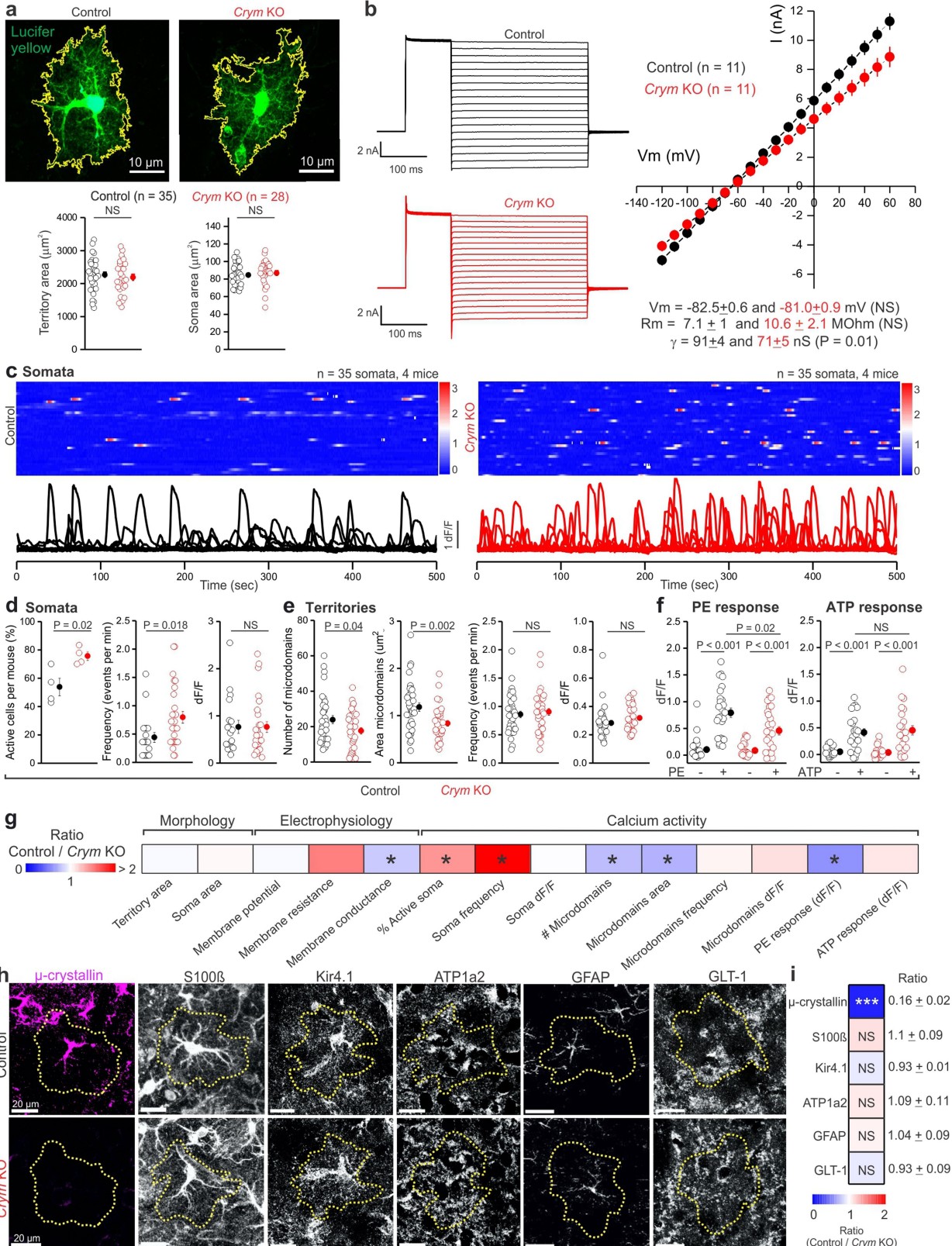

**Extended Data Fig. 7 |** See next page for caption.

**Extended Data Fig. 7 | Functional characterization of control and *Crym* KO astrocytes. a**, Representative images for control and *Crym* KO striatal astrocytes filled with Lucifer yellow by iontophoresis. Bottom scatter graphs show the territory area (left) and soma area (right) (n = 35 cells for control and n = 28 cells for *Crym* KO from 4 mice, two-tailed Mann–Whitney test or two-tailed two-sample t-test). **b**, Whole-cell voltage clamp performed in control and *Crym* KO striatal astrocytes. Representative currents waveforms, average current-voltage relationships, membrane potential (mV), membrane resistance (MOhm) and slope conductance (nS) are shown (n = 11 cells from 4 mice, two-tailed two-sample t-test and two tailed Mann-Whitney test). **c**, Kymographs and traces representing the ΔF/F of $Ca^{2+}$ signals in astrocyte somata. **d**, Scatter graphs show the percent of active cells per mouse (n = 4 mice, two-tailed two-sample t-test), frequency, and ΔF/F in somata (n = 35 cells from 4 mice, two-tailed Mann–Whitney tests). **e**, Scatter graphs show the number, area, frequency, and ΔF/F of calcium signals in the astrocyte territories (n = 35 cells from 4 mice, two-tailed Mann–Whitney test or two-tailed two-sample t-test).

**f**, Scatter graphs of $Ca^{2+}$ signals, represented by ΔF/F, evoked by 10 μm phenylephrine (PE; n = 26 cells from 4 mice) (left) or by 100 μm ATP (right; n = 23 cells from 4 mice) in control and *Crym* KO astrocytes (Two-way ANOVAs followed by Tukey's post-hoc test, ANOVA p-value overall genotype = $7.6 \times 10^{-3}$ for PE and $8.4 \times 10^{-1}$ for ATP). **g**, Heat map summarizes the ratio of metrics from control vs *Crym* KO of all parameters assessed (* = P < 0.05). Statistical tests and P values for the heat map are from the corresponding data reported in the earlier panels in the figure. **h,i**, Marker expression in control and *Crym* KO astrocytes. Representative images (**h**) and ratios (**i**) for the expression of the various canonical astrocyte markers indicated. μ-crystallin was significantly reduced, but the other markers were not. The heat map shows the ratio of control vs *Crym* KO (n = 30 cells from 5 mice for μ-crystallin, S100ß, Kir4.1, GLT1 and n = 24 cells from 5 mice for ATP1a2, GFAP; Mann–Whitney test or two-sample *t*-test, *** P = $3 \times 10^{-11}$, NS = non-significant). Average data are shown as mean ± s.e.m. and all statistics are reported in Supplementary Table 5.

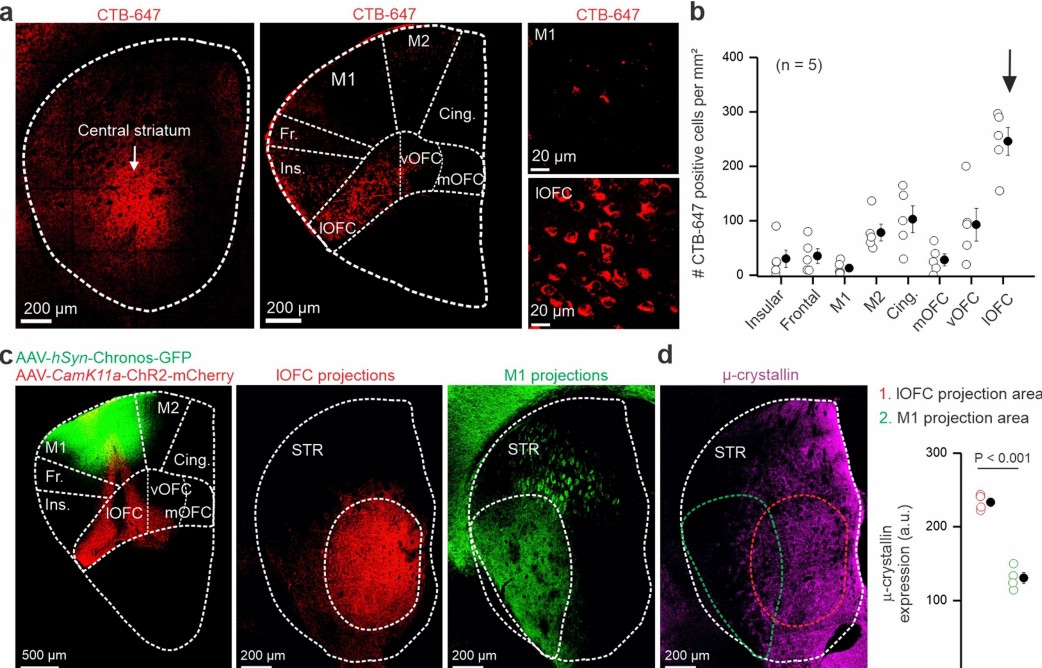

**Extended Data Fig. 8 | Retrograde and anterograde evaluations of major cortical projections to the central striatum, in which *Crym*⁺ astrocytes are abundant. a**, Representative images show retrograde labelling using Alexa 647-conjugated cholera toxin subunit B (CTB-647). Left image shows the CTB-647 injection site in the central striatum (white arrow). Central image shows CTB-647 labelled neurons in different part of the cortex (M1, motor cortex 1; M2, motor cortex 2; Cing., cingulate cortex; Fr., frontal cortex; Ins., insular cortex; OFC, orbitofrontal cortex medial (m), ventral (v) or lateral (l)). Right

images show CTB-647 labelled neurons in M1 and lOFC. **b**, Scatter graph of the number of CTB-647 positive neurons in different parts of the cortex (n = 5 mice). **c**, AAV1-*hSyn*-Chronos-GFP and AAV9-*CamK11a*-ChR2-mCherry were injected into M1 and lOFC to label the projections in the striatum. Images show lOFC and M1 projections in the striatum. **d**, Image and scatter graph show μ-crystallin expression in the M1 and lOFC projection area (n = 4 mice; two-tailed two-sample t-test, P = 3.2 x 10⁻⁵). Average data are shown as mean ± s.e.m. and all statistics are reported in Supplementary Table 5.

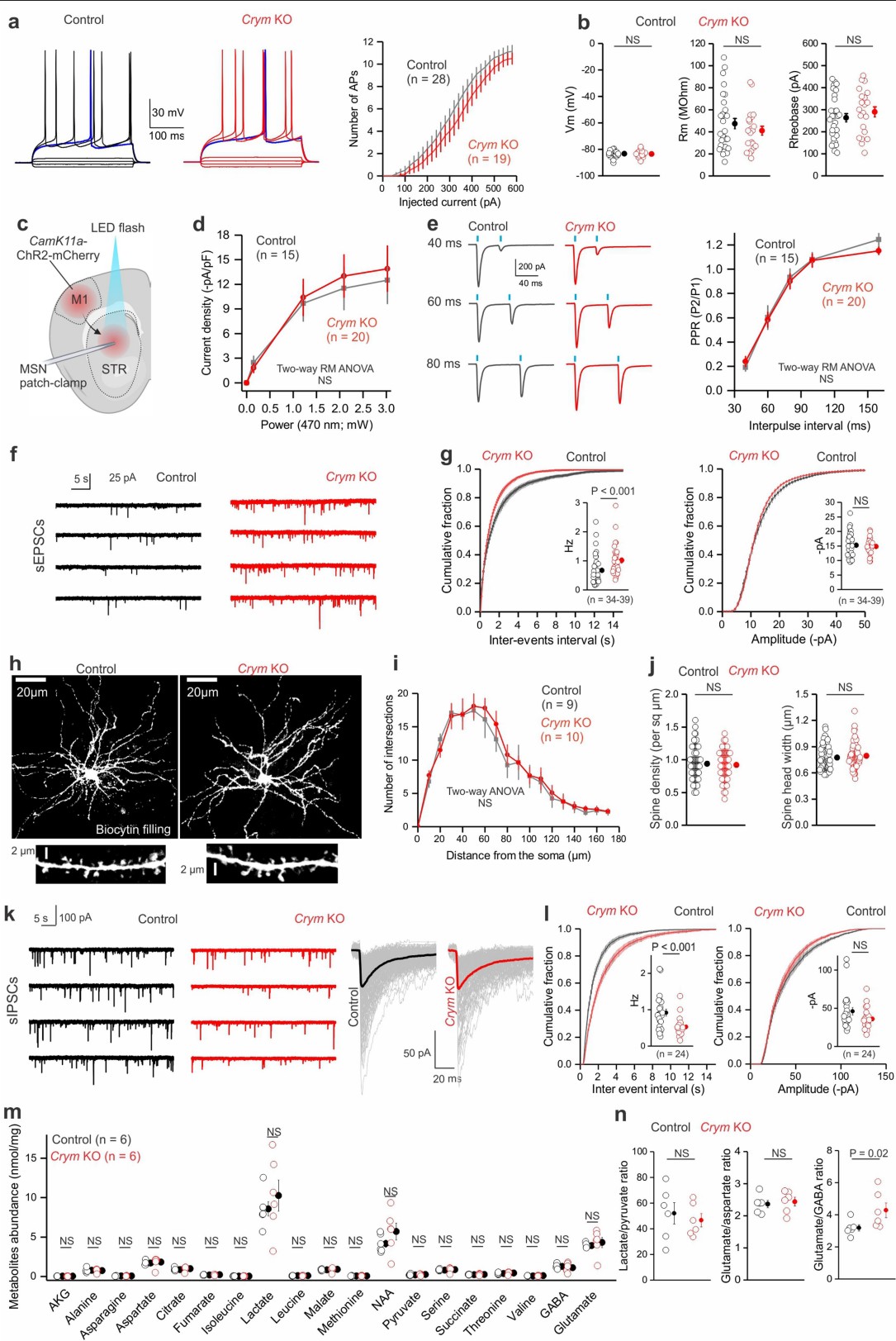

**Extended Data Fig. 9** | See next page for caption.

**Extended Data Fig. 9 | Further MSN properties in *Crym* KO mice. a**, Traces of MSN membrane responses to current injection (left) and relationship of injected current to number of APs (right) in control and *Crym* KO mice. **b**, Scatter graphs show the MSN resting membrane potential, membrane resistance, and rheobase in control (n = 28 cells from 11 mice) and *Crym* KO mice (n = 19 cells from 11 mice; two-tailed Mann–Whitney test or -two-tailed two-sample t-test). **c**, Cartoon of AAV2-*CamK11a*-ChR2-mCherry injection into the M1. Recordings were done in the central striatum. **d**, Graph shows the current density after brief pulses of 470 nm light (2 ms) at different power in control (n = 15 cells from 5 mice) and *Crym* KO (n = 20 cells from 5 mice; two-way repeated measure ANOVA). **e**, Representative traces and graph for evoked EPSCs due to paired stimuli in control (n = 15 cells from 5 mice) and *Crym* KO mice (n = 20 cells from 5 mice; two-way repeated measure ANOVA). **f**, Representative sEPSC traces from one individual representative MSN from control and *Crym* KO mice. **g**, Cumulative probability graph for the inter-event interval (left) and for the amplitude (right). Pooled data for the frequency and amplitude are shown in the inset bar graph (n = 39 cells from 16 mice for control and n = 34 cells from 14 mice for *Crym* KO, two-tailed Mann–Whitney t-tests, P = 2.8 x 10$^{-4}$). **h**, Images of biocytin-labelled MSNs and dendritic spines in control (left) and *Crym* KO (right) mice.

**i**, Sholl analyses performed with increments of 10 μm diameter (n = 9 MSNs in control and n = 10 MSNs in *Crym* KO from 5 mice; two-way repeated measure ANOVA). **j**, Scatter graphs show spine density and spin head width in control (n = 74 dendrites from 10 MSNs from 5 mice) and *Crym* KO mice (n = 86 dendrites from 11 MSNs from 5 mice; two-tailed two-sample t-test and two-tailed Mann–Whitney test). **k**, Representative sIPSC current traces (left) and individual traces and average (right) from one representative MSN from control and *Crym* KO mice. **l**, Cumulative probability graph for the inter-event interval (left) and for the amplitude (right). Pooled data for the frequency and amplitude are shown in the inset bar graph (n = 24 cells from 5 mice, two-tailed Mann–Whitney tests, P = 9.3 ×10$^{-4}$). **m**, Scatter graph of 17 metabolites and 2 neurotransmitters (GABA and glutamate) measured by mass spectrometry (n = 6 mice; two-tailed Mann–Whitney test or two-tailed two-sample t-test; AKG as alpha-ketoglutarate and NAA as N-acetyl-aspartate). **n**, Scatter graphs show lactate/pyruvate, glutamate/aspartate, and glutamate/GABA ratio in control and *Crym* KO mice (n = 6 mice; two-tailed Mann–Whitney test or two-tailed two-sample t-test. Average data are shown as mean ± s.e.m. and all statistics are reported in Supplementary Table 5.

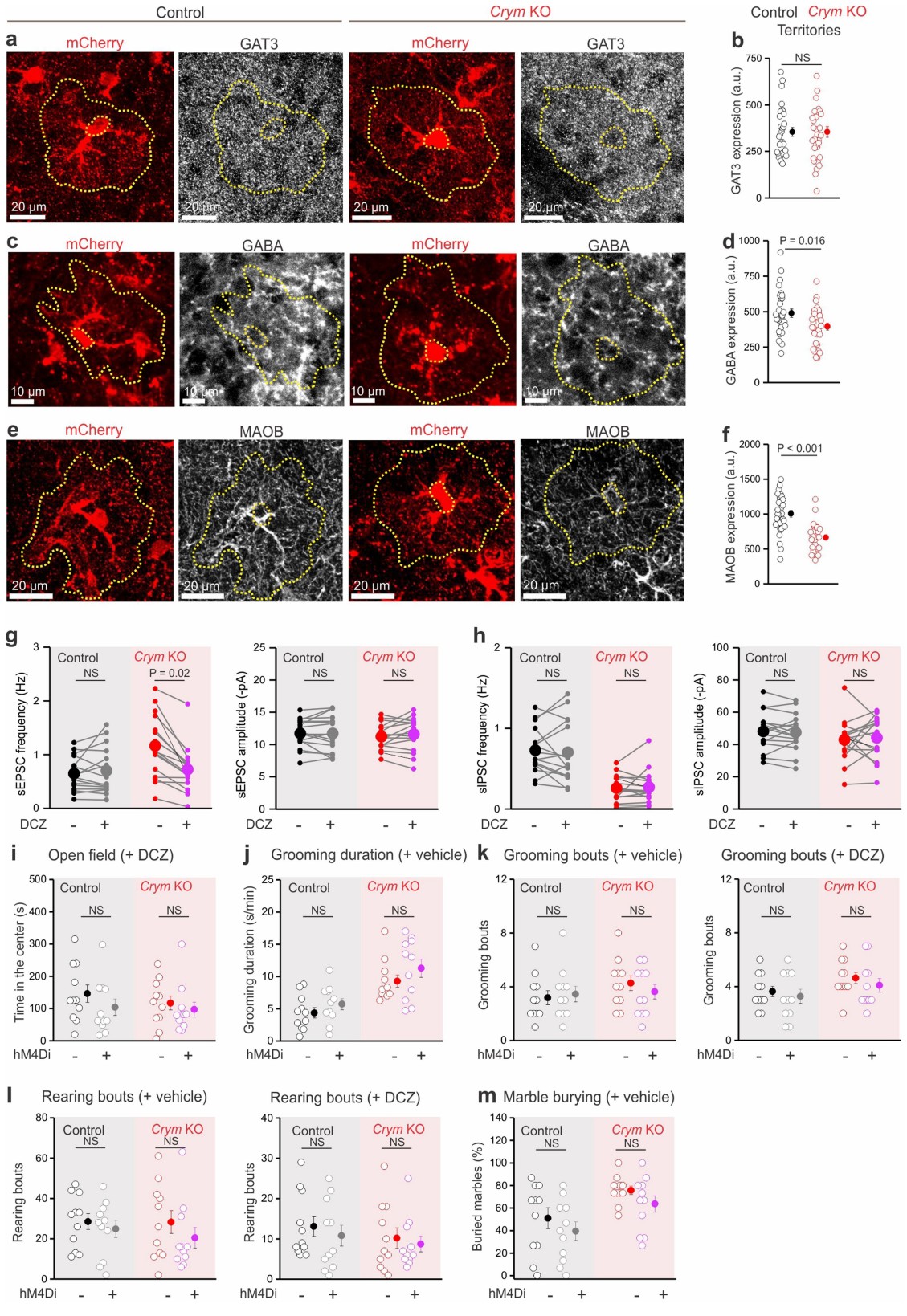

**Extended Data Fig. 10 |** See next page for caption.

**Extended Data Fig. 10 | Assessments of GAT3, GABA and MAOB in control and *Crym* KO astrocytes, and supportive data for presynaptic chemogenetics.** **a**–**f**, Representative images and quantification of the expression of GAT3 (**a**,**b**), GABA (**c**,**d**) and MAOB (**e**,**f**) in control (left) and *Crym* KO astrocytes (right) (n = 30 cells from 6 mice for GABA and MAOB, n = 32 cells from 6 mice for GAT3; two-tailed Mann–Whitney test or two-tailed two-sample t-test, $P = 6.5 \times 10^{-8}$ for MAOB). **g**–**m**, Presynaptic chemogenetics supportive data. **g**, Graphs show the sEPSC frequency (left) and amplitude (right) of control and *Crym* KO mice before and after 200 nM DCZ application (n = 16 cells from 4 mice, two-tailed paired-sample t-test and two-tailed paired sign test). **h**, As in **g**, but for sIPSC (n = 16 cells from 4 mice, two-tailed paired-sample t-test and two-tailed paired sign test). **i**–**m**, Graphs show the time in the centre (**i**), grooming duration (**j**), grooming bouts (**k**), rearing bouts (**l**) and % of buried marbles (**m**) of control and *Crym* KO mice with or without rg-h4DMi after DCZ or vehicle treatment (n = 11 mice per group; two-way ANOVA followed by Tukey's post-hoc test). Average data are shown as mean ± s.e.m. and all statistics are reported in Supplementary Table 5.

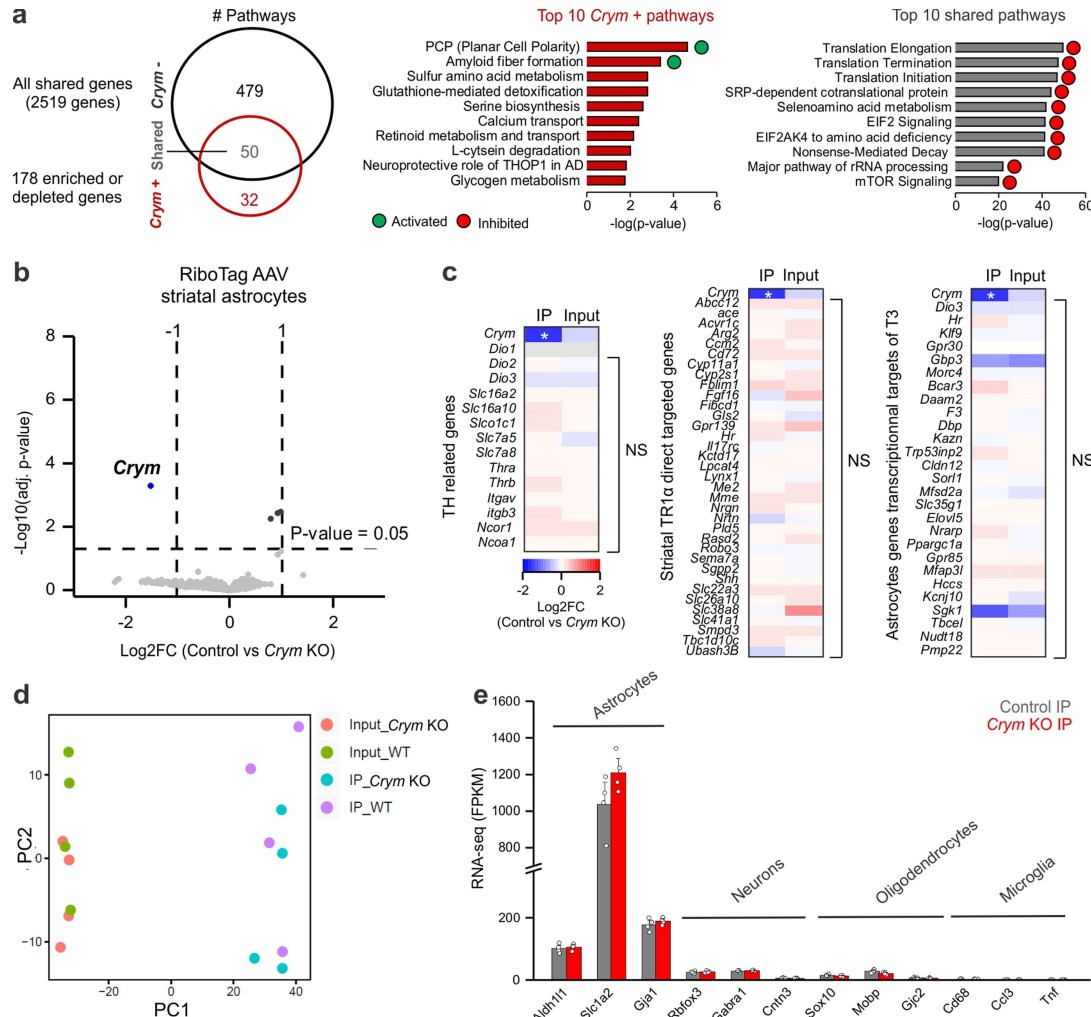

**Extended Data Fig. 11 | Molecular mechanisms of μ-crystallin in striatal astrocytes and supplementary data for RNA-seq. a**, Unique and common pathways for the 2,519 shared genes versus the 178 enriched and depleted genes in the *Crym+* population. Top 10 shared pathways for the *Crym+* astrocytes versus other striatal astrocytes (grey) and top 10 unique pathways for *Crym+* astrocytes (red) are shown. **b**, Volcano plot of differentially expressed genes (DEGs) (limmaVoom, FDR < 0.05) shows the numbers of up- and downregulated astrocyte genes between *Crym* KO IP and control IP. These analyses were restricted to genes with >2-fold enrichment in the IP compared with the input. Only *Crym* was found significantly downregulated. **c**, Heat maps of Log2FC for TH related genes, striatal TR1α directly targeted genes, and astrocytic transcriptional targets of T3 (*P = 5.1 x 10⁻⁴; NS, not significant). (**d**) Striatal astrocyte principal component (PCA) analysis of the 500 most variable genes across 8 samples. **e**, Gene-expression levels (in fragments per kilobase per million, FPKM) of cell-specific markers for astrocytes, neurons, oligodendrocytes, and microglia after a local injection of astrocyte-selective RiboTag AAV in the central striatum (IP samples) (n = 4 mice). Average data are shown as mean ± s.e.m. and all statistics are reported in Supplementary Table 3.

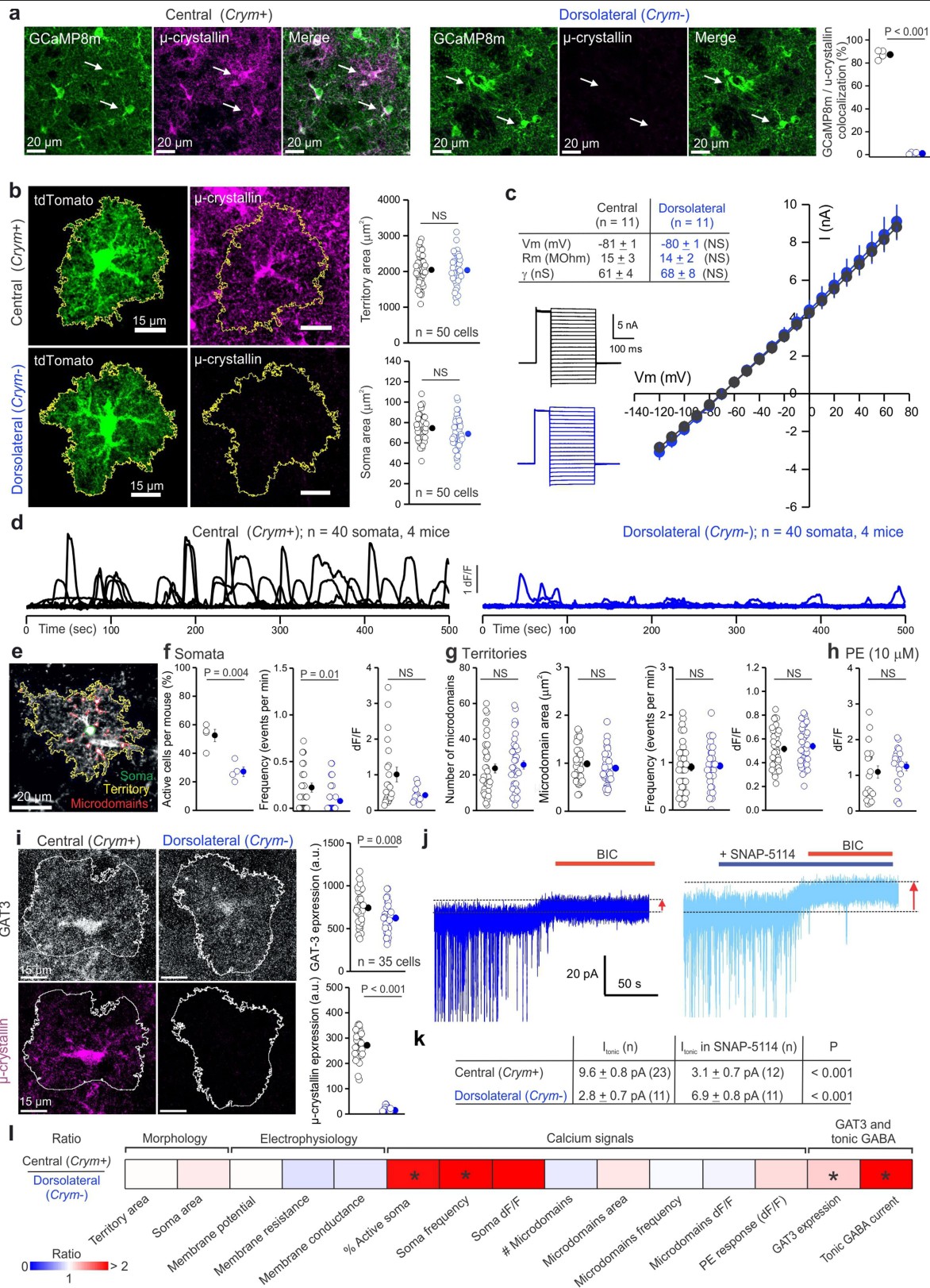

**Extended Data Fig. 12 | Functional properties of *Crym⁺* and *Crym⁻* astrocytes. a**, GCaMP8m expression in central (*Crym⁺*) and dorsolateral (*Crym⁻*) astrocytes. Representative images for central and dorsolateral striatal astrocytes expressing cytosolic GCaMP8m (white arrows show GCaMP8m expressing astrocytes). Scatter graph shows the percent co-localization between GCaMP8m and μ-crystallin (n = 4 mice, two-tailed two-sample t-test, $P = 1.3 \times 10^{-8}$) There was no co-localization between μ-crystallin and GCaMP8m in the dorsolateral region, because astrocytes in this region do not express *Crym*. **b**, Representative images for central (*Crym⁺*) and dorsolateral (*Crym⁻*) striatal astrocytes expressing tdTomato (left) and μ-crystallin (right). Scatter graphs show the territory and soma area (n = 50 cells from 5 mice, two-tailed two-sample t-test). **c**, Whole-cell voltage clamp recordings from central and dorsolateral striatal astrocytes. Representative current waveforms, average current-voltage relationships, membrane potential (mV), membrane resistance (MOhm) and slope conductance (nS) are shown (n = 11 cells from 4 mice, two-tailed two-sample t-test). **d**, Traces representing the ΔF/F of $Ca^{2+}$ signals in somata of central (*Crym⁺*) and dorsolateral (*Crym⁻*) astrocytes. Astrocytes from the central striatum were more active. **e**, Representative image of GCaMP8m expressing astrocyte showing somatic, territory and microdomain regions of interest. **f**, Scatter graphs show the percent of active cells per mouse when assessed over 500 s (n = 4 mice), the frequency of $Ca^{2+}$ signals per cell, and their ΔF/F in somata (n = 40 cells from 4 mice, two-tailed two-sample t-test and two-tailed Mann-Whitney test). As seen in the representative traces in **d**, astrocytes in the central striatum were more active within their somata. **g**, Scatter graphs show the number, area, frequency, and ΔF/F of $Ca^{2+}$ signals in astrocyte territories (n = 40 cells from 4 mice, two-tailed Mann–Whitney or two-tailed two-sample t-test); there were no differences. **h**, Scatter graphs of $Ca^{2+}$ signals, represented by ΔF/F, evoked by 10 μM phenylephrine (PE). There were no differences (n = 20 cells from 4 mice; two-tailed two-sample t-test). **i**, Representative images (left) and quantification (right) of the expression of GAT3 and μ-crystallin in central and dorsolateral striatal astrocytes (n = 35 cells from 4 mice, two-tailed Mann–Whitney or two-tailed two-sample t-test, $P = 6.5 \times 10^{-13}$ for μ-crystallin). GAT3 was higher in the central striatal astrocytes. **j**, Current recordings in voltage clamp (−60 mV) from MSNs in dorsolateral striatal astrocytes. Dashed lines and arrows indicate the changes of baseline holding current induced by application of bicuculline (BIC = 25 μM) or bicuculline after a pre-application of GAT3 inhibitor SNAP-5114 (40 μM). For comparison, traces for central astrocytes are shown in Fig. 3. **k**, The inset table summarizes the tonic GABA currents for central and dorsolateral striatal astrocytes. Tonic GABA currents were larger in the central striatum and were reduced by a pre-application of the GAT3 blocker, SNAP-5114, indicating GAT3 contributed GABA to the extracellular space. However, in the dorsolateral striatum, tonic GABA currents were smaller and increased by pre-application of SNAP-5114, indicating that in this region GAT3 removed GABA from the extracellular space. **l**, Heat map shows the ratio of the various metrics for central (*Crym⁺*) vs dorsolateral (*Crym⁻*) astrocytes for all the parameters assessed above (* = P < 0.05). Statistical tests and P values for the heat map are from the corresponding data reported in the earlier panels in the figure. Average data are shown as mean ± s.e.m. and all statistics are reported in Supplementary Table 5.

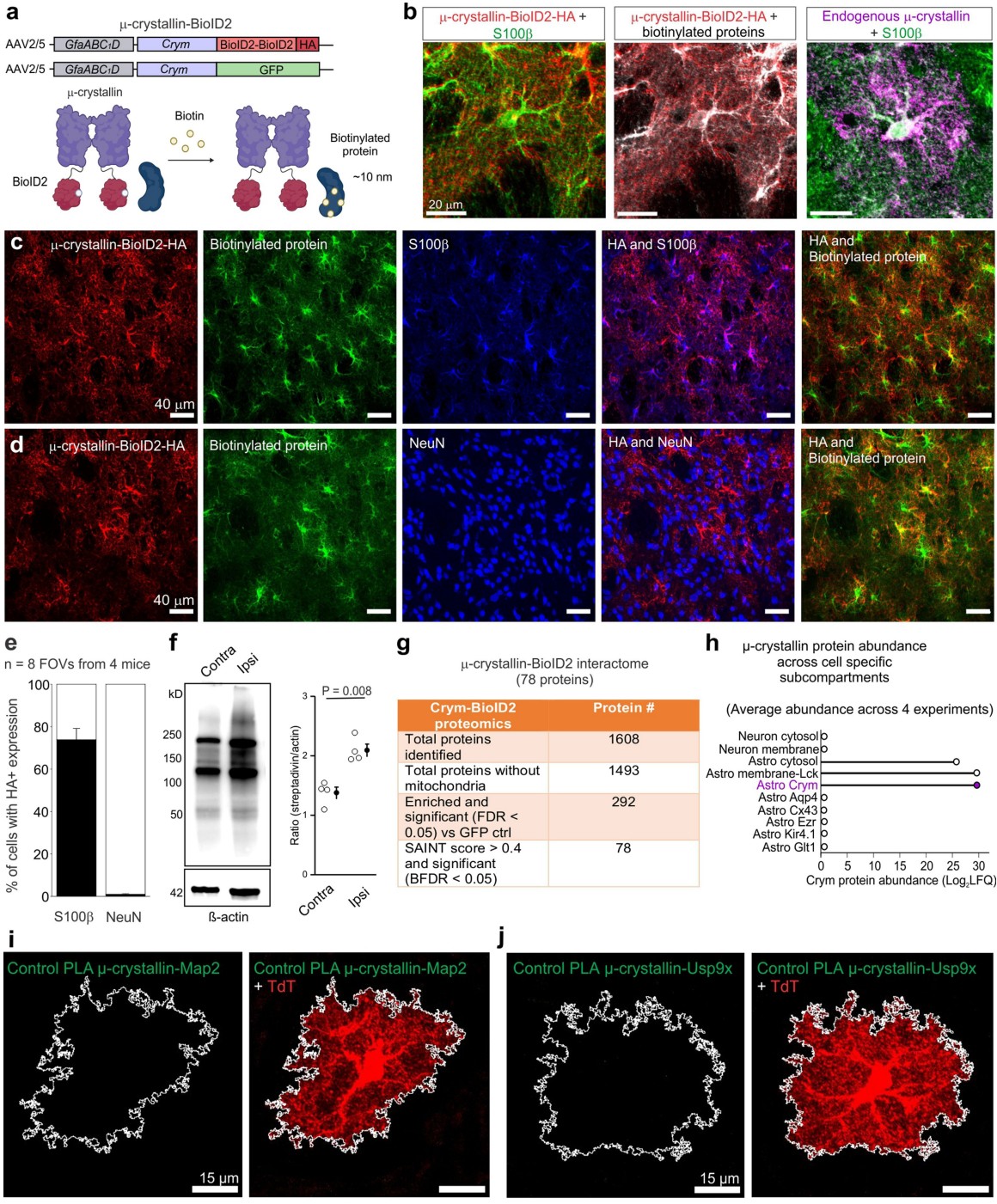

**Extended Data Fig. 13 | Characterization and validation of *Crym*-BioID2 for proteomics. a**, Schematic of the astrocyte-specific *Crym*-BioID2 AAV construct and its corresponding GFP control. Proximal proteins within ~10 nm are biotinylated after the addition of exogenous biotin. **b**, Representative IHC images of striatal astrocytes from mice microinjected with *GfaABC₁D-Crym*-BioID2 AAVs and treated with biotin. The tissue was immunostained with anti-HA antibody (red), S100β (green), or with a fluorophore conjugated streptavidin probe (grey). Right panel shows endogenous μ-crystallin (magenta) co-localized with S100β (green). **c**, Representative images of immunostained mouse striatum injected with astrocyte-specific *Crym*-BioID2 and then treated with biotin for 7 days. Panel shows the immunostaining pattern with S100β as an astrocyte cell marker. **d**, As in **c** but with NeuN as a neuronal cell marker. **e**, Bar graphs depicting the percent of S100β positive or NeuN positive cells with HA expression in a 40x magnification field of view.

Black portion of the bar graphs show percent co-localization. (n = 8 fields of view at 40x magnification from 4 mice). **f**, Western blot analysis of brain unilaterally microinjected with *Crym*-BioID2. Graph depicts the streptavidin signal intensity divided by the β-actin signal intensity for each data point. (n = 4 mice; two-tailed paired t-test). For gel source data, see Supplementary Fig. 1. **g**, Table shows the number of peptides and proteins found in the astrocyte-specific *Crym*-BioID2 proteomics experiments. Each row shows the number of proteins after filtering. **h**, Bar graph shows the relative μ-crystallin protein expression in LFQ from each neuron and astrocyte subcompartment BioID2 proteomics experiment from ref. 48 and the present study with *Crym*-BioID2. **i**, Images of PLA puncta for μ-crystallin and Map2 in tdTomato (tdT) positive astrocytes in a control experiment where the negative PLA probe was omitted. **j**, Same as **i** but for μ-crystallin and Usp9x. Average data are shown as mean ± s.e.m. and all statistics are reported in Supplementary Table 5.

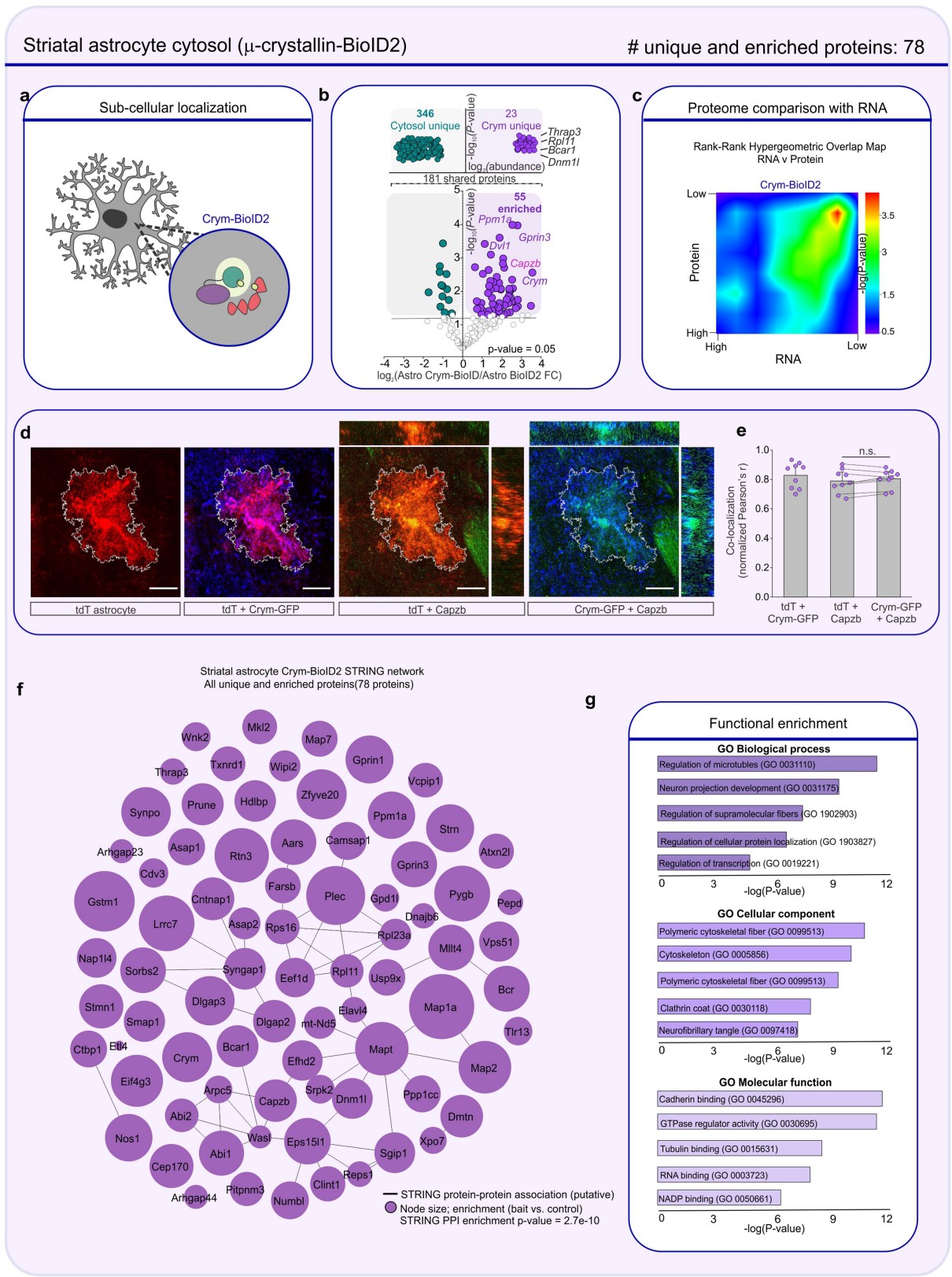

**Extended Data Fig. 14** | See next page for caption.

**Extended Data Fig. 14 | μ-crystallin interactome astrocyte card. a**, BioID2 that is targeted with *Crym*-BioID2 biotinylates proteins that interact with μ-crystallin. **b**, LFQ comparison of significant proteins (Log2FC > 1 and FDR < 0.05 versus GFP controls) detected in the cytosolic Astro BioID2 and *Crym*-BioID2 reveal μ-crystallin enriched proteins. Top half of the volcano plot shows 23 unique *Crym*-BioID2 proteins when compared to cytosol. The top four most abundant proteins for *Crym*-BioID2 are shown. Lower half of volcano plot shows comparison of proteins that were common in both cytosolic BioID2 and *Crym*-BioID2. The five highest enriched proteins for *Crym*-BioID2 are shown. Magenta label shows protein that was validated with immunohistochemistry. **c**, Heat map shows the rank-rank hypergeometric overlap (RRHO) of the RNA and protein rank for the 78 *Crym*-BioID2 proteins. Red pixels represent highly significant overlap. Colour scale denotes the range of P values at the negative log10 scale (Bin size = 100). **d**, IHC analysis of Capzb protein in tdTomato and

*Crym*-GFP labelled astrocytes shows co-localization within the astrocyte territory. Scale bar denotes 20 μm. **e**, Co-localization analysis using Pearson's r coefficient shows co-localization between *Crym*-GFP and Capzb is equivalent to tdT, which is to be expected for these cytosolic proteins. The mean and s.e.m. are shown (n = 8 tdTomato+ cells from 4 mice; Two-tailed paired t-test). **f**, Scale-free STRING analysis protein–protein association map of the 78 unique and enriched biotinylated proteins identified with *Crym*-BioID2. Node size represents the enrichment of each protein vs the GFP control (log2(BioID2/GFP)). Edges represent putative interactions from the STRING database. **g**, Bar graphs show the functional enrichment analysis of all 78 proteins using 'Biological process', "Cellular component", and "Molecular function" terms from Enrichr. Average data are shown as mean ± s.e.m. and all statistics are reported in Supplementary Table 5.

# Reporting Summary

## Statistics

For all statistical analyses, confirm that the following items are present in the figure legend, table legend, main text, or Methods section.

| n/a | Confirmed | |
|---|---|---|
| ☐ | ☒ | The exact sample size (*n*) for each experimental group/condition, given as a discrete number and unit of measurement |
| ☐ | ☒ | A statement on whether measurements were taken from distinct samples or whether the same sample was measured repeatedly |
| ☐ | ☒ | The statistical test(s) used AND whether they are one- or two-sided *Only common tests should be described solely by name; describe more complex techniques in the Methods section.* |
| ☐ | ☒ | A description of all covariates tested |
| ☐ | ☒ | A description of any assumptions or corrections, such as tests of normality and adjustment for multiple comparisons |
| ☐ | ☒ | A full description of the statistical parameters including central tendency (e.g. means) or other basic estimates (e.g. regression coefficient) AND variation (e.g. standard deviation) or associated estimates of uncertainty (e.g. confidence intervals) |
| ☐ | ☒ | For null hypothesis testing, the test statistic (e.g. *F*, *t*, *r*) with confidence intervals, effect sizes, degrees of freedom and *P* value noted *Give P values as exact values whenever suitable.* |
| ☒ | ☐ | For Bayesian analysis, information on the choice of priors and Markov chain Monte Carlo settings |
| ☒ | ☐ | For hierarchical and complex designs, identification of the appropriate level for tests and full reporting of outcomes |
| ☐ | ☒ | Estimates of effect sizes (e.g. Cohen's *d*, Pearson's *r*), indicating how they were calculated |

*Our web collection on statistics for biologists contains articles on many of the points above.*

## Software and code

Policy information about availability of computer code

Data collection: Proteomics: The spectra were collected using data dependent acquisition on Orbitrap Fusion Lumos Tribrid mass spectrometer (Thermo Fisher Scientific) with an MS1 resolution of 120,000 followed by sequential MS2 scans at a resolution of 15,000. Data generated by LC-MS/MS were searched using the Andromeda search engine integrated into the MaxQuant (Cox et al,. 2008) bioinformatic pipelines against the Uniprot Mus musculus reference proteome (UP000000589 9606) and then filtered using a "decoy" database-estimated false discovery rate (FDR) < 1%. Label-free quantification (LFQ) was carried out by integrating the total extracted ion chromatogram (XIC) of peptide precursor ions from the MS1 scan. These LFQ intensity values were used for protein quantification across samples. Label-free quantification was carried out by the MaxQuant software with integrated search engine, Andromeda (https://www.maxquant.org/).
Metabolomics: Derivatized metabolites were analyzed using a DB-35MS column (30m × 0.25mm i.d. × 0.25μm, Agilent J&W Scientific) in an Agilent 7890A gas chromatograph coupled to a 5975C mass spectrometer.
RNA-seq: Sequencing was performed on Illumina HiSeq 4000. Reads were aligned to the mouse mm10 reference genome using the STAR spliced read aligner v2.7.5c (Dobin et al, 2013)
sc-RNA Seq: sc-RNA seq data was processed with the 10X genomics platform. Single cell libraries were generated and sequenced on the Illumina NextSeq500 sequencer.
Behavior: Open field, elevated plus maze, and lickometer testing was collected and analyzed simultanously by Anymaze (v6.3 Stoelting Co. Wooddale, IL, USA).
Calcium imaging: GECIquant sofware (v1.0) was used to analyze calcium signals in microdomains.
Imaging for IHC and RNA-scope was conducted on an Olympus FV3000 confocal microscope using Fluoview software (FV31S-SW, v2.61.243).
Electrophysiological recordings were performed using pCLAMP11.2 (Axon Instruments) using a Multiclamp 700B amplifier (Axon instruments). Analysis for electrophysiological recordings was conducted using ClampFit 10.7 software.
Western blot data was collected on a GE Amersham 680 imager.

Data analysis: Proteomics: Label-free quantification was carried out by the MaxQuant software with integrated search engine, Andromeda (https://www.maxquant.org/). Differential protein expression and enrichment analysis was conducted with Bioconductor R package, limma v 3.54 (https://bioconductor.org/packages/release/bioc/html/limma.html). Protein network visualization, including STRING analysis was conducted

with Cytoscape v 3.8 (https://apps.cytoscape.org/apps/stringapp). The artMS package v 1.16 (https://bioconductor.riken.jp/packages/3.8/bioc/html/artMS.html) was used to re-format the maxquant results (evidence.txt file), to make them compatible with SAINTexpress program. SAINT protein interaction probability scoring was done through (http://saint-apms.sourceforge.net/Main.html).
RNA-seq: Differential gene expression and enrichment analysis used R package limmaVoom v 3.36 to process RNA counts (https://rdrr.io/bioc/limma/man/voom.html).
IHC, RNA-scope, proximity, western blots: Microscopy data and western blot data was imported and analyzed on FIJI (ImageJ v 2.1) using the BioFormats importer for Olympus FV3000 acquired images.
sc-RNA Seq: Sequence reads were processed and aligned to the mouse genome using CellRanger 3.0 (10X Genomics). Processing and visualization were conducted with R-package Seurat (https://CRAN.R-project.org/package=Seurat, Satija Lab)
All data, unless otherwise stated were plotted with OriginPro 2017 (v 9.4.2)
Statistical analysis was conducted with OriginPro 2017 (v 9.4.2)

For manuscripts utilizing custom algorithms or software that are central to the research but not yet described in published literature, software must be made available to editors and reviewers. We strongly encourage code deposition in a community repository (e.g. GitHub). See the Nature Portfolio guidelines for submitting code & software for further information.

## Data

Policy information about availability of data

All manuscripts must include a data availability statement. This statement should provide the following information, where applicable:
- Accession codes, unique identifiers, or web links for publicly available datasets
- A description of any restrictions on data availability
- For clinical datasets or third party data, please ensure that the statement adheres to our policy

The proteomic data are available at PRIDE with accession ID PXD040991. The UniProt reference proteome used was UniProt UP000000589 for Mus musculus. The RNA-seq data are available at GEO with accession ID GSE228506. scRNAseq data are available at GEO with accession ID GSE225741. Proteomic data are provided as Extended data Excel file 3. The analyzed RNA-seq data are provided as Extended data Excel files 1 and 2. Statistical tests for all figures are provided in Extended data Excel file 4.

# Field-specific reporting

Please select the one below that is the best fit for your research. If you are not sure, read the appropriate sections before making your selection.

☒ Life sciences   ☐ Behavioural & social sciences   ☐ Ecological, evolutionary & environmental sciences

For a reference copy of the document with all sections, see nature.com/documents/nr-reporting-summary-flat.pdf

# Life sciences study design

All studies must disclose on these points even when the disclosure is negative.

| | |
|---|---|
| Sample size | Power analysis was conducted using values for power of 0.8 or higher and alpha of 0.1 or lower and an estimated effect size based on pilot data. Furthermore, group sizes were selected based on data from the use of similar models by our laboratory. |
| Data exclusions | No data was excluded from this manuscript |
| Replication | To verify the reproducibility of the experimental findings, all data collection was done in multiple batches comprising at least four replicates. The proteomic data analyses was conducted from 4 independently processed batches that each contained 8 mice  for each experimental group (in all cases). Behavioral data was conducted as the mice became available from the breeding colony and each experiment/recording was done in 2-3 batches containing between 3-6 mice per group. All experiments were successfully replicated. |
| Randomization | For all experiments, the mice were randomly allocated to a group as they became available and of age from the breeding colony in alternation. |
| Blinding | For all analyses, the investigators were blinded to group allocation during data collection, as numerical mouse IDs were the only identifier used. |

# Reporting for specific materials, systems and methods

We require information from authors about some types of materials, experimental systems and methods used in many studies. Here, indicate whether each material, system or method listed is relevant to your study. If you are not sure if a list item applies to your research, read the appropriate section before selecting a response.

## Materials & experimental systems

| n/a | Involved in the study |
|---|---|
| ☐ | ☒ Antibodies |
| ☒ | ☐ Eukaryotic cell lines |
| ☒ | ☐ Palaeontology and archaeology |
| ☐ | ☒ Animals and other organisms |
| ☒ | ☐ Human research participants |
| ☒ | ☐ Clinical data |
| ☒ | ☐ Dual use research of concern |

## Methods

| n/a | Involved in the study |
|---|---|
| ☒ | ☐ ChIP-seq |
| ☒ | ☐ Flow cytometry |
| ☒ | ☐ MRI-based neuroimaging |

## Antibodies

| | |
|---|---|
| Antibodies used | Primaries:<br>mouse anti-μ-crystallin  (Santa Cruz, #sc-376687, clone F-11, lot #E1018)<br>chicken anti-GFP (Abcam #ab13970)<br>mouse anti-NeuN (Millipore #MAB377, clone A60, lot #2884594)<br>guinea pig anti-NeuN (Synaptic system, #266004, lot#G-53)<br>rabbit anti-DARPP32 (Abcam, #ab40801, clone EP720Y, lot #1007414-1)<br>rabbit anti-S100ß (Abcam #ab41548)<br>rabbit anti-cFos (Synaptic system #226008, lot #108B5)<br>rabbit anti-RFP (Rockland #600–401-379, lot #48710)<br>rabbit anti-mCherry (Abcam, #ab167453)<br>rabbit anti-Opioid Receptor μ, pain (MOR1) (Millipore-sigma, #AB5511, lot #1007414-1)<br>chicken anti-Calbindin D-28K (Novus biologicals, #NBP2-50028)<br>chicken anti-GFAP (Abcam, #ab4674, lot #1012209-2)<br>rabbit anti-HA tag (Abcam, #ab9110, #)<br>mouse anti-HA tag (Biolegend, #901514)<br>rabbit anti-β-actin (Abcam, #ab8227)<br>rabbit anti-USP9X (Proteintech, #55054-1-AP, lot #00043048)<br>rabbit anti-MAP2 (Thermofisher scientific, #PA1-10005, lot #YG3995721)<br>guinea pig anti-RFP (Synaptic Systems, #390004)<br>rabbit anti-Sox9 (EMD Millipore, #AB5535, lot #3836442)<br>rabbit anti-Olig2 (EMD Millipore #AB9610, lot #3814881))<br>rabbit anti-Kir4.1 (Alomone labs, #APC-035, lot #AN1002)<br>rabbit anti-ATP1a2 (Proteintech, #16836-1-AP)<br>rabbit anti-GLT1 (Synaptic Systems, #250203, lot #1-10)<br>mouse anti-GABA (abcam, #ab86163, lot #GR3423846-5)<br>rabbit anti-MAOB (Thermofisher #PA5-28338, lot #YH4009980A)<br>rabbit anti-GAT3 (gift from the Brecha lab, UCLA)<br>rabbit anti-Capzb (Thermo Fisher, #PA5-83196)<br><br>Secondaries:<br>Goat anti-rabbit plus 647 (Invitrogen, A32733, lot #VC299350)<br>Alexa Fluor 405 goat anti-mouse (A31553, lot #2491371)<br>Alexa Fluor 488 goat anti-chicken (A11039, lot #2566343)<br>Alexa Fluor 488 goat anti-rabbit( A11008, lot #2420730))<br>Alexa Fluor 488 goat anti-mouse (A11001, lot #2610355)<br>Alexa Fluor 546 goat anti-mouse (A11030, lot #2155294)<br>Alexa Fluor 546 goat anti-rabbit (A11010, lot #2570547)<br>Alexa Fluor 647 goat anti-rabbit (A21244, lot #2497486)<br>Streptavidin, Alexa Fluor 488 conjugate (S11223, lot #18585036)<br>Streptavidin-HRP (Sigma, RABHRP3)<br>Donkey anti-guinea pig Cy3 (Jackson ImmunoResearch, #706-165-148, lot #159084)<br>IR-dye 800CW anti-rabbit (Li-Cor, #925-32211, lot #C80118-01) |
| Validation | The antibodies used in this manuscript have been validated and reproduced by our lab across at least 7 manuscripts by checking cell specificity, background signal, and noting antigen specificity using western blot techniques (Srinivasan et al., 2016; Chai et al., 2017, Nagai et al., 2019; Yu et al., 2020; Diaz-Castro et al., 2019; Endo et al., 2022, Gangwani et al., 2023). All Khakh lab manuscripts. |

## Animals and other organisms

Policy information about studies involving animals; ARRIVE guidelines recommended for reporting animal research

| | |
|---|---|
| Laboratory animals | All animal experiments were conducted in accordance with the National Institutes of Health Guide for the Care and Use of Laboratory Animals and were approved by the Chancellor's Animal Research Committee at the University of California, Los Angeles. All mice were housed with food and water available ad libitum in a 12-hour light/dark environment. All animals were healthy with no obvious |

behavioral phenotype, were not involved in previous studies, and were sacrificed during the light cycle. Data for experiments were collected from adult mice aged 9-15 weeks old. To characterize u-crystallin expression during development and ageing, mice were used between P0 and 22 months old. C57Bl/6NTac mice were maintained as an in-house breeding colony or purchased from Taconic Biosciences. CAG-Cas9 transgenic mice (B6J.129(Cg)-Gt(ROSA)26Sortm1.1(CAG-cas9*,-EGFP)Fezh/J, JAX Stock # 026179) and SAPAP3 -/- mice (B6.129-Dlgap3tm1Gfng/J, JAX Stock # 008733) were purchased from the Jackson Laboratory and maintained as breeding colonies at UCLA. SAPAP3 -/- mice were used at 6 months of age. Tg(Crym-EGFP)GF82Gsat(strain 012003-UCD) reporter mice were obtained from MMRRC and maintained as a breeding colony at UCLA. Both males and females were used in alternating batches.

| Wild animals | The study did not involve wild animals |
| Field-collected samples | This study did not use field-collected samples |
| Ethics oversight | All experiments were conducted in accordance with the National Institutes of Health (NIH) Guide for the Care and Use of Laboratory Animals and were approved and overseen by the Chancellor's Animal Research Committee (ARC) at the University of California, Los Angeles (UCLA) |

Note that full information on the approval of the study protocol must also be provided in the manuscript.

