## [Peer Review File · Nature]

Manuscript Title: Crym-positive striatal astrocytes gate perseverative behavior

Reviewer Comments & Author Rebuttals

Reviewer Reports on the Initial Version:

Referees' comments:

Referee #1 (Remarks to the Author):

In this manuscript by Dr. Khakh and colleagues the role of Crym (crystallin) in a subset of striatum astrocytes is evaluated. They first show that Crym is expressed in a subset of astrocytes in the striatum and that its expression is decreased in human patients with OCD. Next, they use CRISPR/Cas9 approaches to KO Crym in striatal astrocytes and show a series of behavioral deficits that are endophenotypes of OCD. At the physiological level they demonstrate that reduced GABA levels in Crym-KO mice results in increased glutamatergic synaptic activity from IOFC projections into MSNs. Strikingly, suppressing IOFC activity in the Crym-KO rescues the OCD-associate behaviors. Finally, they survey the Crym proteome and identify a pathways and proteins that serve as a prospective functional mechanism.

Overall, this is an impressive and consequential paper on the role of astrocytes in functioning brain circuits. They identify a new gene that has clear roles in astrocyte regulation of circuit function and use it to identify a new circuit-based mechanism for how astrocytes regulate behaviors, which could have important implications for humans with OCD. This part of the paper further reinforces new thinking in the neurosciences that astrocyte dysregulation is a key driver of psychiatric and neurodevelopmental diseases. Furthermore, I will add that the circuit based rescue experiment in figure 4 is stunning. Technically very demanding, conceptually sound, and extremely well executed by the authors---they are to be commended for this level of rigor. Despite these obvious strengths, there are a few relatively minor weaknesses that I hope the authors will address.

1) Broadly, its very difficult to say that the crystallin-expressing astrocytes have a unique function, when there is no comparison with crystallin-negative astrocytes. To formally do so would require the generation of mouse tools that enable manipulation of crystallin-positive and crystallin-negative astrocytes, followed by a systematic comparison of how these population perturbations influence striatum circuits and astrocyte phenotypes. This is beyond the scope of this paper and not something I would ask the authors to do. What the authors have actually done is manipulate a gene that is present in ~50% of striatal astrocytes and show that it plays an essential role in circuit function, specifically in the striatum---on its own, this is an important and consequential finding. Accordingly, I would suggest that they emphasize this aspect and dial back some of the language about "subsets of astrocytes" and subpopulations as this paper does not directly or formally address this topic.

2) Figure 1, staining with s100b is not totally convincing to this reviewer as this marker is also in oligos. Maybe also try co-staining with a nuclear marker of astrocytes like NFIA or Sox9 or Sox2. Maybe also try a co-stain with Olig2 or Sox10 to rule out the Oligo lineage.

3) Overall, the link to OCD and HD is a bit oversold---I understand why its included, but there is no validation in human samples (understood that this is very difficult). My suggestion is to omit this sentence from the abstract. The decrease in expression of crystallin in OCD patients is a nice rationale for the behavioral studies, but there needs to be a firmer link to include in the abstract. Also—and very importantly---the crystallin-KO mice do not have bona fide OCD phenotypes (figure 2 and lines 144-145), but rather an endophenotype of OCD. They offer an explanation at the end of this section, but I remain unsure of how the human data fits in with this. To provide the reader more clarity, I think a clearer explanation is needed here.

4) In Figure 2, do Crystallin-expressing astrocytes have any unique core properties---morphological complexity, Ca²⁺ activity, expression of core markers---when compared with crystallin-negative astrocytes. Given that they have the Crystallin-GFP mouse, it seems possible to include some basic analysis of these features to compare these populations. Along, these same lines, can they FACS isolate these crystallin-GFP astrocytes and run RNA-Seq to see how different these astrocytes are from other striatum astrocytes? This may require some complex breeding of the Aldh1l1-CreER with the tdTomato reporter and the Crystallin-GFP mouse, so its not a requirement---but if possible, it would be a very nice experiment that might help further distinguish these subsets of astrocytes (also see comment 1).

5) In Figure 3, the anatomical tracing to the IOFC is very nice, but it seems like the projections between the striatum and the cortex are essentially unaffected. The only evidence (based on these anatomical studies) for a disruption in communication seems to be the elevated cFos in the IOFC. Not sure why the tracing data is necessary and what they add to the story, other than to say that the macrocircuit projections remain intact in the absence of crystallin. Seems like the physiology studies were guided by the cFos staining and that's what is really important for the first part of figure 3. Perhaps either move the tracing to supplement or provide a clearer explanation for their inclusion.

6) The electrophysiology is sound, however their explanation needs to be clarified. Based on the compendium of data, they conclude that reduced tonic GABA in the striatum results in increased activity of presynaptic IOFC projections on to MSNs. While I accept this explanation, I'm left wondering why there is more tonic GABA. Astrocytes can release GABA, so I'm left wondering whether crystallin-KO astrocytes have defective GABA production or release? Obviously, this is a complex mechanism, but some speculation in the discussion or even some simple staining experiments with MOAB or GABA in astrocytes to address the source of the increase in tonic GABA would be a welcome addition to this part of the story.

7) In figure 3J, it would help the reader if the diagram distinguished which neurons are from the IOFC and which are the MSN's.

8) Figure 5 is a very nice survey of the crystallin proteome, but I'm still left wondering about the cellular function of this protein. The links to OCD are fine, but there is limited insight into how crystallin is influencing astrocyte function at the cellular level. There are two simple ways to remedy this. First, validation of some of the interacting proteins would give more confidence in the prospective functions suggested by the proteomics. Second, it would be nice (see comments linked to figure 2) to know more about the core properties of crystallin-KO astrocytes. Admittedly,

astrocytes are very good at keeping their secrets and there are limited number of assays (morphology, Ca²⁺, markers, etc.), but understanding how the cellular phenotypes are linked to this impressive proteomic analysis would be important.

Referee #2 (Remarks to the Author):

In this manuscript, Ollivier et al. detail the discovery of an anatomically distinct subpopulation of astroglia, defined by their expression of μ -crystallin, a relatively unknown and unstudied marker. The authors report an important role for μ -crystallin expression in human obsessive-compulsive disorder and Huntington's disease, based on its downregulation in post-mortem human brain samples. They nicely combine electrophysiology, chemogenetics, and behavior in mice lacking Crym in central striatal astrocytes to characterize the role of Crym⁺ astroglia in neural regulation and behavior. The authors find remarkable synaptic regulation by Crym⁺ astroglia, such that knockdown of Crym expression leads to an increased E/I ratio of transmission into the central striatum. Mice lacking central striatal Crym expression in astroglia exhibit unique behavioral deficits that appear to reflect repetitive, perseverative behaviors, including grooming, spout licking, marble burying, etc. without changes in anxiety or motor defects. Thus, their findings not only reflect a critical role for astroglial μ -crystallin in synaptic regulation, but also distinguish perseveration as a unique and distinct behavioral domain, independent from anxiety, and identify the circuitry that produces these behaviors. In all, this work is careful and complete; a comprehensive study that has produced clear outcomes with far-reaching implications. The statistical analyses applied are appropriate and clearly described. This study is one of few that clearly identifies a fundamental physiological role for striatal astroglia in modulating behaviors that are disrupted in a number of psychiatric and neurological disorders. This study also highlights the value gained from detailed assessment of astrocyte subpopulations tuned to regulate specific neural subcircuits. Notably, the behavioral deficits following from Crym deletion in this population of glia is not corrected by fluoxetine, highlighting a relevant disease aspect that remains untreated by classic pharmacotherapies. Below are outlined a few points that, if discussed in greater detail, might further highlight the importance of these findings.

- 1) The authors show that Crym expression changes over the course of postnatal development. What is known regarding changes in Crym expression with age/senescence?
- 2) What is the relative decrease in Crym expression in HD compared with OCD? In what brain regions was decreased Crym observed in humans?
- 3) Does the dichotomy between astroglial and neuronal expression of Crym exist in humans as well? Are there data to demonstrate whether the decrease in Crym in HD and OCD occur in astroglia, neurons, or both?
- 4) Does Crym expression (in astroglia or neurons) appear to be sexually dimorphic? Is there any relationship with disease severity in the human samples?

Minor

In all figures, i and I panel denotations cannot be distinguished.

Referee #3 (Remarks to the Author):

A+B

Key results: Please summarise what you consider to be the outstanding features of the work.

There are four outstanding features in this study:

1) This very complete work shades new lights on the role of a protein discovered decades ago. While founding biochemical studies showed that Crym has probable key roles in regulating thyroid hormones effects, the true novelty here is that the study demonstrates that Crym has a preferential expression profile in astrocytes in the striatum, and that through this preferential localization in these cells (and not neurons), Crym controls synaptic plasticity in this brain region.

2) The present study also brings another new concept linked to key trait of animal and human brain function, flexibility/inflexibility. Mind flexibility is central in adaptive behavior to environment /external stimulus in animal and human. The complex processes and molecular mechanisms underlying flexibility or inflexibility in pathological conditions are not known and have been often related to synaptic plasticity, involving mainly “neuronal” circuits”. Here, the authors show that a particular subset of astrocytes (expressing high levels of Crym) can regulate flexibility, which is really puzzling. This is probably one of the most significant scientific demonstration that molecular heterogeneity in astrocytes plays a direct role on complex brain functions (as considered for neurons). This likely will pave the way to future investigations on the role of molecular heterogeneity astrocytes of other brain regions/circuits and functions, such as procedural and spatial memory, decision making, etc.

3) The methodological approaches of the authors to come to this conclusion is mainly based on robust results from experiments in genetically modified mice and/or gene-transfer experiments with viral vectors for opto- and chemo-genetics. In addition, they provide correlational data using post mortem human brain samples from patients with well-recognized inflexibility traits showing that at least in part inflexibility/perseveration in these patients- could be due to loss of Crym in this subset of astrocytes

4) One major consequence of the present findings on a subset of astrocytes is that it also opens up new avenues in terms of therapeutic targets, not only for inflexibility-related pathological behaviors (such as seen in patients with Huntington’s diseases), but beyond, for other brain disorders resulting from the dysfunction of other subsets of astrocytes, yet to be discovered.

In summary, the present work will be founding to a large spectrum of future studies on the role of subsets of astrocytes in brain health.

Originality and significance:

As mentioned, in the section/question above, I believe that this work is outstanding and of interest for all researchers in the field of neuroscience, neurology and psychiatry.

C- Validity: Does the manuscript have flaws which should prohibit its publication? If so, please

provide details.

Not that I could detect.

Data & methodology: Please comment on the validity of the approach, quality of the data and quality of presentation. Please note that we expect our reviewers to review all data, including any extended data and supplementary information. Is the reporting of data and methodology sufficiently detailed and transparent to enable reproducing the results?

C - The methodological approaches and the experimental design are robust. Main figures show an impressive set of data with appropriate controls. In addition, supplemental figures/data that are provided by the authors contains control experiments that significantly strengthen the results displayed in the main figures.

The figures are of high quality in terms of presentation (organization, quality of drawing, image resolution, statistics etc.).

Regarding the description of methods/approaches, it is fair to say that the Nature format does not permit an in depth description of the methods so that usually, the reader has to go back to previous publications by the authors or those from other research groups. From my point of view, for most aspects, there are sufficient details. For example, while going through the manuscript and reading the results the referee had concerns about the selective knockout of Crym in astrocytes. Regarding the exact description of AAVs used to locally delete Crym. Everything useful could be found in the M&M (promoter, backbone, construct pseudotypes etc.), addressing the referee questions on this precise point.

Minor point: a scale bar legend should be included in all figures displaying histological images.

D - Appropriate use of statistics and treatment of uncertainties: All error bars should be defined in the corresponding figure legends; please comment if that's not the case. Please include in your report a specific comment on the appropriateness of any statistical tests, and the accuracy of the description of any error bars and probability values.

The statistics were made in accordance to general guidelines to analyze biological data. Non parametric tests have been used for datasets with a distribution not satisfying normal distribution. When distribution of data was considered normal, parametric tests (student t test, one-way or two way factorial ANOVA have been applied with a statistical levels of 5%, which is the general rule. In many case, statistical levels are satisfying. This is clearly explained in materials and methods section and in figures. Graphs consistently show the mean +/-error bars corresponding to standard error of the mean (SEM). For biostatistics, using available tools (such String for protein interaction and enrichment analysis, as well as Enrichr), the authors described statistical values (FDR, False Discovery Rates) that are robust, i.e. $p < 0.05$)

E - Conclusions: Do you find that the conclusions and data interpretation are robust, valid and reliable?

YES, absolutely.

F - Suggested improvements: Please list additional experiments or data that could help strengthening the work in a revision.

There are two point that might be worth pursuing or at least clarify.

1. The exact mechanism through which alteration of Crym expression in the subset of astrocytes in the medial striatum leads to perseveration remains not yet totally understood. The present data indicate that it involves Crym⁺ astrocyte-mediated presynaptic glutamatergic terminals from the cerebral cortex (in particular the lateral orbitofrontal cortex). What is the molecular /chemical mediator of this effect? Is it only via modulation of glutamate release? If it is via modulation of glutamate release, what could cell signaling or cell-cell (i.e. astrocyte to neuron) mechanism(s) be at stake. Biochemical/cell experiments demonstrate that Crym is a thyroid T3 binding protein and transports/carries the hormone to the nucleus to activate gene expression. Crym also possesses an additional enzymatic activity, ketimine reductase (E.C. 1.5.1.25). Could this reductase activity of Crym (and thus the effects/roles of cyclic ketamine and catabolites) be involved in astrocyte- neuron functional interactions? In line with this, the authors performed GC/MS analysis of striatal samples from Crym KO and control mice. Do they have data related to ketimine pathway? Measuring ketimine metabolites (precursor/breakdown products) might be an interesting aspect of the effects of Crym KO.

2. The interactors of Crym identified in the present work often have known preferential expression in astrocytes. However, the presence of MAPT (coding the protein Tau) which is considered to be more expressed in neurone than in astrocytes might appear a little bit confusing at first. The presence of MAPT in astrocyte interactome is also described in the paper by the same authors (Soto et al., *Nature*. 2023; 616(7958): 764–773). However, this is particularly interesting to consider that the interactome of Crym includes such a protein. The authors might comment or highlight more explicitly that the interactome described using the method used does not necessarily reflect the stoichiometry of proteins in one cell type but also (probably) the strength of interaction in subcompartments. The interaction between two proteins might also depend on the state of the interaction of others partners, since “binding” competition might exist between different partners.

3. The present data convincingly demonstrate that the levels of Crym in a subset of astrocytes is much higher than levels seen in neurons? However transcriptomic studies in the past showed that striatal neurons also express Crym (at a lower degree) (Heinman M et al *Cell* 2018, 135(4):738-48; see supplementary table S5). In line with this, increasing Crym levels in MSN in the dorsal striatum is neuroprotective against mutant huntingtin (Francelle et al., *Hum Mol Genet* . 2015, 24(6):1563-73). Thus, it might be explicitly mentioned in the manuscript that Crym might also play an important role in neurons, although its level of expression in neuronal cells is much lower than that in astrocytes.

G. References: Does this manuscript reference previous literature appropriately? If not, what references should be included or excluded?

See point # 3 above:

Heinman M et al. *Cell* 2018, 135(4):738-48).

Francelle et al. *Hum Mol Genet* 2015, 24(6):1563-73.

H - Clarity and context: Is the abstract clear, accessible? Are abstract, introduction and conclusions appropriate?

Yes, abstract is clear, accessible. Interpretation of results is fair so that conclusion are appropriate

Author Rebuttals to Initial Comments:

Referee #1:

“In this manuscript by Dr. Khakh and colleagues the role of Crym (crystallin) in a subset of striatum astrocytes is evaluated. They first show that Crym is expressed in a subset of astrocytes in the striatum and that its expression is decreased in human patients with OCD. Next, they use CRISPR/Cas9 approaches to KO Crym in striatal astrocytes and show a series of behavioral deficits that are endophenotypes of OCD. At the physiological level they demonstrate that reduced GABA levels in Crym-KO mice results in increased glutamatergic synaptic activity from IOFC projections into MSNs. Strikingly, suppressing IOFC activity in the Crym-KO rescues the OCD-associate behaviors. Finally, they survey the Crym proteome and identify a pathways and proteins that serve as a prospective functional mechanism.

Overall, this is an impressive and consequential paper on the role of astrocytes in functioning brain circuits. They identify a new gene that has clear roles in astrocyte regulation of circuit function and use it to identify a new circuit-based mechanism for how astrocytes regulate behaviors, which could have important implications for humans with OCD. This part of the paper further reinforces new thinking in the neurosciences that astrocyte dysregulation is a key driver of psychiatric and neurodevelopmental diseases. Furthermore, I will add that the circuit based rescue experiment in figure 4 is stunning. Technically very demanding, conceptually sound, and extremely well executed by

the authors---they are to be commended for this level of rigor. Despite these obvious strengths, there are a few relatively minor weaknesses that I hope the authors will address.”

Thank you for these supportive comments and for the detailed review of our manuscript. All your additional points are addressed in turn below.

1) *“Broadly, its very difficult to say that the crystallin-expressing astrocytes have a unique function, when there is no comparison with crystallin-negative astrocytes. To formally do so would require the generation of mouse tools that enable manipulation of crystallin-positive and crystallin-negative astrocytes, followed by a systematic comparison of how these population perturbations influence striatum circuits and astrocyte phenotypes. This is beyond the scope of this paper and not something I would ask the authors to do. What the authors have actually done is manipulate a gene that is present in ~50% of striatal astrocytes and show that it plays an essential role in circuit function, specifically in the striatum---on its own, this is an important and consequential finding. Accordingly, I would suggest that they emphasize this aspect and dial back some of the language about “subsets of astrocytes” and subpopulations as this paper does not directly or formally address this topic.”*

Thank you. We have now dialed back the language regarding “subsets of astrocytes” throughout the revised manuscript, as suggested. We dropped the word “subset” and have also changed the title and abstract to reflect this guidance. We did however need to use a noun to refer to the *Crym*+ astrocytes. We now refer to them as the “*Crym*+ astrocyte population” or simply as “*Crym*+ astrocytes”. We hope the reviewer will find the new presentation satisfactory. If they have a preferred noun for us to use, we will happily consider it.

Please also note that in relation to comment 4 below, we have now provided substantial more data reporting the properties of *Crym*+ and *Crym*- astrocytes.

2) *“Figure 1, staining with *s100b* is not totally convincing to this reviewer as this maker is also in oligos. Maybe also try co-staining with a nuclear marker of astrocytes like *NFIA* or *Sox9* or *Sox2*. Maybe also try a co-stain with *Olig2* or *Sox10* to rule out the Oligo lineage.”*

Thank you. We now include *Sox9* and *Olig2* immunohistochemistry. In reporting these new data, on page 2 we write

*“In accord, using immunohistochemistry (IHC) we found that μ -crystallin expressing astrocytes within the striatum represented ~49% of the total and were precisely anatomically located, being essentially absent in the dorsolateral regions and enriched ventrally and in the central region (Fig. 1e). There, μ -crystallin was expressed in ~90% of *S100 β* + and ~90% of *Sox9*+ astrocytes, but in no *NeuN*+ neurons (Fig. 1f-h) or *Olig2*+ oligodendrocytes (Extended data Fig 1).”*

3) *“Overall, the link to OCD and HD is a bit oversold---I understand why its included, but there is no validation in human samples (understood that this is very difficult). My suggestion is to omit this sentence from the abstract. The decrease in expression of crystallin in OCD patients is a nice rationale for the behavioral studies, but there needs to be a firmer link to include in the abstract. Also---and very importantly---the crystallin-KO mice do not have bona fide OCD phenotypes (figure 2 and lines 144-145), but rather an endophenotype of OCD. They offer an explanation at the end of this section, but I*

remain unsure of how the human data fits in with this. To provide the reader more clarity, I think a clearer explanation is needed here.”

Thank you. We have now revised the abstract as suggested and provided greater explanation of the connection to OCD and HD throughout. In particular, on page 2 we now write

*“Since striatal astrocytes express OCD and HD-related genes^{2,36}, we evaluated expression of the top 20-striatal astrocyte enriched genes within postmortem striatal tissue data for OCD and HD¹⁷⁻¹⁹. Of these, *Crym* was similarly downregulated in caudate of OCD and HD¹⁷⁻¹⁹ to about 40% of control values and was within the top 4% of the downregulated genes, including in HD mouse models^{19,37,38}, implying potentially important functions of *Crym*+ astrocytes¹⁶ (Fig. 1a).”*

Later on page 9, we write the following section, which also captures changes requested by Reviewer #2.

*“Clinically, perseveration represents inappropriate continuation or repetition of a response or activity and is associated with psychiatric and neurological disorders such as Tourette’s syndrome, Autism, OCD, HD, and suicide-associated perseveration in HD²¹⁻²⁴. Our data reveal that μ -crystallin loss leads to perseveration, which is of relevance to HD and OCD where *Crym* is reduced in postmortem human striatal tissue¹⁷⁻¹⁹. In the case of HD for which several gene expression studies are available, *Crym* downregulation increases with disease severity based on RNAseq of human caudate¹⁸ and striatal HD mouse model tissue³⁸. Similar to our findings, *Crym* was expressed in greater numbers of astrocytes than neurons in human tissue³¹, and it decreased in both cell types in postmortem human HD samples¹⁹ and in HD mouse models^{19,37}. In these regards, by exploring astrocytes, our mechanistic findings show that elevated neurotransmitter release probability of IOFC terminals is regulated by striatal *Crym*+ astrocytes in a manner that is causal for perseveration phenotypes that accompany OCD and HD.”*

*4) “In Figure 2, do Crystallin-expressing astrocytes have any unique core properties---morphological complexity, Ca2+ activity, expression of core markers---when compared with crystallin-negative astrocytes. Given that they have the Crystallin-GFP mouse, it seems possible to include some basic analysis of these features to compare these populations. Along, these same lines, can they FACS isolate these crystallin-GFP astrocytes and run RNA-Seq to see how different these astrocytes are from other striatum astrocytes? This may require some complex breeding of the *Aldh111-CreER* with the *tdTomato* reporter and the *Crystallin-GFP* mouse, so its not a requirement---but if possible, it would be a very nice experiment that might help further distinguish these subsets of astrocytes (also see comment 1).”*

Thank you; we address your suggestion below, but explain why we cannot use the *Crym*-EGFP mice for this. The *Crym*-EGFP reporter is a BAC transgenic mouse, not a knock-in at the *Crym* locus. In BAC transgenics, there are often genomic insertional effects despite the large size of the engineered fragment (~250 kb). In this case, our data show that every GFP positive astrocyte is μ -crystallin and S100 β positive (Extended data Fig 6a,b). However, not every μ -crystallin-positive cell is GFP positive in the central striatum (as shown in Extended data Fig 6b). We interpret this result to indicate that the presence of GFP shows faithfully *Crym* positive astrocytes in the central striatum, but that the lack of GFP is not interpretable (as is often the case with a lack of signal). Because of this, the *Crym*-EGFP mice cannot be used in the way the

reviewer wrote, as the lack of GFP is not interpretable with regards to μ -crystallin. Instead, we address the suggestion of reporting differences between *Crym*⁺ and *Crym*⁻ astrocytes in a different and more reliable way.

Thus, we include a new section and figure comparing *Crym*⁺ and *Crym*⁻ astrocytes using a range of approaches. Notably, we found that the mechanism of direct relevance to *Crym*⁺ astrocytes and perseveration is GABA homeostasis mediated by GAT3. On pages 6-7, we write

“Properties of *Crym*⁺ and *Crym*⁻ striatal astrocytes

We compared *Crym*⁺ and *Crym*⁻ astrocytes from scRNAseq (Fig. 5a) to shed light on their molecular properties. We found several established astrocyte markers were equivalently expressed within *Crym*⁺ and *Crym*⁻ astrocytes (Fig. 5b), implying that basic astrocytic functions are likely similar. Interestingly, however, recalling our physiological studies in Fig 3, expression of *Slc6a11* (GAT3) was higher in *Crym*⁺ astrocytes (Fig. 5b,c). This was part of the differentially expressed genes (DEGs) between *Crym*⁺ and *Crym*⁻ astrocytes used for gene pathway analysis (Fig. 5c). We thus compared 178 genes enriched or depleted in *Crym*⁺ astrocytes to the top 200 genes shared with other striatal astrocytes and identified shared and unique pathways for *Crym*⁺ astrocytes, including those for neurovascular coupling and insulin growth factor (IGF) signaling, for example (Fig. 5d). Extended data Fig. 17a reports additional analyses for enriched/depleted genes from *Crym*⁺ cells and 2519 shared genes, supporting shared and separable functions between *Crym*⁺ and *Crym*⁻ astrocytes.

We next extended our evaluations with a specific set of studies to document differences between *Crym*⁺ astrocytes located in the central striatum and *Crym*⁻ astrocytes from the dorsolateral striatum (Fig. 1e-g). For these *Crym*⁺ and *Crym*⁻ astrocytes, we systematically compared: (i) μ -crystallin expression; (ii) astrocyte somata and territory areas (μm^2); (iii) resting membrane potentials (mV); (iv) membrane resistances (M Ω); (v) slope conductances (nS); (vi) frequency and amplitude of somatic spontaneous Ca^{2+} signals; (vii) frequency and amplitude of spontaneous Ca^{2+} signals in astrocyte territories; (viii) amplitude of phenylephrine-evoked Ca^{2+} signals; (ix) GAT3 expression; (x) tonic GABA currents from local MSNs (pA); and (xi) GAT3 dependent tonic GABA currents from local MSNs and thus the ability of *Crym*⁺ and *Crym*⁻ astrocytes to either reduce or contribute to tonic extracellular GABA levels. These data are reported in Fig. 5e-n, Extended data Fig. 18, and summarized with a heat map in Fig. 5o. In brief, in accord with RNA-seq showing that several astrocyte markers were expressed equally, we found that the core properties of *Crym*⁺ and *Crym*⁻ astrocytes were similar. We found only subtle differences in somatic Ca^{2+} signaling (Fig. 5g,i,o), which are reminiscent of pathway analyses suggesting differences in Ca^{2+} transport between *Crym*⁺ and *Crym*⁻ astrocytes (Extended data Fig. 17a). We note that altered astrocyte Ca^{2+} signaling is often explored in the regulation of neural circuits⁶². However, in line with recent ideas^{28,63,64} our studies of *Crym*⁺ and *Crym*⁻ astrocytes, as well as of *Crym* KO astrocytes, suggest μ -crystallin works independently of causal roles for Ca^{2+} in the metrics evaluated in relation to perseveration. Furthermore, consistent with scRNA-seq and functional evaluations in *Crym* KO mice reported in earlier sections, *Crym*⁺ astrocytes in the central striatum displayed higher GAT3 expression and contributed GABA to the extracellular space, whereas those within the dorsolateral region expressed lower GAT3 levels and removed GABA (Fig. 5l-o). In accord, GAT3 is known to remove or contribute GABA to the extracellular space^{56,57,65}. Together, these studies indicate *Crym*⁺ and *Crym*⁻ astrocytes perform shared and separable functions, providing a basis for future detailed studies (Fig. 5). The mechanism of direct relevance to *Crym*⁺ astrocytes and perseveration is GABA homeostasis mediated by GAT3.”

5) *“In Figure 3, the anatomical tracing to the IOFC is very nice, but it seems like the projections between the striatum and the cortex are essentially unaffected. The only evidence (based on these anatomical studies) for a disruption in communication seems to be the elevated cFos in the IOFC. Not sure why the tracing data is necessary and what they add to the story, other than to say that the macrocircuit projections remain intact in the absence of crystallin. Seems like the physiology studies were guided by the cFos staining and that’s what is really important for the first part of figure 3. Perhaps either move the tracing to supplement or provide a clearer explanation for their inclusion.”*

As suggested, we moved the anatomical tracing to the Extended data Fig 12.

6) *“The electrophysiology is sound, however their explanation needs to be clarified. Based on the compendium of data, they conclude that reduced tonic GABA in the striatum results in increased activity of presynaptic IOFC projections on to MSNs. While I accept this explanation, I’m left wondering why there is more tonic GABA. Astrocytes can release GABA, so I’m left wondering whether crystallin-KO astrocytes have defective GABA production or release? Obviously, this is a complex mechanism, but some speculation in the discussion or even some simple staining experiments with MOAB or GABA in astrocytes to address the source of the increase in tonic GABA would be a welcome addition to this part of the story.”*

Thank you. This was an excellent suggestion, which we addressed on page 5 with the following section and new data.

*“During electrophysiological recordings, we noticed significantly decreased MSN tonic GABA currents in *Crym* KO mice, which indicates lower extracellular GABA levels in the central striatum (Fig. 3i,j). Furthermore, tonic GABA currents were blocked in control mice by pre-exposure to the astrocytic⁵⁵ GABA transporter type 3 (GAT3) antagonist⁵⁶ SNAP-5114 (Fig. 3i-k; 40 μ M). However, the reduced tonic GABA currents observed in *Crym* KO mice were spared (Fig. 3i-k), indicating that astrocytic GAT3 within the central striatum contributes GABA to the extracellular space^{56,57}, and that such contributions are reduced in *Crym* KO mice (Fig. 3i-k). Although there were no changes in GAT3 expression within astrocyte territories of *Crym* KO relative to controls (Extended data Fig. 15 a,b), we detected significant reduction in GABA and in monoamine oxidase B expression (MAOB; Extended data Fig. 15c-f). MAOB is an astrocytic enzyme⁵⁸⁻⁶⁰ that generates GABA, implying that reduced tonic GABA levels in *Crym* KO mice reflect reduced GAT3-dependent GABA contribution to the extracellular space as well as reduced astrocytic GABA.”*

7) *“In figure 3J, it would help the reader if the diagram distinguished which neurons are from the IOFC and which are the MSN’s.”*

We suspect the Reviewer wanted clarifications to the previous Figure 3K, which we now provide in the revised Figure 3l by labelling the IOFC terminals and dendritic spines of MSNs in the diagram appropriately.

8) *“Figure 5 is a very nice survey of the crystallin proteome, but I’m still left wondering about the cellular function of this protein. The links to OCD are fine, but there is limited insight into how crystallin is influencing astrocyte function at the cellular level. There are two simple ways to remedy*

this. First, validation of some of the interacting proteins would give more confidence in the prospective functions suggested by the proteomics. Second, it would be nice (see comments linked to figure 2) to know more about the core properties of crystallin-KO astrocytes. Admittedly, astrocytes are very good at keeping their secrets and there are limited number of assays (morphology, Ca²⁺, markers, etc.), but understanding how the cellular phenotypes are linked to this impressive proteomic analysis would be important.”

Thank you. In terms of validations, the revised manuscript contains the requested experiments, which are reported on page 8 and in the appropriate figures with the following sentences.

“Two μ -crystallin interactors identified by proteomics were validated within striatal tissue using the proximity ligation assay⁶⁵ (PLA; Fig. e-g; Extended data Fig. 19g,h).”

In terms of the properties of *Crym* KO astrocytes, on page 4 we include a new section and figures reporting these data.

*“We found no evidence of apoptosis or of neuron or astrocyte loss in the striatum of *Crym* KO mice (Extended data Fig. 9a-h). Furthermore, the morphology and electrophysiology of astrocytes were essentially normal (Extended data Fig. 10a,b,g). We detected only subtle changes in Ca²⁺ signaling within astrocyte somata and territories (Extended data Fig. 10d-f,g), and there was no change in the expression of several striatal astrocytic marker proteins^{1,41,44} in *Crym* KO mice relative to controls (S100 β , Kir4.1, ATP1a2, GFAP, Glt-1; Extended data Fig. 11a,b). In the absence of notable astrocyte alterations, we considered if *Crym*+ astrocytes exert effects on neuronal function.”*

We also refer the Reviewer to our detailed experiments related to comment 4 that explore *Crym*+ and *Crym*- astrocytes directly using multiple approaches. In brief, our studies of *Crym* KO and of *Crym*+ versus *Crym*- astrocytes converge on GAT3 mediated GABA homeostasis. The Reviewer’s comments have made the study stronger – thank you.

Referee #2:

*“In this manuscript, Ollivier et al. detail the discovery of an anatomically distinct subpopulation of astroglia, defined by their expression of m-crystallin, a relatively unknown and unstudied marker. The authors report an important role for m-crystallin expression in human obsessive-compulsive disorder and Huntington’s disease, based on its downregulation in post-mortem human brain samples. They nicely combine electrophysiology, chemogenetics, and behavior in mice lacking *Crym* in central striatal astrocytes to characterize the role of *Crym*+ astroglia in neural regulation and behavior. The authors find remarkable synaptic regulation by *Crym*+ astroglia, such that knockdown of *Crym* expression leads to an increased E/I ratio of transmission into the central striatum. Mice lacking central striatal *Crym* expression in astroglia exhibit unique behavioral deficits that appear to reflect repetitive, perseverative behaviors, including grooming, spout licking, marble burying, etc. without changes in anxiety or motor defects. Thus, their findings not only reflect a critical role for astroglial m-crystallin in synaptic regulation, but also distinguish perseveration as a unique and distinct behavioral domain, independent from anxiety, and identify the circuitry that produces these behaviors. In all, this work is careful and complete; a comprehensive study that has produced clear outcomes*

*with far-reaching implications. The statistical analyses applied are appropriate and clearly described. This study is one of few that clearly identifies a fundamental physiological role for striatal astroglia in modulating behaviors that are disrupted in a number of psychiatric and neurological disorders. This study also highlights the value gained from detailed assessment of astrocyte subpopulations tuned to regulate specific neural subcircuits. Notably, the behavioral deficits following from *Crym* deletion in this population of glia is not corrected by fluoxetine, highlighting a relevant disease aspect that remains untreated by classic pharmacotherapies. Below are outlined a few points that, if discussed in greater detail, might further highlight the importance of these findings.”*

Thank you for these supportive comments and for the detailed review of our manuscript. We address your specific comments in turn below.

1) *“The authors show that *Crym* expression changes over the course of postnatal development. What is known regarding changes in *Crym* expression with age/senescence?”*

Thank you for this comment, which we address on pages 2-3 with the following sentences and related figures with new data on aged mice (22 months old).

“Striatal astrocyte μ -crystallin expression increased between postnatal day 7 and 15 (Extended data Fig. 2, 3, 4a-c), did not change significantly in mice 2, 12, and 22 months of age (Extended data Fig. 5a,b), and was identical in male and female mice (Extended data Fig. 5c,d).”

2) *“What is the relative decrease in *Crym* expression in HD compared with OCD? In what brain regions was decreased *Crym* observed in humans?”*

Thank you for this comment, which we address on page 2 with the following sentences.

*“Of these, *Crym* was similarly downregulated in caudate of OCD and HD¹⁷⁻¹⁹ to about 40% of control values and was within the top 4% of the downregulated genes, including in HD mouse models^{19,37,38}, implying potentially important functions of *Crym*+ astrocytes¹⁶ (Fig. 1a).”*

3) *“Does the dichotomy between astroglial and neuronal expression of *Crym* exist in humans as well? Are there data to demonstrate whether the decrease in *Crym* in HD and OCD occur in astroglia, neurons, or both?”*

In terms of greater astrocytic expression of *Crym*, the answer is yes, based on human data that are available. In relation to the second point, unfortunately, single cell and single nucleus RNAseq data are only available for HD to address whether *Crym* loss occurs in astrocytes and/or neurons. We address both comments on pages 8-9 with the following sentences.

*“Our data reveal that μ -crystallin loss leads to perseveration, which is of relevance to HD and OCD where *Crym* is reduced in postmortem human striatal tissue¹⁷⁻¹⁹. In the case of HD for which several gene expression studies are available, *Crym* downregulation increases with disease severity based on RNAseq of human caudate¹⁸ and striatal HD mouse model tissue³⁸. Similar to our findings, *Crym* was expressed in greater numbers of astrocytes than neurons in human tissue³¹, and it decreased in both cell types in postmortem human HD samples¹⁹ and in HD mouse models^{19,37}.”*

4) “Does *Crym* expression (in astroglia or neurons) appear to be sexually dimorphic? Is there any relationship with disease severity in the human samples?”

***Crym* is not sexually dimorphic and these new data are reported on pages 2-3 with the following sentences and related figures.**

“Striatal astrocyte μ -crystallin expression increased between postnatal day 7 and 15 (Extended data Fig. 2, 3, 4a-c), did not change significantly in mice 2, 12, and 22 months of age (Extended data Fig. 5a,b), and was identical in male and female mice (Extended data Fig. 5c,d).”

In terms of the question about disease severity in human samples, data are available only for HD. In that case, there is evidence that *Crym* loss increases with disease severity. On pages 8-9 we write

“In the case of HD for which several gene expression studies are available, *Crym* downregulation increases with disease severity based on RNAseq of human caudate¹⁸ and striatal HD mouse model tissue³⁸.”

In relation to comments 3 and 4 as a whole by Reviewer 2, we note that Reviewer 1 asked us to decrease the consideration of HD and OCD and to focus more on the mouse phenotypes (see comment 3 from Reviewer 1). In revising the manuscript, we thus struck a balance between comments from Reviewer 1 and 2. We believe the text changes mentioned above address Reviewer 2’s comments in full, but we provide Figure 1 below as additional supportive evidence. If Reviewer 2 feels it is needed and if the Editor agrees (given comments from Reviewer 1), we would be happy to include this as an Extended data Figure.

Minor

“In all figures, i and l panel denotations cannot be distinguished.”

Thank you. We fixed those typos.

Referee #3:

A+B Key results: Please summarise what you consider to be the outstanding features of the work.

“There are four outstanding features in this study:

1) This very complete work shades new lights on the role of a protein discovered decades ago. While founding biochemical studies showed that Crym has probable key roles in regulating thyroid hormones effects, the true novelty here is that the study demonstrates that Crym has a preferential expression profile in astrocytes in the striatum, and that through this preferential localization in these cells (and not neurons), Crym controls synaptic plasticity in this brain region.

2) The present study also brings another new concept linked to key trait of animal and human brain function, flexibility/inflexibility. Mind flexibility is central in adaptive behavior to environment /external stimulus in animal and human. The complex processes and molecular mechanisms underlying flexibility or inflexibility in pathological conditions are not known and have been often related to synaptic plasticity, involving mainly “neuronal” circuits”. Here, the authors show that a particular subset of astrocytes (expressing high levels of Crym) can regulate flexibility, which is really puzzling. This is probably one of the most significant scientific demonstration that molecular heterogeneity in astrocytes plays a direct role on complex brain functions (as considered for neurons). This likely will pave the way to future investigations on the role of molecular heterogeneity astrocytes of other brain regions/circuits and functions, such as procedural and spatial memory, decision making, etc.

3) The methodological approaches of the authors to come to this conclusion is mainly based on robust results from experiments in genetically modified mice and/or gene-transfer experiments with viral vectors for opto- and chemo-genetics. In addition, they provide correlational data using post mortem human brain samples from patients with well-recognized inflexibility traits showing that at least in part inflexibility/perseveration in these patients- could be due to loss of Crym in this subset of astrocytes

4) One major consequence of the present findings on a subset of astrocytes is that it also opens up new avenues in terms of therapeutic targets, not only for inflexibility-related pathological behaviors (such as seen in patients with Huntington’s diseases), but beyond, for other brain disorders resulting from the dysfunction of other subsets of astrocytes, yet to be discovered. In summary, the present work will be founding to a large spectrum of future studies on the role of subsets of astrocytes in brain health.”

Thank you for these supportive comments and for the detailed review of our manuscript. We address your specific comments in turn below.

Originality and significance:

“As mentioned, in the section/question above, I believe that this work is outstanding and of interest for all researchers in the field of neuroscience, neurology and psychiatry.”

Thank you for these supportive comments recognizing the broad relevance of our findings.

C- Validity: Does the manuscript have flaws which should prohibit its publication? If so, please provide details.

“Not that I could detect.”

Thank you; we have worked very hard for nearly 6 years to report robust findings and are pleased this was recognized.

Data & methodology: Please comment on the validity of the approach, quality of the data and quality of presentation. Please note that we expect our reviewers to review all data, including any extended data and supplementary information. Is the reporting of data and methodology sufficiently detailed and transparent to enable reproducing the results?

C – “The methodological approaches and the experimental design are robust. Main figures show an impressive set of data with appropriate controls. In addition, supplemental figures/data that are provided by the authors contains control experiments that significantly strengthen the results displayed in the main figures. The figures are of high quality in terms of presentation (organization, quality of drawing, image resolution, statistics etc.). Regarding the description of methods/approaches, it is fair to say that the Nature format does not permit an in depth description of the methods so that usually, the reader has to go back to previous publications by the authors or those from other research groups. From my point of view, for most aspects, there are sufficient details. For example, while going through the manuscript and reading the results the referee had concerns about the selective knockout of Crym in astrocytes. Regarding the exact description of AAVs used to locally delete Crym. Everything useful could be found in the M&M (promoter, backbone, construct pseudotypes etc.), addressing the referee questions on this precise point. Minor point: a scale bar legend should be included in all figures displaying histological images.”

Thank you. We added scale bars to every image as suggested. We also now referenced a detailed protocol paper on the proteomic methods and constructs that is now in press at *Nature Protocols*, which we believe will help the readers of this manuscript with the methodological aspects for proteomics.

D - Appropriate use of statistics and treatment of uncertainties: All error bars should be defined in the corresponding figure legends; please comment if that’s not the case. Please include in your report a specific comment on the appropriateness of any statistical tests, and the accuracy of the description of any error bars and probability values.

“The statistics were made in accordance to general guidelines to analyze biological data. Non parametric tests have been used for datasets with a distribution not satisfying normal distribution. When distribution of data was considered normal, parametric tests (student t test, one-way or two way factorial ANOVA have been applied with a statistical levels of 5%, which is the general rule. In many case, statistical levels are satisfying. This is clearly explained in materials and methods section and in figures. Graphs consistently show the mean +/-error bars corresponding to standard error of the mean (SEM). For biostatistics, using available tools (such String for protein interaction and enrichment analysis, as well as Enrichr), the authors described statistical values (FDR, False Discovery Rates) that are robust, i.e. $p < 0.05$)”

Thank you. We also point out that Supplementary Excel file 4 includes all the results of statistical tests used throughout the paper.

E - Conclusions: Do you find that the conclusions and data interpretation are robust, valid and reliable?

“YES, absolutely.”

Thank you.

Suggested improvements: Please list additional experiments or data that could help strengthening the work in a revision.

“There are two point that might be worth pursuing or at least clarify. 1. The exact mechanism through which alteration of Crym expression in the subset of astrocytes in the medial striatum leads to perseveration remains not yet totally understood. The present data indicate that it involves Crym+ astrocyte- mediated presynaptic glutamatergic terminals from the cerebral cortex (in particular the lateral orbitofrontal cortex). What is the molecular /chemical mediator of this effect? Is it only via modulation of glutamate release? If it is via modulation of glutamate release, what could cell signaling or cell-cell (i.e. astrocyte to neuron) mechanism(s) be at stake. Biochemical/cell experiments demonstrate that Crym is a thyroid T3 binding protein and transports/carries the hormone to the nucleus to activate gene expression. Crym also possesses an additional enzymatic activity, ketimine reductase (E.C. 1.5.1.25). Could this reductase activity of Crym (and thus the effects/roles of cyclic ketamine and catabolites) be involved in astrocyte- neuron functional interactions? In line with this, the authors performed GC/MS analysis of striatal samples from Crym KO and control mice. Do they have data related to ketimine pathway? Measuring ketimine metabolites (precursor/breakdown products) might be an interesting aspect of the effects of Crym KO.”

We have made many changes throughout the manuscript in response to this comment and those from Reviewer 1, providing both new data (Reviewer 1) and the requested clarifications (Reviewer 3). In brief, we explored mechanisms at molecular, synaptic, extracellular, and circuit levels and report the findings in the results, which includes several experiments requested by Reviewer 1. In summarizing these mechanisms, on page 8 we write

*“We discovered and studied a molecularly defined astrocyte population, precisely anatomically allocated, predominant in the central striatum, and identified by expression of μ -crystallin (*Crym*) – a protein of hitherto largely unknown functions in the brain^{16,33}. We explored molecular, synaptic, and neural circuit mechanisms of *Crym*+ astrocytes. At the molecular level, we provide data on how μ -crystallin works within astrocytes from the perspective of its interactome, which includes intracellular signal transduction and cytoskeletal binding proteins. These data also provide a basis for understanding the functions of μ -crystallin in striatal neurons^{37,39,40} in future work. Additional studies are needed to fully understand these interactions and the intracellular signaling cascades regulated by μ -crystallin^{16,33} (e.g. ketimine reductase). Such studies will benefit from the development of cell-specific metabolomic methods to assess biochemical pathways *in vivo*, which we could not explore in this study. At the synaptic level, using electrophysiology, pharmacology, and chemogenetics we found that *Crym*-positive astrocytes regulate neurotransmitter release probability of IOFC-to-striatum terminals. The extracellular signaling mechanism by which *Crym*+ astrocytes gate such phenotypes is through*

tonic GABA mediated presynaptic modulation of neurotransmitter release from IOFC terminals arriving onto MSNs within the central striatum. Thus, we found that *Crym*⁺ astrocytes within the central striatum contribute GABA to the extracellular space, whereas *Crym*⁻ astrocytes in the dorsolateral striatum remove it. At the circuit level, using cFos mapping, electrophysiology, behavioral analyses, and chemogenetics following reduction of μ -crystallin locally within striatal astrocytes in adult mice *in vivo*, we identified the IOFC-to-striatum circuit within the basal ganglia cortico-striatal-thalamo-cortical loop by which *Crym*⁺ astrocytes regulate perseveration.”

In terms of the clarification requested about the chemical mediator of the synaptic effect, our data show it is mediated by GABA. These new experiments were requested by Reviewer 1 and are summarized on page 5 with the following sentences

“During electrophysiological recordings, we noticed significantly decreased MSN tonic GABA currents in *Crym* KO mice, which indicates lower extracellular GABA levels in the central striatum (Fig. 3i,j). Furthermore, tonic GABA currents were blocked in control mice by pre-exposure to the astrocytic⁵⁵ GABA transporter type 3 (GAT3) antagonist⁵⁶ SNAP-5114 (Fig. 3i-k; 40 μ M). However, the reduced tonic GABA currents observed in *Crym* KO mice were spared (Fig. 3i-k), indicating that astrocytic GAT3 within the central striatum contributes GABA to the extracellular space^{56,57}, and that such contributions are reduced in *Crym* KO mice (Fig. 3i-k). Although there were no changes in GAT3 expression within astrocyte territories of *Crym* KO relative to controls (Extended data Fig. 15 a,b), we detected significant reduction in GABA and in monoamine oxidase B expression (MAOB; Extended data Fig. 15c-f). MAOB is an astrocytic enzyme⁵⁸⁻⁶⁰ that generates GABA, implying that reduced tonic GABA levels in *Crym* KO mice reflect reduced GAT3-dependent GABA contribution to the extracellular space as well as reduced astrocytic GABA.”

We appreciate the comment from Reviewer 3 about measuring ketimine metabolites and spent 3 months trying to do so. These experiments are summarized in Figure 2 below and show that the major metabolite of relevance (L-pipecolate) cannot be measured from *in vivo* samples with sufficient sensitivity. We have included positive controls that indicate we can measure L-pipecolate from standards, indicating that our equipment is capable of measuring it. However, the amount of L-pipecolate is too low within the striatum to quantify it above background levels. We also looked at the published literature and could not find instances when it had been measured directly or from *in vivo* samples. Based on these experiments, we conclude that with currently available GC/MS methods it is not possible to measure the key ketimine reductase metabolite, L-pipecolate. Nonetheless, we hope the Reviewer will agree that we have provided significant new mechanistic insight for how *Crym*⁺ astrocytes work at molecular, synaptic, neural circuit, and behavioural levels to warrant publication as summarized in the paragraphs above. We also do not dismiss the Reviewer’s comment and mention it specifically as the topic of future work with the following sentences on page 8.

“Additional studies are needed to fully understand these interactions and the intracellular signaling cascades regulated by μ -crystallin^{16,33} (e.g. ketimine reductase). Such studies will benefit from the development of cell-specific metabolomic methods to assess biochemical pathways *in vivo*, which we could not explore in this study.”

Reviewer Figure 2: L-Pipecolate is below the lower limit of quantitation in striatal tissue samples with standard GC/MS techniques. (A) *Top*: Chemical structure of L-pipecolate (L-pipecolic acid). *Bottom*: The predicted ionized derivative of L-pipecolic acid after treating samples with TBDMS, a standard procedure that forms silyl methyl esters on compounds to make them amenable for GC/MS analysis. (B) The mass spectrum of the ionized L-pipecolic acid derivative predicted by NIST indicates fragments at 198 m/z and 300 m/z. (C) The predicted fragments from (B) are identified when analyzing chemical standards of L-Pipecolic acid (Sigma #P2519). (*inset*) L-Pipecolic acid has a retention time of 32.64 min. (D) (*Left*) Nanomolar amounts of L-pipecolic acid standards are detected and in a linear range of quantitation. (*Right*) The lower end of the standard curve (corresponding to the red box on the left graph) shows that L-Pipecolic acid cannot be reliably quantified below 70 pmol. (E) (*Top*) In striatal samples (~5 mg, analogous to those used to conduct metabolomics in the main manuscript), L-pipecolate cannot be reliably detected (*Bottom*) As a positive control, pure chemical standards of L-pipecolate measured during the same experiment are can be detected. (F) (*Top*) Analysis of the data in (E) shows that L-pipecolate in striatal samples cannot be distinguished from experimental background and does not form a clean, measurable

Gaussian peak like the chemical standard (*Bottom*). (**G**) As a positive control, metabolites such as glutamate and lactate are readily detected in the same striatal samples as in (**E**). (**H**) Glutamate (Glu.), lactate (Lac.), N-acetyl-aspartate (NAA), and pyruvate (Pyr.) in striatal samples (~5 mg) are reliably quantified, while L-pipecolic acid (L-PA) is below the previously identified lower limit of detection of 70 pmol.

2. “*The interactors of Crym identified in the present work often have known preferential expression in astrocytes. However, the presence of MAPT (coding the protein Tau) which is considered to be more expressed in neurone than in astrocytes might appear a little bit confusing at first. The presence of MAPT in astrocyte interactome is also described in the paper by the same authors (Soto et al., Nature. 2023; 616(7958): 764–773). However, this is particularly interesting to consider that the interactome of Crym includes such a protein. The authors might comment or highlight more explicitly that the interactome described using the method used does not necessarily reflect the stoichiometry of proteins in one cell type but also (probably) the strength of interaction in subcompartments. The interaction between two proteins might also depend on the state of the interaction of others partners, since “binding” competition might exist between different partners.*”

Thank you for this comment. We now state that the μ -crystallin interactors reflects the strength of the interactions in subcompartments, as suggested. On page 7 we now write

“In accord with recent findings^{68,69}, the μ -crystallin interactome represents proteins in close proximity within subcompartments containing μ -crystallin and not their abundance within astrocytes as a whole. Thus, by mapping the interactome with subproteomes of major astrocyte physiological subcompartments such as near the plasma membrane, branches, and end feet⁶⁸, we found that 21 proteins were closely associated with μ -crystallin and not within other compartments (Fig. 6d). We note that the interaction between any protein and μ -crystallin is expected to depend on the totality of their interactions, since binding competition might exist between different partners.”

3. “*The present data convincingly demonstrate that the levels of Crym in a subset of astrocytes is much higher than levels seen in neurons? However transcriptomic studies in the past showed that striatal neurons also express Crym (at a lower degree) (Heinman M et al Cell 2018, 135(4):738-48; see supplementary table S5). In line with this, increasing Crym levels in MSN in the dorsal striatum is neuroprotective against mutant huntingtin (Francelle et al., Hum Mol Genet . 2015, 24(6):1563-73). Thus, it might be explicitly mentioned in the manuscript that Crym might also play an important role in neurons, although its level of expression in neuronal cells is much lower than that in astrocytes.*”

We had already stated in the previous submission that some striatal neurons also express Crym. As requested, we have now edited that section to make that point clearer on page 2 with the following section. We have also cited the Heiman et al. and Francelle et al., papers. Thank you for suggesting this.

“In accord with scRNAseq (Fig. 1b) and past studies^{37,39,40}, we also identified a population of μ -crystallin expressing neurons (in the subventricular zone (SVZ); Fig. 1e-h).”

As requested, we also now explicitly mention that μ -crystallin within neurons needs to be studied in future work with the following sentences on page 8.

“At the molecular level, we provide data on how μ -crystallin works within astrocytes from the perspective of its interactome, which includes intracellular signal transduction and cytoskeletal binding proteins. These data also provide a basis for understanding the functions of μ -crystallin in striatal neurons^{37,39,40} in future work.”

G. References: Does this manuscript reference previous literature appropriately? If not, what references should be included or excluded?

“See point # 3 above: Heinman M et al. *Cell* 2018, 135(4):738-48). Francelle et al. *Hum Mol Genet* 2015, 24(6):1563-73.”

Thank you. Those two papers have been cited in the sections mentioned above and throughout the revised manuscript.

H - Clarity and context: Is the abstract clear, accessible? Are abstract, introduction and conclusions appropriate?

“Yes, abstract is clear, accessible. Interpretation of results is fair so that conclusion are appropriate”

Thank you.

Overall, many thanks to all three Reviewers. Their comments have made the manuscript stronger and clearer.

References cited in this response

- 1 Siletti, K., Hodge, R., Mossi Albiach, A., Lee, K. W., Ding, S. L., Hu, L., Lönnerberg, P., Bakken, T., Casper, T., Clark, M., Dee, N., Gloe, J., Hirschstein, D., Shapovalova, N. V., Keene, C. D., Nyhus, J., Tung, H., Yanny, A. M., Arenas, E., Lein, E. S. & Linnarsson, S. Transcriptomic diversity of cell types across the adult human brain. *Science (New York, N.Y.)* **382**, eadd7046, doi:10.1126/science.add7046 (2023).
- 2 Lee, H., Fenster, R. J., Pineda, S. S., Gibbs, W. S., Mohammadi, S., Davila-Velderrain, J., Garcia, F. J., Therrien, M., Novis, H. S., Gao, F., Wilkinson, H., Vogt, T., Kellis, M., LaVoie, M. J. & Heiman, M. Cell Type-Specific Transcriptomics Reveals that Mutant Huntingtin Leads to Mitochondrial RNA Release and Neuronal Innate Immune Activation. *Neuron* **107**, 891-908, doi:10.1016/j.neuron.2020.06.021 (2020).
- 3 Hodges, A., Strand, A. D., Aragaki, A. K., Kuhn, A., Sengstag, T., Hughes, G., Elliston, L. A., Hartog, C., Goldstein, D. R., Thu, D., Hollingsworth, Z. R., Collin, F., Synek, B., Holmans, P. A., Young, A. B., Wexler, N. S., Delorenzi, M., Kooperberg, C., Augood, S. J., Faull, R. L., Olson, J. M., Jones, L. & Luthi-Carter, R. Regional and cellular gene expression changes in human Huntington's disease brain. *Human molecular genetics* **15**, 965-977, doi:10.1093/hmg/ddl013 (2006).
- 4 Diaz-Castro, B., Gangwani, M., Yu, X., Coppola, G. & Khakh, B. S. Astrocyte molecular signatures in Huntington's disease. *Sci Transl Med* **11(514)**. pii: eaaw8546. doi: 10.1126/scitranslmed.aaw8546 (2019).

Reviewer Reports on the First Revision:

Referees' comments:

Referee #1 (Remarks to the Author):

The authors have done an excellent job in responding to my critiques and those of the other reviewers. I have no further critiques and find this paper appropriate for publication in its current form. This is a fine study, that will have a profound impact on the field.

Referee #2 (Remarks to the Author):

The authors have sufficiently addressed all of my questions and concerns in their revised manuscript. Nicely done.

Referee #3 (Remarks to the Author):

The revised version of the manuscript by Ollivier et al. is stronger than the first version which was already exceptional and from my point of view new, with a major impact for our understanding of the functions of astrocytes in the brain, and methodologically robust.

One important point that the referee found a bit vague in the first draft concerned the mechanisms by which losing Crym could lead to perseverative behavior. The authors propose in this revised version an interesting possibility linked to GAT3 and the regulation of GABA concentrations in the medial striatum. This provides good evidence for a new (likely crucial) aspect of neuron-astrocyte interaction. Indeed, the regulation of glutamatergic neurotransmission has been the main focus of research on glia-neuron interactions. In addition, the new version allows the validation of Crym interactors by proximity ligation assays. The other points I raised were all addressed in depth (e.g. Crym in neurons, Crym interaction in a subcompartment, etc.), including an attempt to develop a new MS assay to measure concentrations of a major metabolite of ketimine, a compound whose synthesis is believed to involve Crym. I thank the authors for their efforts to experimentally address the possible role of this chemical.

In conclusion, going through the authors' responses to the reviewers, I believe the manuscript has been markedly strengthened. I see no flaws in this major discovery. It deserves to be published in Nature.

Author Rebuttals to First Revision:

Response to reviewers

We thank the reviewers for their careful review of our revised manuscript. There were no further points raised during review for us to address. The editor's changes are addressed in the cover letter to the editor.